



Atmospheric
Measurement
Techniques

# An improved TROPOMI tropospheric HCHO retrieval over China

**Wenjing Su**[1], **Cheng Liu**[2,3,4,5,6], **Ka Lok Chan**[7], **Qihou Hu**[2], **Haoran Liu**[4], **Xiangguang Ji**[2,8], **Yizhi Zhu**[2], **Ting Liu**[1], **Chengxin Zhang**[1], **Yujia Chen**[2], and **Jianguo Liu**[2]

[1]School of Earth and Space Sciences, University of Science and Technology of China, Hefei 230026, China
[2]Key Lab of Environmental Optics and Technology, Anhui Institute of Optics and Fine Mechanics,
Hefei Institutes of Physical Science, Chinese Academy of Sciences, Hefei 230031, China
[3]Center for Excellence in Regional Atmospheric Environment, Institute of Urban Environment,
Chinese Academy of Sciences, Xiamen 361021, China
[4]Department of Precision Machinery and Precision Instrumentation, University of Science and Technology of China,
Hefei 230027, China
[5]Key Laboratory of Precision Scientific Instrumentation of Anhui Higher Education Institutes,
University of Science and Technology of China, Hefei 230027, China
[6]Anhui Province Key Laboratory of Polar Environment and Global Change, USTC, Hefei 230026, China
[7]Remote Sensing Technology Institute (IMF), German Aerospace Center (DLR), Oberpfaffenhofen, Germany
[8]School of Environmental Science and Optoelectronic Technology, University of Science and Technology of China,
Hefei 230026, China

**Correspondence:** Cheng Liu (chliu81@ustc.edu.cn) and Ka Lok Chan (ka.chan@dlr.de)

**Abstract.** We present an improved TROPOspheric Monitoring Instrument (TROPOMI) retrieval of formaldehyde (HCHO) over China. The new retrieval optimizes the slant column density (SCD) retrieval and air mass factor (AMF) calculation for TROPOMI observations of HCHO over China. Retrieval of HCHO differential SCDs (DSCDs) is improved using the basic optical differential spectroscopy (BOAS) technique resulting in lower noise and smaller random error, while AMFs are improved with a priori HCHO profiles from a higher resolution regional chemistry transport model. Compared to the operational product, the new TROPOMI HCHO retrieval shows better agreement with ground-based Multi-AXis Differential Optical Absorption Spectroscopy (MAX-DOAS) measurements in Beijing. The improvements are mainly related to the AMF calculation with more precise a priori profiles in winter. Using more precise a priori profiles in general reduces HCHO vertical column densities (VCDs) by 52.37 % (± 27.09 %) in winter. Considering the aerosol effect in AMF calculation reduces the operational product by 11.46 % (± 1.48 %) and our retrieval by 17.61 % (± 1.92 %) in winter. The improved and operational HCHO are also used to investigate

the spatial–temporal characteristics of HCHO over China. The result shows that both improved and operational HCHO VCDs reach maximum in summer and minimum in winter. High HCHO VCDs mainly located over populated areas, i.e., Sichuan Basin and central and eastern China, indicate a significant contribution of anthropogenic emissions. The hotspots are more obvious on the map of the improved HCHO retrieval than the operational product. The result indicates that the improved TROPOMI HCHO retrieval is more suitable for the analysis of regional- and city-scale pollution in China.

## 1 Introduction

Formaldehyde (HCHO) is an important trace gas playing a crucial role in atmospheric chemistry processes. Hydroxyl radicals ($HO_x$=$OH + HO_2$) produced from HCHO photolysis show a strong influence on the oxidative capacity of the atmosphere (Li et al., 2011; Xue et al., 2016) and contribute to the formation of secondary organic aerosol (SOA) (Jang and Kamens, 2001). Atmospheric HCHO can be emit-

ted from both primary and secondary sources. Primary emissions include industrial sources, vehicle emissions (Wei et al., 2008), biogenic emissions from vegetation and biomass burning (Bauwens et al., 2016). The contribution of secondary sources to ambient HCHO sources is in general much higher than primary emissions, especially in summer (Su et al., 2019). Secondary HCHO is formed through the oxidation of almost all volatile organic compounds (VOCs). Therefore, it is usually regarded as an indicator of VOCs and is used to analyze sensitivity regimes of ozone ($O_3$) formation (Martin et al., 2004; Choi et al., 2012; Jin and Holloway, 2015; Liu et al., 2016). Accurate HCHO measurement is important for the investigation of HCHO and $O_3$ interactions and for the understanding of atmospheric chemistry processes.

Satellite observations provide indispensable information on HCHO spatial distribution. Satellite observations of HCHO have been conducted since 1996 with a series of satellite-borne instruments (Bovensmann et al., 1999; Martin et al., 2004; De Smedt et al., 2012; Barkley et al., 2013; González Abad et al., 2015, 2016). TROPOspheric Monitoring Instrument (TROPOMI) was launched on 13 October 2017. Compared to its predecessor satellite instruments, TROPOMI provides HCHO observations with a much higher spatial resolution with daily global coverage. The high-resolution satellite measurement enables us to analyze the finer scale spatiotemporal characteristics of HCHO.

In this study, we have improved the TROPOMI HCHO retrieval over China by optimizing the slant column density (SCD) retrieval and the air mass factor (AMF) calculation. Different from the operational product (De Smedt et al., 2018), our retrieval uses the basic optical differential spectroscopy (BOAS) technique for the differential SCD (DSCD) retrieval. In addition, the AMF calculation is improved by using higher resolution a priori profiles from the regional Weather Research and Forecasting model (WRF-Chem). It takes the fine-scale pollution into account, and the result is expected to be more realistic for the investigation of spatiotemporal variation of HCHO over China. Vigouroux et al. (2020) validated the operational TROPOMI HCHO product using ground-based solar-absorption Fourier transform infrared (FTIR) HCHO measurements. However, validation and improvement of TROPOMI HCHO observations over China are still necessary. Therefore, we have analyzed the quality of the operational HCHO product in detail and further improved TROPOMI HCHO observations over China.

The paper is organized as follows. Section 2 describes all data sets used in this study. Section 3 presents the improved TROPOMI HCHO retrieval algorithm over China. Uncertainty analysis in HCHO vertical column density (VCD) retrieval is discussed in Sect. 4. The comparisons to TROPOMI HCHO VCDs and ground-based Multi-AXis Differential Optical Absorption Spectroscopy (MAX-DOAS) are shown in Sect. 5. Finally, the summary and conclusions are drawn in Sect. 6.

## 2    Data sets

### 2.1    The TROPOMI instrument

TROPOMI is on board the Sentinel-5 Precursor (S5P) satellite. The S5P satellite orbits on the near-polar sun-synchronous orbit at an attitude of 824 km with a 17 d repeat cycle and Equator-crossing time of 13:30 local solar time (LST) on the ascending node. A scanning swath of TROPOMI covers a width of 2600 km, providing daily global coverage. TROPOMI has four spectrometers for medium-wave ultraviolet (UV), long-wave ultraviolet combined with visual (UVIS), near-infrared (NIR), and short-wave infrared (SWIR), covering nonoverlapping and noncontiguous wavelengths from 270 to 2385 nm, which are divided into eight spectral bands. Band 3, with a wavelength from 320 to 405 nm, is used for HCHO retrieval. Radiance in Band 3 is measured by the UVIS spectrometer. The detector for the UVIS spectrometer is a two-dimensional charge-coupled device (CCD), with one dimension for wavelengths and the other dimension for across-track spatial coverage (450 rows). Earth radiance is collected along the dayside of the earth, while solar irradiance measurements are performed near the North Pole every 15 orbits, approximately once a day. In the UVIS channel, the spectral resolution and spectral sampling are about 0.5 and 0.2 nm, respectively. Individual ground pixels' size of radiance measurement is approximately 3.5 km in the across-track direction and 7 km in the along-track direction (5.5 km since August 2019), with an integration time of 1.08 s (0.84 s since August 2019). The Level 1B radiance and solar irradiance are available in the Copernicus Open Access Hub (https://scihub.copernicus.eu/, last access: 22 May 2019).

### 2.2    Operational TROPOMI HCHO product

The operational TROPOMI HCHO product was jointly developed at the German Aerospace Center (DLR) and Royal Belgian Institute for Space Aeronomy (BIRA) (available at https://scihub.copernicus.eu/). The operational product is used for comparison with the improved TROPOMI HCHO data set over China. The operational product retrieves HCHO DSCDs using the DOAS spectral fitting technique in the wavelength range of 328.5–359 nm. The DSCD retrieval settings of the operational product are listed in Table 1. Using earthshine radiance over the remote Pacific Ocean as reference significantly reduces the influence from unresolved spectral structures which could significantly improve the spectral retrieval of weak absorbers, i.e., HCHO. Using radiance as reference reduces the fit residual as it already accounts for the $O_3$ absorption and the Ring effect. Daily detector row averaged radiance over the equatorial Pacific (latitude from 5° S to 5° N and longitude from 180 to 140° W) is used as reference spectra. Due to residual HCHO signals in reference, the differential SCD (DSCD) is retrieved in spec-

tra fitting. The conversion of DSCD to VCD uses the AMF approach. A priori HCHO profiles are taken from the global chemistry transport model Tracer Model 5 (TM5-MP) with a spatial resolution of $1° \times 1°$, with 34 vertical layers up to 0.1 hPa and spatiotemporally interpolated to the measurement time and location. Information about data sets used in AMF calculation is also listed in Table 1. In order to remove the residual HCHO signal in the reference spectra, zonal reference sector correction is applied following the equation

$$N_v = \left( N_s + N_{v,0,CTM} \times M_0 - N_{s,0} \right) / M, \tag{1}$$

where $N_v$ is the vertical column. $N_s$ is the uncorrected DSCDs. $N_{s,0}$ and $M_0$ are DSCD and AMF retrieved over the reference sector ([90° S, 90° N], [180 to 120° W]). $N_{v,0,CTM}$ is the simulated HCHO vertical column of the reference sector. The operational algorithm assumes $M_0$ is equal to M, and Eq. (1) can be expressed as

$$N_v = \left( N_s - N_{s,0} \right) / M + N_{v,0,CTM}. \tag{2}$$

The latitude dependency is approximated by a polynomial. DSCDs are averaged into a 5° latitude bin and subsequently used to approximate the coefficient of the polynomial. Detail of the S5P operational HCHO algorithm can be found in De Smedt et al. (2018).

## 2.3 Operational TROPOMI cloud product

Cloud parameters used in AMF calculation in our retrieval are from the operational TROPOMI cloud product. Cloud fraction is retrieved using the Optical Cloud Recognition Algorithm (OCRA), and cloud top height (pressure) and optical thickness (albedo) are retrieved using the Neural Networks (Retrieval of Cloud Information using Neural Networks, ROCINN) algorithm using the "Clouds-as-Reflecting-Boundaries" (CRB) model, treating clouds as simple Lambertian surfaces (Loyola et al., 2018). The operational TROPOMI cloud product is also available in the Copernicus Open Access Hub (https://scihub.copernicus.eu/).

## 2.4 WRF-Chem model

In our retrieval, the chemistry transport model WRF-Chem is used to simulate a priori HCHO profiles over China (Su et al., 2017; Zhang et al., 2019, 2020). Compared to TM5-MP, the regional WRF-Chem simulation has a higher spatial resolution of $20\,km \times 20\,km$. WRF-Chem simulates air pollution at 44 vertical layers extending from the ground up to 50 hPa. The initial and boundary conditions of the meteorological field for simulation are taken from the National Centers for Environmental Prediction (NCEP) 6 h Final Operational Global (FNL) reanalysis data, with spatial and temporal resolutions of $1° \times 1°$ and 6 h. The CBMZ (Carbon-Bond Mechanism version Z) photochemical mechanism combined with the MOSAIC (Model for Simulating

Aerosol Interactions and Chemistry) aerosol model was used to simulate the chemical processes in the atmosphere. WRF-Chem simulations also have a more up-to-date emission inventory over China. The anthropogenic and biogenic emissions are obtained from the Multi-resolution Emission Inventory for China (MEIC) and the Model of Emissions of Gases and Aerosols from Nature (MEGAN) (Guenther et al., 2006; M. Li et al., 2017), respectively. The MEIC emission inventory has improved the emissions estimation from power plants (Liu et al., 2015), vehicles (Zheng et al., 2014) and residential combustion of non-methane volatile organic compounds (NMVOCs) (Li et al., 2014; Peng et al., 2019). The open burning emission is obtained from the Fire INventory from NCAR (FINN) model (Wiedinmyer et al., 2011). The WRF-Chem simulation is carried out from July 2019 to July 2019, with 5 d spun up prior to the simulation.

## 2.5 MAX-DOAS HCHO measurements

MAX-DOAS HCHO measurements are used to validate TROPOMI HCHO observations in this study. The MAX-DOAS measurements are performed at three sites in Beijing (Fig. 1), including one urban and two suburban sites. One suburban site is located in Nancheng (NC) on the southern side of Beijing, while another suburban site is located at the University of Chinese Academy of Sciences (UCAS) on the northeastern side of Beijing. The urban site is located in the Chinese Academy of Meteorological Sciences (CAMS). The locations of three MAX-DOAS sites are indicated in Fig. S1 in the Supplement. The distances of the three MAX-DOAS sites, CAMS, NC and UCAS, from the city center of Beijing are 8, 27 and 61 km. These sites are representative of urban and suburban regions of Beijing. The MAX-DOAS instrument consists of a scanning telescope, a stepping motor controlling the viewing direction of the telescope and a spectrometer with a Hamamatsu back-thinned charge-coupled device (CCD) detector. The spectrometer measures scattered sunlight in the spectral range of 300–505 nm. Details of the MAX-DOAS measurement setup are shown in Table 2. Scattered sunlight spectra measured by MAX-DOAS are recoded and analyzed using DOAS Intelligent System (DOASIS) spectral fitting software (Kraus, 2006). Details of the MAX-DOAS HCHO DCSD retrieval settings are listed in Table S1. HCHO vertical profiles are retrieved using the Munich Multiple wavelength MAX-DOAS retrieval algorithm ($M^3$) (Chan et al., 2018). The algorithm is developed based on the optimal estimation method (Rodgers, 2000) and utilizes the radiative transfer model LibRadTran (Emde et al., 2016) as the forward model. In the MAX-DOAS profile retrieval, the lowest 1 km is divided into 10 layers, each with a thickness of 100 m, while the thickness of the layers between 1 and 3 km is set to 200 m. Details of MAX-DOAS HCHO profile retrieval can be found in Chan et al. (2018, 2019).

**Table 1.** The retrieval settings for TROPOMI HCHO DSCD and information used in AMF calculations in the operational product and our retrieval.

|  | Operational product (De Smedt et al., 2018) | Our retrieval |
|---|---|---|
| **SCD retrieval** | | |
| Algorithm | DOAS | BOAS |
| Fitting window | 328.5–359 nm | 328.5–359 nm |
| Radiance reference spectrum | Daily average of radiances of the Pacific orbit between 5° S and 5° N, 180 and 120° W | Daily average of radiances of the Pacific orbit between 30° S and 30° N, 180 and 140° W |
| Polynomial | Fifth order | Scaling polynomial: third order Baseline polynomial: third order |
| Instrument slit function | TROPOMI ISRF Calibration Key Data (CKD) v1.0.0 | TROPOMI ISRF Calibration Key Data (CKD) v3.0.0 |
| Solar reference spectrum | Chance and Kurucz (2010) | Chance and Kurucz (2010) |
| Ring effect | Ring cross section (Chance and Spurr, 1997) | Raman spectrum (Chance and Spurr, 1997) |
| HCHO cross sections | Meller and Moortgat (2000), 298 K | Chance and Orphal (2011), 300 K |
| $O_3$ cross sections | Serdyuchenko et al. (2014), 223 and 243 K | Malicet et al. (1995), 228 and 295 K, $I_0$ corrected |
| $NO_2$ cross sections | Vandaele et al. (1998), 220 K | Vandaele et al. (1998), 220 K |
| BrO cross sections | Fleischmann et al. (2004), 223 K | Wilmouth et al. (1999), 228 K |
| $O_4$ cross sections | Thalman and Volkamer (2013), 293 K | Thalman and Volkamer (2013), 293 K |
| **AMF calculations** | | |
| Altitude-dependent AMFs | VLIDORT, 340 nm, 6-D AMF lookup table | VLIDORT, 340 nm, 6-D AMF lookup table |
| Cloud parameter | S5P operational cloud product (see Sect. 2.2) | Same as left |
| Treatment of partly cloudy scenes | IPA, no correction for $f_{eff} < 10\%$ | Same as left |
| Surface albedo | OMI-based monthly minimum LER at 342 nm (Kleipool et al., 2008) | Same as left |
| A priori HCHO profile | TM5-MP daily forecast with the spatial resolution of $1° \times 1°$ (lat $\times$ long) | WRF-Chem daily simulations with the spatial resolution of 20 km $\times$ 20 km |
| Aerosols | No explicit correction | Same as left |

**Table 2.** Information on the MAX-DOAS measurements. The viewing azimuth angle of the north is taken as 0°. Relative error is calculated by mean of relative errors of MAX-DOAS measurements within $\pm 1$ h around the TROPOMI overpass time. TS1

| Site name | Location | Region | Height above sea level (m) | Viewing azimuth angle (°) | Elevation angles (°) | Relative error | Available days |
|---|---|---|---|---|---|---|---|
| CAMS | 39.94° N, 116.32° E | urban area | 100 | 130 (southeast) | 1, 2, 3, 4, 5, 6, 8, 10, 15, 30, 90 | 14.32 % | 352 |
| UCAS | 40.41° N, 116.67° E | suburb | 120 | 67 (northeast) | 1, 2, 3, 4, 5, 6, 8, 10, 15, 30, 90 | 14.03 % | 321 |
| NC | 39.78° N, 116.13° E | suburb | 60 | 48 (northeast) | 1, 2, 3, 4, 5, 6, 8, 10, 15, 30, 90 | 19.00 % | 331 |

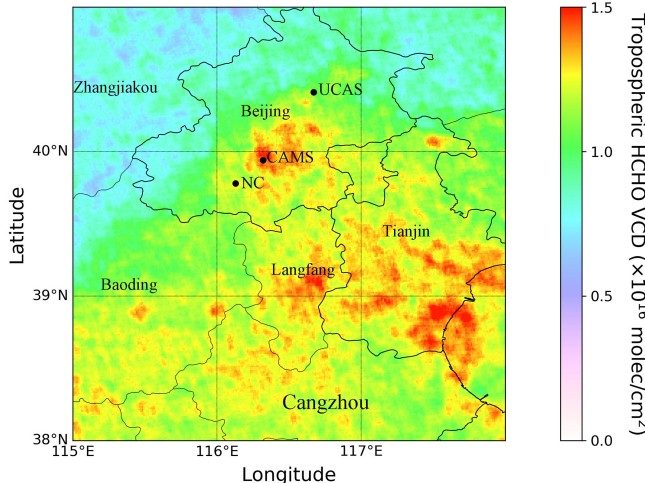

**Figure 1.** Annual average of the improved TROPOMI tropospheric HCHO VCD from August 2018 to July 2019 in Beijing and its surrounding region on a $0.01° \times 0.01°$ grid. The black dots indicate the locations of MAX-DOAS sites.

## 3 Improved HCHO retrieval algorithm

The retrieval of HCHO vertical column densities (VCDs) from TROPOMI observations can be separated into three major steps. The first step is the retrieval of HCHO DSCDs. The second step is the reference sector correction, converting DSCDs into SCDs. The final step is the conversion of SCD to VCD using AMF. Details of the improved HCHO retrieval algorithm are presented in the following.

### 3.1 HCHO DSCD retrieval

#### 3.1.1 Wavelength calibration

Measured solar irradiances and earthshine radiances are often slightly misaligned with the on-ground calibration due to temperature variation of the spectrograph, Doppler shift, nonuniform slit illumination due to presence of clouds or other high reflectance surface. To achieve accurate radiance fitting, it is necessary to calibrate the wavelength mapping before further processing. Irradiance and radiance wavelengths are calibrated using a high-resolution solar spectrum, with an accuracy of 0.001 nm (Chance and Kurucz, 2010). The high-resolution solar spectrum is convolved with the TROPOMI slit function to calculate the solar spectrum with the instrument resolution. This process can be described by Eq. (3):

$$I_{\mathrm{r}} = I_0^{\mathrm{h}} \otimes S(\lambda + \Delta\lambda) \times P_{\mathrm{s}}^{\mathrm{m}}(\lambda) + P_{\mathrm{b}}^{\mathrm{m}}(\lambda), \tag{3}$$

where $I_{\mathrm{r}}$ is the calculated solar spectrum with the instrument resolution. $I_0^{\mathrm{h}}$ is the high-resolution solar spectrum. $\lambda$ represents the wavelength. $\Delta\lambda$ indicates the wavelength shift parameter. $S$ is the TROPOMI slit function. $\otimes$ represents the convolution procedure. $P_{\mathrm{s}}^{\mathrm{m}}(\lambda)$ and $P_{\mathrm{b}}^{\mathrm{m}}(\lambda)$ are scaling

and baseline polynomials, respectively, which account for the low-frequency structures of the measured spectra. The polynomial is calculated as

$$P^{\mathrm{m}}(\lambda) = \sum_{i=0}^{\mathrm{m}} C_i (\lambda - \lambda_{\mathrm{avg}})^i, \tag{4}$$

where $\lambda_{\mathrm{avg}}$ is the center wavelength of the fitting window. $C_i$ is the coefficient of the fitted polynomial, and m is the order of polynomial. The full-width at half-maximum (FWHM) and asymmetric factor of the instrument slit function are obtained by fitting the $I_{\mathrm{r}}$ to the measured irradiance using the Gauss–Newton nonlinear least squares (NLLS) method assuming the asymmetric Gaussian shape of the slit function. The time series of the fitted TROPOMI slit function parameter from August 2018 to July 2019 in Fig. S2 shows that the TROPOMI slit function is stable after launch. Therefore, the preflight TROPOMI slit function is used in the wavelength calibration procedure. The preflight slit function is obtained from the TROPOMI Calibration Key Data (CKD) (available at http://www.tropomi.eu/data-products/isrf-dataset, last access: 22 May 2019) which are derived from TROPOMI calibration measurements performed in March 2015 at CSL in Liège. Comparing the spectral fit residual using different versions of the preflight slit function in the spectral fitting, we found that using v3.0.0 results in the lowest residual (Fig. S3). The preflight instrument slit function v3.0.0 is used in our retrieval, while the operational product uses the v1.0.0 preflight slit function. In the operational algorithm, polynomials are not considered in Eq. (3). In our retrieval, the third-order polynomials are selected through sensitivity analysis (Fig. S3). The result shows that using the third-order polynomials contributes to reducing the residual in the wavelength calibration. In the wavelength calibration procedure, wavelength shift and coefficients of the polynomials are obtained.

### 3.1.2 Radiance fitting

HCHO DCSDs are retrieved using the basic optical differential spectroscopy (BOAS) method (Chance, 1998), which has been applied to OMI and OMPS HCHO retrieval (González Abad et al., 2015, 2016). The BOAS method is based on the direct radiance fit. HCHO DSCDs are determined by fitting the simulated and measured radiance using the NLLS method. The simulated radiance $I_{\mathrm{s}}(\lambda)$ is calculated following Eq. (5):

$$I_{\mathrm{s}}(\lambda) = \left[ (a I_0(\lambda + \Delta\lambda) + \alpha_{\mathrm{r}} X_{\mathrm{r}}(\lambda)) \times e^{-\sum_j \alpha_j X_j(\lambda)} \right]$$
$$\times P_{\mathrm{s}}^{\mathrm{m}}(\lambda) + P_{\mathrm{b}}^{\mathrm{m}}(\lambda), \tag{5}$$

where $\lambda$ represents the wavelength, $a$ is the scaling factor of $I_0$ and $I_0(\lambda + \Delta\lambda)$ denotes the daily average earthshine radiance over the remote Pacific. Radiances measured 1 d before the processing day over the Pacific with latitudes ranging from 30° S to 30° N and longitudes ranging from 180

to 140° W are averaged and used as reference in the spectral fit. $\Delta\lambda$ indicates the wavelength shift parameter. $X_r(\lambda)$ refers to the Raman spectrum (Chance and Spurr, 1997), and $\alpha_r$ is the fitted coefficient to $X_r(\lambda)$. $\alpha_j$ refers to the cross section of trace gas $j$. Detail of TROPOMI HCHO DSCD retrieval setting is listed in Table 1. Sensitivity analysis is performed to optimize the absorption cross section used in the spectral fit. The sensitivity analysis results are shown in Fig. S4. As expected, considering all absorption cross sections listed in Table 1 significantly reduces fitting residual. Figure 2 shows the fitted optical depth and residual for two spectra measured on 6 August 2018 over China (orbit 4211). For the spectrum with higher HCHO DSCD, the HCHO absorption can be clearly distinguished in the fitting (see Fig. 2a). The HCHO absorption structures are less significant in the other spectrum as the HCHO DSCD is reaching the detection limit (see Fig. 2b). The detection limit of HCHO DSCD can be expressed by the ratio of the root mean square (rms) to the peak-to-peak optical density of HCHO in the fitting window (Schönhardt A, 2008). The rms of the fit residual for TROPOMI observation typically varies from $5 \times 10^{-4}$ to $1 \times 10^{-3}$, corresponding to the detection limits of HCHO DSCD from $\sim 7 \times 10^{15}$ to $\sim 1 \times 10^{16}$ molec cm$^{-2}$.

## 3.2 AMF calculation

HCHO SCDs represent the integration of HCHO concentration along the light path. Therefore, it is strongly dependent on the viewing and solar geometries, surface albedo, and the state of the atmosphere (presence of clouds, vertical HCHO distribution, pressure, temperature, etc.). The AMF approach is used to convert SCD to VCD. For optically thin species, the height-dependent sensitivity (box AMF) is insensitive to the vertical profile of the species. Therefore, tropospheric HCHO AMF ($M$) is calculated following the approach of Palmer et al. (2001), which can be described as

$$M = \frac{\int_{zs}^{zt} w \times n_a \mathrm{d}z}{\int_{zs}^{zt} n_a \mathrm{d}z}, \qquad (6)$$

where zs and zt are the altitude of surface and the tropopause, respectively. $n_a$ is the partial column of the corresponding layer of the a priori profile, which is taken from the regional WRF-Chem simulation. A priori HCHO profiles for TROPOMI AMF calculations are calculated by interpolating the WRF-Chem simulation spatiotemporally to the measurement time and location. The box AMF ($w$) represents the height-dependent sensitivity of the TROPOMI measurement. The box AMF depends on wavelength, solar zenith angle (SZA), viewing zenith angle (VZA), relative azimuth angle (RAA), surface albedo, surface pressure, cloud albedo, cloud fraction and cloud pressure. As the wavelength dependency of the box AMF in the HCHO fitting windows is rather small (less than 5 % for SZA < 70°), the box AMF can be calculated at a representative wavelength of 340 nm. In order to

improve the computational efficiency, the box AMF is precalculated using the radiative transfer model VLIDORT (version 2.6; Spurr, 2008) with a number of surface albedos, surface pressures and solar and viewing geometries and stored in a lookup table (LUT). The grid points of these parameters in creating the LUT are the same as in De Smedt et al. (2018). An aerosol-free U.S. standard atmosphere is assumed in the radiative transfer calculation of the box AMF. The box AMF within the LUT is interpolated into each particular observation condition. Linear interpolation is performed in solar and viewing geometries and surface albedo dimensions, and a nearest neighbor interpolation is performed in the surface pressure dimension. The box AMF at each particular observation condition is then linearly interpolated to the pressure level of the a priori HCHO profile.

For partly cloudy pixels, cloud correction is applied using the independent pixel approximation (Martin et al., 2002) in which scattering weights of inhomogeneous scenes are considered to be a linear combination of the box AMF for cloud-free scenes ($w_{clear}$) and cloudy scenes ($w_{cloud}$) following Eq. (7):

$$w(z) = (1 - CF_{iw}) \times w_{clear} + CF_{iw} \times w_{cloud}, \qquad (7)$$

where $CF_{iw}$ is the intensity-weighted cloud fraction, defined as

$$CF_{iw} = \frac{C_f \times I_{cloud}}{(1 - C_f) \times I_{clear} + C_f \times I_{cloud}}, \qquad (8)$$

where $C_f$ is the cloud fraction. $I_{cloud}$ and $I_{clear}$ are the radiance intensities for cloudy scenes and cloud-free scenes, respectively. Radiance intensities are precalculated using VLIDORT and saved in the LUT with the same setting as the box AMF LUT. Detailed information about data sets in AMF calculation is listed in Table 1.

## 3.3 Reference sector correction

Since there are residual HCHO signals in the earthshine reference spectra, an offset correction has to be applied to the retrieved DSCDs. The first step of reference sector correction is retrieving the DSCDs using the average earthshine radiance reference, calculating the corresponding AMFs and storing them as a separate database. The HCHO VCD over the remote Pacific Ocean is simulated by GEOS-Chem, assuming HCHO over this region is mainly from the oxidation of CH$_4$. The simulated HCHO SCD is calculated by multiplying the VCD (VCD$_G$) taken from GEOS-Chem with the corresponding AMF ($M_0$). Assuming HCHO in the reference sector correction is well simulated by GEOS-Chem, the difference between the simulated and retrieved DSCD (DSCD$_0$) is recognized as the DSCD bias caused by the residual HCHO signal in reference spectrum. TROPOMI measurements over the Pacific (latitudes from 90° S to 90° N and longitudes from 160 to 140° W) are first binned according to their latitude to 500 bins with a resolution of 0.36°. The median value of each

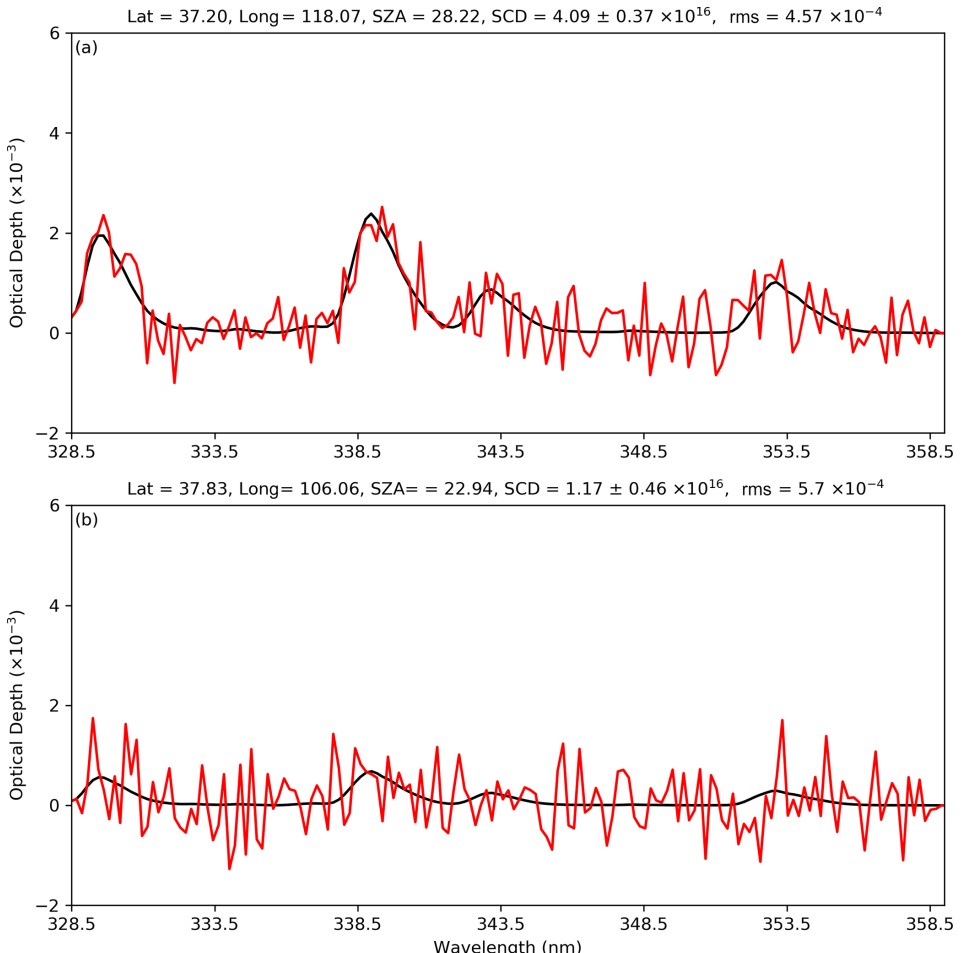

**Figure 2.** An example of the spectral retrieval of HCHO DSCD from two TROPOMI spectra measured on 6 August 2018 over China (orbit 4211), for **(a)** a polluted case and **(b)** a clean case. The black lines represent the simulated HCHO optical depth, and the red lines represent the fitted HCHO optical depth plus fitting residuals.

bin is then used for the calculation of the DSCD correction (González Abad et al., 2015, 2016). The DSCD correction (Corr(lat)) can be expressed as the following:

$$\text{Corr}(\text{lat}) = \text{Median}\left(\text{VCD}_\text{G}(\text{lat}) \times M_0 - \text{DSCD}_0\right). \quad (9)$$

Assuming the DSCD correction is constant in the longitudinal direction, the DSCD correction at 500 gridded latitude points is linearly interpolated to the latitude of each pixel over China. The interpolated DSCD correction is then applied on the retrieved DSCDs to calculate SCDs. Finally, the HCHO VCDs are calculated as follows:

$$\text{VCD} = \frac{\text{DSCD} + \text{Corr}(\text{lat})}{M} = \frac{\text{SCD}}{M}. \quad (10)$$

We improved the reference sector correction by considering the variability of the $M_0/M$ ratio.

## 4 Uncertainty analysis

Based on Eq. 10, the uncertainty of VCD can be derived analytically by uncertainty propagation. As VCD retrieval steps in Sect. 3 are performed independently, their uncertainties are assumed to be uncorrelated. The total uncertainty in HCHO VCD can be expressed following (Boersma et al., 2004; De Smedt et al., 2008)

$$
\begin{aligned}
\sigma_{N_\text{V}}^2 = {} & \left(\frac{\partial N_\text{V}}{\partial N_\text{S}}\right)^2 \sigma_{N_\text{S}}^2 + \left(\frac{\partial N_\text{V}}{\partial M}\right)^2 \sigma_M^2 \\
& + \left(\frac{\partial N_\text{V}}{\partial N_{\text{S},0}}\right)^2 \sigma_{N_{\text{S},0}}^2 + \left(\frac{\partial N_\text{V}}{\partial N_{\text{V},0,\text{G}}}\right)^2 \sigma_{N_{\text{V},0,\text{G}}}^2 \\
& + \left(\frac{\partial N_\text{V}}{\partial M_0}\right)^2 \sigma_{M_0}^2,
\end{aligned} \quad (11)
$$

where $N_\text{S}$ is retrieved DSCDs. $M$ and $M_0$ are both AMFs, while $M_0$ refers to AMF in the reference sector. $N_{\text{S},0}$ and $N_{\text{V},0,\text{G}}$ are retrieved DSCDs and simulated HCHO VCDs by

GEOS in the reference sector. $\sigma_{N_S}$ and $\sigma_M$ are the uncertainties on DSCD and AMF. $\sigma_{N_{S,0}}$, $\sigma_{N_{V,0},G}$ and $\sigma_{M_0}$ are the uncertainties on DSCD, simulated HCHO VCDs and AMF in the reference sector. Equation (11) can be transformed into the following equation:

$$\sigma_{N_V}^2 = \frac{1}{M^2} \left( \sigma_{N_S}^2 + \frac{N_S - N_{S,0} + N_{V,0,G} \times M_0}{M^2} \sigma_M^2 \right.$$
$$\left. + \sigma_{N_{S,0}}^2 + M_0^2 \sigma_{N_{V,0},G}^2 + N_{V,0,G}^2 \sigma_{M_0}^2 \right). \tag{12}$$

## 4.1 Uncertainties in DSCDs

Uncertainties of DSCD retrieval can be separated into random and systematic uncertainties, and they can be expressed as the following equation:

$$\sigma_{N_S}^2 = \sigma_{N_{S,rand}}^2 + \sigma_{N_{S,sys}}^2. \tag{13}$$

Random uncertainties are mainly related to instrument noise. Random uncertainties can be approximated by the root mean square (rms) of the spectral fitting residual, the degrees of freedom and the diagonal term of the covariance matrix for HCHO ($C_{j,j}$):

$$\sigma_{N_{S,rand}}^2 = \text{rms}^2 \frac{m}{m-n} \mathbf{C}_{j,j}^2, \tag{14}$$

where $m$ is the number of spectral pixels and $n$ is the number of fitted parameters. $j$ is the index of the HCHO cross section in fitting parameters. The covariance matrix ($\mathbf{C}_{j,j}$) is calculated from the Jacobian of the forward model corresponding to the fitting parameters (Chan Miller et al., 2014). The mean rms values for measurements taken on 6 August 2018 over the region of 73–130° E, 18–54° N and the corresponding random uncertainty are $5.52 \times 10^{-4}$ and $0.49 \times 10^{16}$ molec cm$^{-2}$, respectively. Systematic uncertainties are mainly from absorption cross sections, choice of fitting window, choice of polynomials and TROPOMI slit function used in the spectral fit. Applying reference sector correction helps to reduce systematic uncertainties significantly. Differences in HCHO SCDs are used to estimate systematic uncertainties. We perform sensitivity analysis to evaluate these effects on HCHO DSCD and SCD retrieval (Table 3). Fitting results using five different fitting windows which are used in previous satellite HCHO retrievals (De Smedt et al., 2008, 2012; González Abad et al., 2015) are compared to our result. The effect of using the Taylor series of the O$_3$ cross section in the spectral retrieval (Puîte et al., 2010) is also evaluated through sensitivity analysis (Table 3). After reference sector correction, systematic differences relating to spectral fitting window, polynomials, slit function and Taylor series of O$_3$ cross section (Puîte et al., 2010) are estimated to be 15.11 %, 2.33 %, 2.33 % and 3.49 %, respectively. Put together with the systematic uncertainty (7 %) caused by absorption cross sections used in the spectral fit (González Abad et al., 2016), we estimated the total systematic uncertainty to be about 17.3 %.

## 4.2 Uncertainties in AMF calculations

According to Eq. (6), the uncertainties of AMF calculations can be approximated by uncertainty propagation following the equation below.

$$\sigma_M^2 = \left( \frac{\partial M}{\partial a_s} \sigma_{a_s} \right)^2 + \left( \frac{\partial M}{\partial c_f} \sigma_{c_f} \right)^2 + + \left( \frac{\partial M}{\partial c_p} \sigma_{c_p} \right)^2$$
$$+ \left( \frac{\partial M}{\partial p_h} \sigma_{p_h} \right)^2, \tag{15}$$

where $\sigma_{a_s}$, $\sigma_{c_f}$, $\sigma_{c_p}$ and $\sigma_{p_h}$ represent the uncertainties of surface albedo ($a_s$), cloud fraction ($c_f$), cloud pressure ($c_p$) and a priori HCHO profile height ($p_h$), respectively. A priori HCHO profile height is defined as 75 % HCHO being located below this altitude (De Smedt et al., 2018). The observation geometries are measured with high accuracy; therefore, their uncertainties toward AMF uncertainties are negligible. We estimated AMF uncertainties related to surface albedo, cloud fraction, cloud pressure and profile height through sensitivity analysis with a fixed set of observation geometry (SZA = 30°, VZA = 30°, RAA = 0°) (Fig. 3). Through the sensitivity analysis, we estimated the VCD uncertainty caused by surface albedo, cloud top pressure, cloud fraction and profile height to be 0 %–9 %, 3 %–10 %, 1 %–15 % and 1 %–16 % for the clean case and 0 %–10 %, 3 %–15 %, 1 %–11 % and 7 %–31 % for the polluted case, respectively. Details of the error analysis are summarized in Table 4. The VCD uncertainty from the wavelength dependency of AMF, LUT error, cloud correction and no explicit aerosol correction is about 15 %–35 % (De Smedt et al., 2018). Considering the above uncertainty sources, the VCD uncertainties from AMF calculation are about 17 %–51 % and 15 %–43% in polluted and clean cases.

## 4.3 Uncertainties in reference sector correction

Uncertainty in reference sector correction ($\sigma_{N_{V,0}}$) is caused by HCHO VCD simulation, AMF calculation and SCD retrieval in the reference sector. The uncertainty can be expressed as

$$\sigma_{N_{V,0}}^2 = \frac{1}{M^2} \left( \sigma_{N_{S,0}}^2 + M_0^2 \sigma_{N_{V,0},G}^2 + N_{V,0,G}^2 \sigma_{M_0}^2 \right). \tag{16}$$

We take into account the mean standard deviation of simulated HCHO VCD in the reference sector to estimate the uncertainty from model simulation. The uncertainty from VCD simulation is $1.43 \times 10^{16}$ molec cm$^{-2}$. The uncertainties from SCD retrieval and AMF calculations in the reference sector are calculated the same as in Eqs. (13) and (15).

**Table 3.** Parameter effects on daily mean HCHO DSCDs and SCDs, $\pm 1$ standard deviation of DSCDs ($\pm 1\sigma_D$) and SCDs ($\pm 1\sigma_S$) and mean root mean square (rms) of spectral fitting residual on 6 August 2018 of the region between 73 and 130° E and 18 and 54° N. Units of DSCDs, SCDs, $\pm 1\sigma_D$ and $\pm 1\sigma_S$ are $\times 10^{16}$ molec cm$^{-2}$. The rms units are $\times 10^{-4}$. The detailed retrieval settings for the reference case are listed in the column with the header "Our retrieval" in Table 1. The rows below "Reference case" with specific settings are modified with respect to the reference case, while all other settings are kept unchanged.

| Retrieval settings | DSCDs | $\pm 1\sigma_D$ | rms | SCDs | $\pm 1\sigma_S$ |
|---|---|---|---|---|---|
| Reference case | 0.34 | $\pm 0.57$ | 5.84 | 0.86 | $\pm 0.58$ |
| 328.5–346.0 nm | 0.46 | $\pm 0.96$ | 5.83 | 0.87 | $\pm 0.96$ |
| 328.5–356.5 nm | 0.38 | $\pm 0.62$ | 5.88 | 0.88 | $\pm 0.63$ |
| 337.0–353.0 nm | 0.49 | $\pm 1.65$ | 5.69 | 0.99 | $\pm 1.65$ |
| 337.5–359.0 nm | 0.40 | $\pm 0.99$ | 5.84 | 0.82 | $\pm 0.99$ |
| 334.0–348.0 nm | 0.28 | $\pm 2.43$ | 5.60 | 0.74 | $\pm 2.43$ |
| Fourth-order baseline and scaling polynomials | 0.41 | $\pm 0.65$ | 6.00 | 0.88 | $\pm 0.67$ |
| TROPOMI ISRF Calibration Key Data (CKD) v1.0.0 | 0.34 | $\pm 0.61$ | 6.04 | 0.88 | $\pm 0.61$ |
| The first-order Taylor series approach for O$_3$ SCDs (Puïte et al., 2010) | 0.33 | $\pm 0.63$ | 6.04 | 0.83 | $\pm 0.65$ |

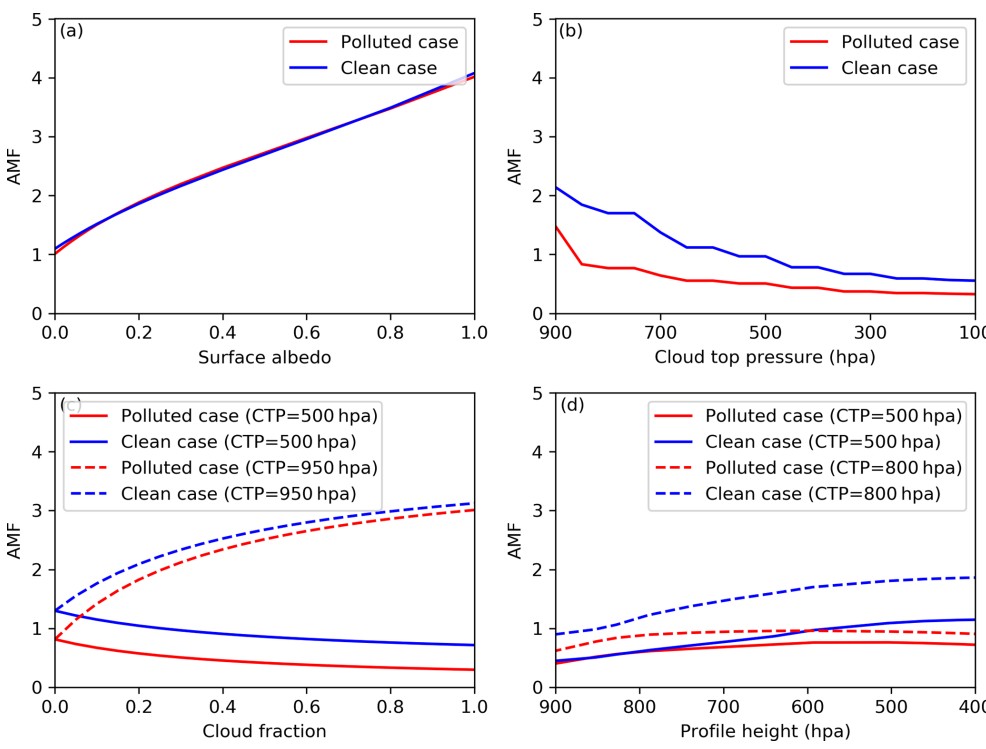

**Figure 3.** AMF variations with surface albedo **(a),** cloud top pressure **(b)**, cloud fraction **(c)** and profile height **(d)** over clean and polluted cases. In panel **(a)**, cloud fraction is 0. In panels **(b)** and **(d)**, cloud fractions are 0.3, and surface albedos are 0.05.

## 5 Results and discussions

### 5.1 Comparison of operational and improved HCHO product

To analyze the improvement of the new TROPOMI HCHO product, we compared the new HCHO data set to the opera-tional product. The contribution of SCD retrieval and AMF calculation are investigated separately. Both the improved and operation data sets are compared to the MAX-DOAS HCHO measurements to quantify the improvements. The normalized mean bias (NMB) between satellite and MAX-DOAS data is used as the benchmark in the comparison.

**Table 4.** Uncertainties of TROPOMI HCHO VCD from the parameters used in AMF calculation. Uncertainties are estimated with a fixed set of observation geometry (SZA = 30°, VZA = 30°, RAA = 0°).

| Uncertainty source | Clean area | Polluted area |
|---|---|---|
| Surface albedo | 0 %–9 % | 0 %–10 % |
| Cloud top pressure | 3 %–10 % | 3 %–15 % |
| Cloud fraction | 1 %–15 % | 1 %–11 % |
| Profile height | 1 %–16 % | 7 %–31 % |
| AMF LUT and no explicit aerosol correction | 15 %–35 % (De Smedt et al., 2018) | 15 %–35 % (De Smedt et al., 2018) |
| Total | 15 %–43 % | 17 %–51 % |

NMBs between two data sets can be calculated following

$$
\mathrm{NMB} = \frac{\sum\limits_{i=1}^{i=n} (V_{\mathrm{T}}(i) - V_{\mathrm{M}}(i))}{\sum\limits_{i=1}^{i=n} V_{\mathrm{M}}(i)} \times 100\,\%, \qquad (17)
$$

where $V_{\mathrm{T}}(i)$ and $V_{\mathrm{M}}(i)$ are the average tropospheric HCHO VCD measured by TROPOMI and MAX-DOAS on day $i$, respectively. The standard error (SE) is calculated by dividing the standard deviation (SD) by the square root of the sample size (SE = SD/$\sqrt{n}$); 2 times the SE (95 % confidence interval; Streiner, 1996) is regarded as the error of the NMBs in this study. The NMBs larger than the error are considered statistically significant. The error of NMB is calculated following TS2

$$
\mathrm{Error}_{\mathrm{NMB}} = 2 \times \mathrm{SE}
$$

$$
= 2 \times \sqrt{\frac{1}{n(n-1)} \frac{\sum\limits_{i=1}^{i=n} \left( V_{\mathrm{T}}(i) - V_{\mathrm{M}}(i) - \overline{V_{\mathrm{T}}(i) - V_{\mathrm{M}}(i)} \right)^2}{\left( \sum\limits_{i=1}^{i=n} V_{\mathrm{M}}(i)/n \right)^2}}
$$

$$
\times 100\,\%. \qquad (18)
$$

MAX-DOAS measurements are temporally averaged within ± 1 h around the TROPOMI overpass time, while TROPOMI pixels within 20 km of the MAX-DOAS site are spatially averaged for comparison. TROPOMI pixels in our retrieval and operational product are both filtered for intensity-weighted cloud fraction smaller than 0.3, a root mean square (rms) value of the spectral fit residual smaller than $10^{-3}$, AMF larger than 0.1 and SZA smaller than 70°, quality assurance (QA) value larger than 0.55 and successful SCD retrieval. The histograms showing the distributions of rms values of the spectral fit of the operational product and our retrieval on 6 August 2018 over China are shown in Fig. S5. About 15 % and 17 % of measurements in the operational product and our retrieval show rms values larger than $10^{-3}$.

### 5.1.1 SCD retrieval

In this section, we compare DSCDs and SCDs after applying reference sector correction between the operational product and the new retrieval. The improved TROPOMI HCHO is retrieved using the BOAS approach, while the operational product uses the DOAS technique to retrieve the HCHO DSCDs. Both data sets are filtered for rms values smaller than $10^{-3}$. Figure 4a and c shows the TROPOMI HCHO DSCDs retrieved by BOAS and DOAS methods. HCHO DSCDs from both data sets show a very similar spatial pattern over China, with higher values over industrialized regions such as the Beijing–Tianjin–Hebei region (BTH), Yangtze River Delta (YRD), Pearl River Delta (PRD) and Sichuan Basin (SCB). However, the DOAS DSCDs show more scattered outliers, while the BOAS data set shows a smoother appearance. As the anthropogenic VOC emission in Tibet is small (Wu et al., 2016; H. Li et al., 2017), and the oxidation of VOCs is a significant source of HCHO, HCHO in this region is expected to have small spatial variation. The averaged HCHO DSCD of the BOAS retrieval is $0.076 \times 10^{16}$ molec cm$^{-2}$, with a standard deviation (SD) of $0.51 \times 10^{16}$ molec cm$^{-2}$ over Tibet (area indicated by blue line), while the DOAS data set shows a higher averaged column of $0.34 \times 10^{16}$ molec cm$^{-2}$ as well as a standard deviation of $0.66 \times 10^{16}$ molec cm$^{-2}$. The numbers of valid satellite measurements over Tibet for the BOAS and DOAS retrieval are 22 244 and 21 987, respectively. Larger standard deviation implies the data set contains more outliers. The DOAS HCHO DSCDs are rather noisy over Shaanxi province and its surrounding regions, while BOAS DSCDs appear to be much smoother with a significant hotspot over Xi'an (provincial capital of Shaanxi). As the spatial distribution of HCHO is expected to be smooth, less noisy DSCDs indicate that the BOAS technique is less sensitive to measurement noise.

The scatterplot of the BOAS and DOAS retrieval of HCHO DSCDs over China on 6 August 2018 is in Fig. 5a. The result shows that BOAS and DOAS retrievals agree well

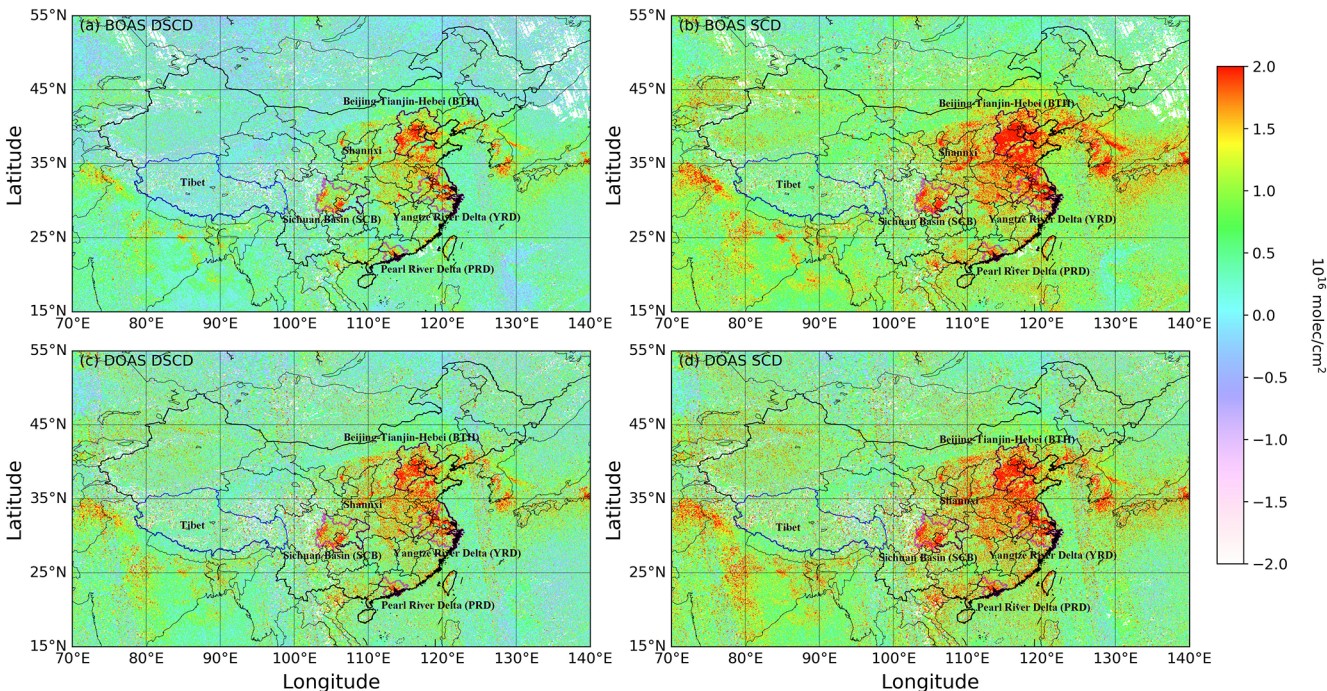

**Figure 4.** Spatial distributions of BOAS **(a)** and DOAS **(c)** HCHO DSCDs and BOAS **(b)** and DOAS **(d)** HCHO SCDs on 6 August 2018 over China from orbit 04210 to orbit 04213. The regional boundaries of BTH, YRD, PRD and SCB are delineated by magenta lines.

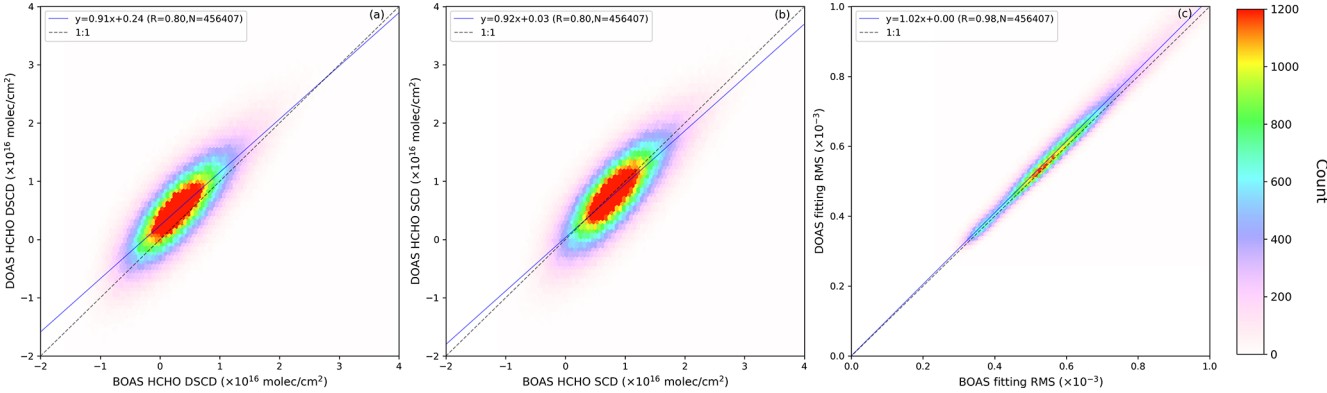

**Figure 5. (a)** Pixel-to-pixel comparisons of DOAS and BOAS HCHO DSCDs, **(b)** DOAS and BOAS HCHO SCDs and **(c)** DOAS and BOAS fitting rms values on 6 August 2018 in the region between 73 and 130° E and 18 and 54° N.

with each other, with a Pearson correlation coefficient ($R$) of 0.80. The slope and offset of the linear least squares regression line are 0.91 and $0.24 \times 10^{16}$ molec cm$^{-2}$, respectively. Averaged DSCDs on 06 August 2018 over China retrieved by DOAS and BOAS algorithms are 0.57 ($\pm 0.0020$) $\times 10^{16}$ and 0.36 ($\pm 0.0018$) $\times 10^{16}$ molec cm$^{-2}$, respectively. Averaged DSCD retrieved by the DOAS method is 58.33 % higher ($0.21 \pm 0.0012 \times 10^{16}$ molec cm$^{-2}$) than the BOAS retrieval. On the other hand, rms comparisons shown in Fig. 5c indicate that the average rms of the DOAS retrieval ($6.1 \pm 0.003 \times 10^{-4}$) is slightly higher than that of the BOAS retrieval ($5.95 \pm 0.003 \times 10^{-4}$). The biases between DSCDs

and rms values in the operational product and our retrieval are mainly related to the difference of retrieval method, retrieval settings and selection of reference.

To investigate the impact of the selection of earthshine radiance reference, we retrieved HCHO DSCDs using daily detector row averaged radiance over the equatorial Pacific (latitude from 5° S to 5° N and longitude from 180 to 120° W) as reference using the BOAS method. Comparisons of HCHO DSCDs and HCHO SCDs after applying reference sector correction using different earthshine radiance reference are presented in Fig. S6. The retrieved DSCDs using different earthshine radiance reference corre-

lated well ($R = 0.99$, slope $= 0.99$) and had a difference of 30 % ($0.11 \pm 0.0002 \times 10^{16}$ molec cm$^{-2}$). The bias is compensated for (1.9 %) by the reference sector correction (Fig. S6b). To investigate the impact of the retrieval method, BOAS HCHO DSCDs using the same retrieval settings as the operational DSCD retrieval are compared with DOAS HCHO DSCDs (Fig. S7). Using the same retrieval settings, the difference between DOAS HCHO DSCDs and BOAS HCHO DSCDs (27.33 %) is significantly reduced, and the remaining difference is due to the retrieval algorithm. In addition, a smaller difference (4.41 %) in HCHO SCDs indicates that reference sector correction reduces the effect of the retrieval method (Fig. S7b).

HCHO SCDs after applying reference sector correction in the operational product and in our retrieval also show good agreement, with a Pearson correlation coefficient ($R$) of 0.80 (Fig. 5b). The slope and intercept of the linear least squares regression line are 0.92 and $0.03 \times 10^{16}$ molec cm$^{-2}$. BOAS and DOAS HCHO SCDs also show a very similar spatial pattern, and the biases between BOAS and DOAS HCHO SCDs reduce after background correction (Fig. 4b and d). Averaged SCD taken from the operational product on 06 August 2018 over China ($0.85 \pm 0.0020 \times 10^{16}$ molec cm$^{-2}$) is on average 4.49 % lower than our retrieval ($0.89 \pm 0.0018 \times 10^{16}$ molec cm$^{-2}$).

To evaluate improvement of the SCD retrieval, we updated the operational product by using BOAS HCHO SCDs and keeping the AMF unchanged. The NMBs of the updated HCHO VCD and operational HCHO VCD are shown in Table 5. The bias between the two data sets is caused by a difference in SCD retrieval. The updated and operational product is also compared to the MAX-DOAS observations. In summer, using different SCD retrieval methods results in a difference of 7.00 % ($\pm 1.71$ %) from the TROPOMI operational HCHO VCD. The result shows that using the BOAS HCHO SCDs reduced the underestimation in summer and overestimation in winter of the operational product (Table 5). Additionally, the mean random errors relative to BOAS are about 22 % lower than DOAS at three MAX-DOAS sites. The result indicates that the BOAS technique provides a slightly more stable retrieval of HCHO SCDs.

### 5.1.2 AMF calculation

The improved TROPOMI HCHO over China uses a priori profile information from the higher resolution regional WRF-Chem simulation, while the operational product uses the a priori profile from the global TM5-MP model with $1° \times 1°$ resolution. In AMF calculations, both WRF-Chem and TM5-MP simulations are interpolated to TROPOMI spatial resolution. Interpolated WRF-Chem and TM5-MP simulations within 20 km of the MAX-DOAS site are spatially averaged to compare with MAX-DOAS profiles. MAX-DOAS profiles are temporally averaged in the period 13:30–14:30 (local time) within $\pm 1$ h around the TROPOMI overpass time.

Figure 6a and c show daily averaged a priori HCHO profiles from WRF-Chem and TM5-MP simulations and MAX-DOAS measured profiles. In addition to higher horizontal resolution, the WRF-Chem simulation also has a higher vertical spatial resolution compared to the TM5-MP data set. Although the difference in vertical resolution only shows a negligible effect on the box AMF under clear sky conditions (Fig. 6b), lower vertical resolution profiles would cause a significant impact for cloudy cases due to the interpolation of the coarse grid (Fig. 6d). Figures 7 and 8 show comparisons of seasonal averaged a priori HCHO profiles from MAX-DOAS, WRF-Chem and TM5-MP simulations at urban (CAMS) and suburban (UCAS) sites in spring (March, April, May), summer (June, July, August), autumn (September, October, November) and winter (December, January, February). A priori HCHO profiles simulated by WRF-Chem are similar to MAX-DOAS measurements, while TM5-MP a priori profiles show a larger difference from the MAX-DOAS measurements. The improvement of WRF-Chem simulations at urban sites is more significant than at suburban sites, which is mainly related to the finer spatial resolution and more up-to-date emission inventory over China used in the simulations. The bias between simulated and measured HCHO profiles at urban sites is larger than that at suburban sites, which is mainly due to a smaller spatial gradient over suburban areas. The AMFs calculated with WRF-Chem and TM5-MP HCHO profiles as a priori information are shown in Fig. 9. The results show a similar spatial pattern. However, WRF-Chem AMF is mostly higher than the one calculated with TM5-MP HCHO profiles. The big differences of two AMFs mainly occur in pixels with large cloud fraction. To evaluate the influence of the a priori HCHO profile on VCD retrieval, we updated the operational product by keeping the SCDs unchanged and used the a priori profile from WRF-Chem for the AMF calculation. The AMF is calculated following Sect. 3.2 and using the same parameters as the operational algorithm except for the a priori HCHO profile. The NMBs between the updated and operational HCHO and MAX-DOAS HCHO are shown in Table 5. The effect of the a priori HCHO profile is much larger than that of the SCD retrieval in winter. Using the a priori HCHO profile from WRF-Chem reduces the bias between TROPOMI and MAX-DOAS by 11.48 % ($\pm 8.39$ %) in summer and 52.37 % ($\pm 27.09$ %) in winter. The reduction at urban sites in winter is larger than at suburban sites.

### 5.2 Comparison between HCHO VCDs observed by MAX-DOAS and TROPOMI

To validate TROPOMI tropospheric HCHO VCD, TROPOMI HCHO VCDs are compared to the MAX-DOAS measurements. Figure 10a, c and e show the comparison of TROPOMI HCHO VCDs from our retrieval (red lines) and the operational product (green lines) to the MAX-DOAS HCHO VCDs at three MAX-DOAS

**Table 5.** Normalized mean biases (NMBs) between the updated and operational TROPOMI HCHO VCDs ($NMB_{U,O}$) and NMBs between TROPOMI and MAX-DOAS observations ($NMB_{T,M}$). TROPOMI HCHO VCDs are updated with three different settings: (1) replacing DOAS SCDs using BOAS SCDs in the operational product, (2) changing the a priori profiles from TM5-MP to regional WRF-Chem simulations in the operational product and (3) changing both (1) and (2) in the operational product. The error bars ($\pm\,Error_{NMB}$) are also presented. The NMBs and theirs errors are calculated following Eqs. (17) and (18), respectively. All values are in %.

| Settings | | CAMS | | UCAS | | NC | |
|---|---|---|---|---|---|---|---|
| | | $NMB_{U,O}$ | $NMB_{T,M}$ | $NMB_{U,O}$ | $NMB_{T,M}$ | $NMB_{U,O}$ | $NMB_{T,M}$ |
| Operational product | Year | – | $-8.23 \pm 3.09$ | – | $10.75 \pm 4.82$ | – | $3.73 \pm 3.77$ |
| (DOAS & TM5-MP) | Summer | – | $-14.92 \pm 5.03$ | – | $-21.96 \pm 5.31$ | – | $-18.12 \pm 5.73$ |
| | Winter | – | $20.28 \pm 8.74$ | – | $44.10 \pm 11.27$ | – | $22.87 \pm 8.82$ |
| BOAS & TM5-MP | Year | $0.20 \pm 3.24$ | $-9.85 \pm 4.57$ | $-0.43 \pm 2.95$ | $15.33 \pm 4.84$ | $-1.50 \pm 2.24$ | $2.48 \pm 3.59$ |
| | Summer | $3.91 \pm 1.89$ | $-11.69 \pm 4.94$ | $10.00 \pm 3.90$ | $-14.16 \pm 5.01$ | $7.20 \pm 2.74$ | $-12.23 \pm 5.33$ |
| | Winter | $-2.67 \pm 4.31$ | $18.98 \pm 10.15$ | $-4.15 \pm 11.54$ | $36.19 \pm 18.44$ | $-3.75 \pm 9.56$ | $18.12 \pm 17.44$ |
| DOAS & WRF-Chem | Year | $7.86 \pm 2.84$ | $-7.10 \pm 6.58$ | $-9.49 \pm 3.25$ | $6.70 \pm 3.19$ | $-5.61 \pm 2.73$ | $-2.83 \pm 3.29$ |
| | Summer | $1.12 \pm 4.47$ | $-13.80 \pm 7.76$ | $5.15 \pm 4.81$ | $-18.66 \pm 4.57$ | $3.13 \pm 4.87$ | $-15.96 \pm 5.19$ |
| | Winter | $-17.83 \pm 8.00$ | $10.38 \pm 9.36$ | $-15.19 \pm 8.66$ | $20.44 \pm 11.95$ | $-14.26 \pm 2.70$ | $10.37 \pm 9.01$ |
| BOAS & WRF-Chem | Year | $2.61 \pm 1.87$ | $-5.78 \pm 3.49$ | $-10.77 \pm 2.11$ | $5.22 \pm 4.10$ | $-6.42 \pm 1.75$ | $-3.30 \pm 3.29$ |
| | Summer | $3.18 \pm 3.41$ | $-12.21 \pm 4.81$ | $3.65 \pm 5.53$ | $-17.87 \pm 4.43$ | $0.27 \pm 3.60$ | $-17.34 \pm 4.82$ |
| | Winter | $-13.87 \pm 2.93$ | $6.38 \pm 6.35$ | $-11.89 \pm 3.22$ | $19.38 \pm 11.44$ | $-12.56 \pm 3.46$ | $7.94 \pm 4.45$ |

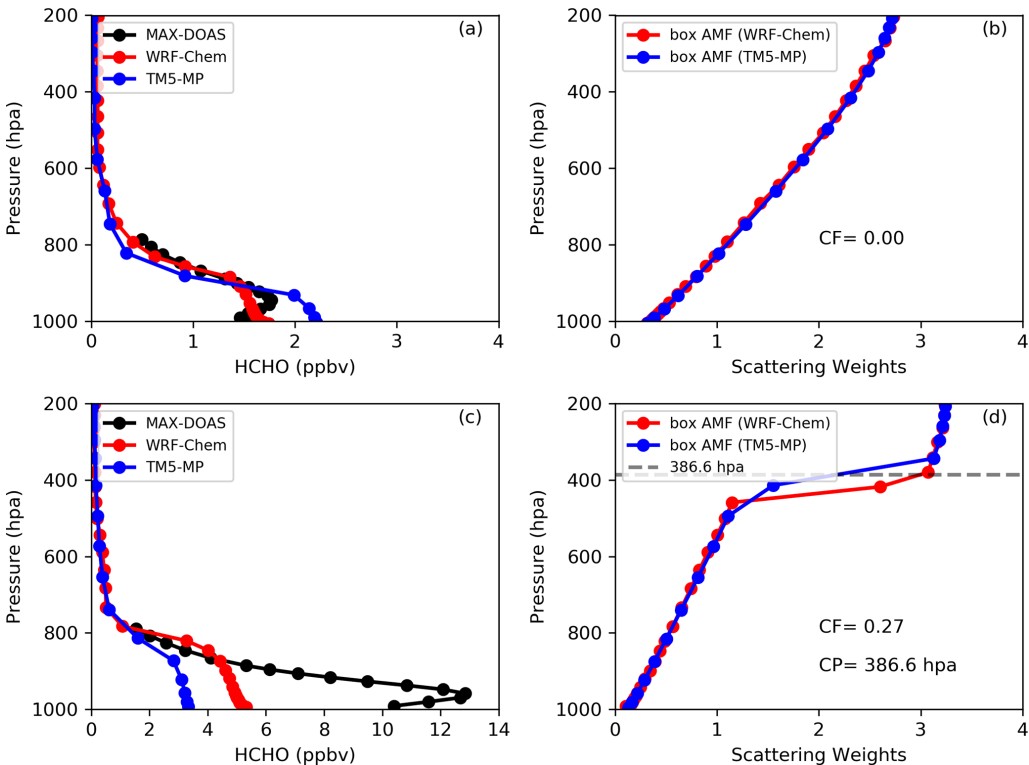

**Figure 6.** Daily averaged vertical HCHO profiles obtained from MAX-DOAS, WRF-Chem and TM5-MP model in the clean case on 3 March 2019 **(a)** and in the polluted case on 26 June 2019. Comparisons of the box AMF using WRF-Chem and TM5-MP simulations in the clean case with clear sky on 3 March 2019 **(b)** and in the polluted case with cloudy sky on 26 June 2019 **(c)**. The locations of the two pixels are within 20 km of the CAMS site.

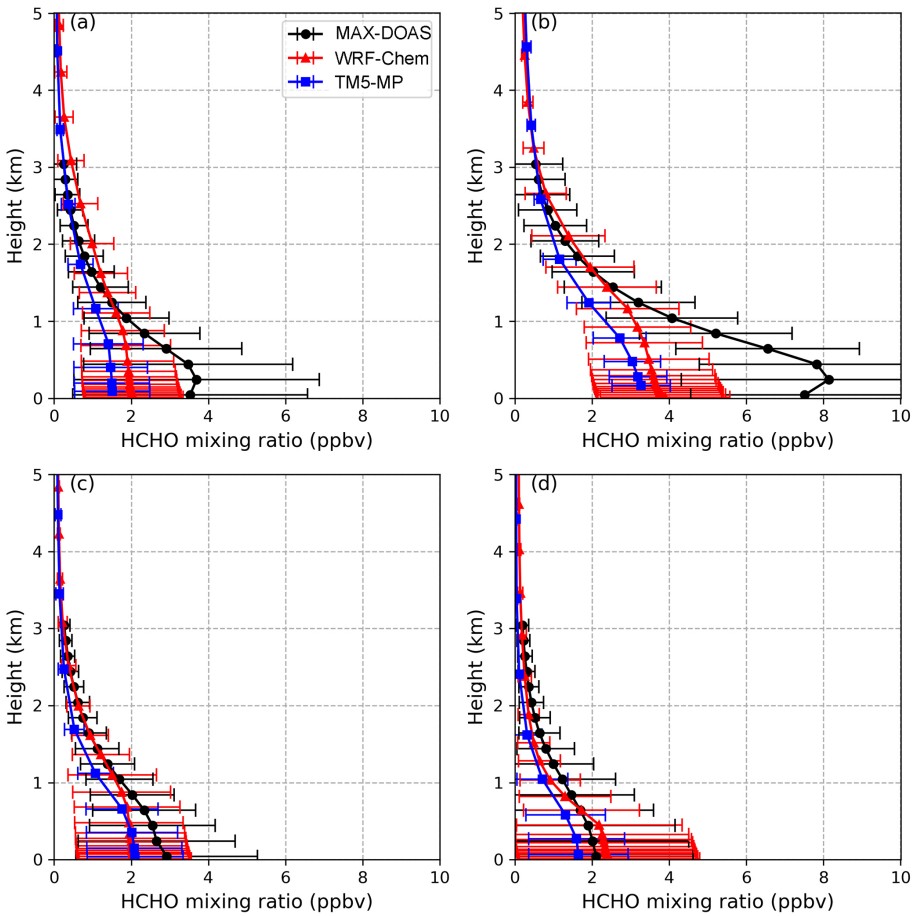

**Figure 7.** Seasonal average of vertical HCHO profiles obtained from MAX-DOAS, WRF-Chem and the TM5-MP model in spring **(a)**, summer **(b)**, autumn **(c)** and winter **(d)** at the CAMS site. The error bars represent $1\sigma$ standard deviation of variation.

sites in Beijing. Our retrieval shows better agreement with MAX-DOAS tropospheric HCHO VCD compared to the operational product. The Pearson correlation coefficients ($R$) of our retrieval are 0.83, 0.84 and 0.83 at CAMS, UCAS and NC sites, respectively. The corresponding R values of the operational product are 0.82, 0.80 and 0.82, which are slightly lower than our retrieval. In addition, the slopes of linear least squares regression lines of our retrieval are 0.77, 0.72 and 0.70 at CAMS, UCAS and NC sites. The corresponding values for the operational retrieval are 0.66, 0.67, and 0.62. In addition, the offsets of linear least squares regression lines of our retrieval are 0.37, 0.20 and $0.37 \times 10^{16}$ molec cm$^{-2}$ at three MAX-DOAS sites, with the corresponding values for the operational product of 0.41, 0.31 and $0.47 \times 10^{16}$ molec cm$^{-2}$. The agreement between our retrieval and MAX-DOAS observations (higher Pearson correlation coefficient and the regression line is closer to 1 : 1 reference line) is in general better than the operational product, suggesting that our retrieval is better than the operational product both in urban and suburban areas in China.

In order to investigate the influence of the HCHO a priori profile in the satellite retrieval, we used the MAX-DOAS HCHO profiles as a priori information in our TROPOMI AMF calculation following Sect. 3.2. The results show an improvement of correlation and absolute agreement. The Pearson correlation coefficient ($R$) increases by about 0.03. The linear fitting slopes increase by 0.14, and the offsets decrease by $0.09 \times 10^{16}$ molec cm$^{-2}$. We have recalculated HCHO VCD for the operational product by using MAX-DOAS HCHO profiles as a priori profiles. The Pearson correlation coefficient ($R$) between the recalculated operational product and MAX-DOAS HCHO VCD decreases by 0.02 to 0.79. The slope of the regression line increases by 0.19 to 0.84, with the offset reduced by $0.24 \times 10^{16}$ to $0.15 \times 10^{16}$ molec cm$^{-2}$. Although the recalculated TROPOMI operational product shows improved correlation with MAX-DOAS observations, our retrieval using MAX-DOAS measurements as a priori profile information still shows better agreement with MAX-DOAS HCHO VCDs. It also indicates that the BOAS spectral retrieval of

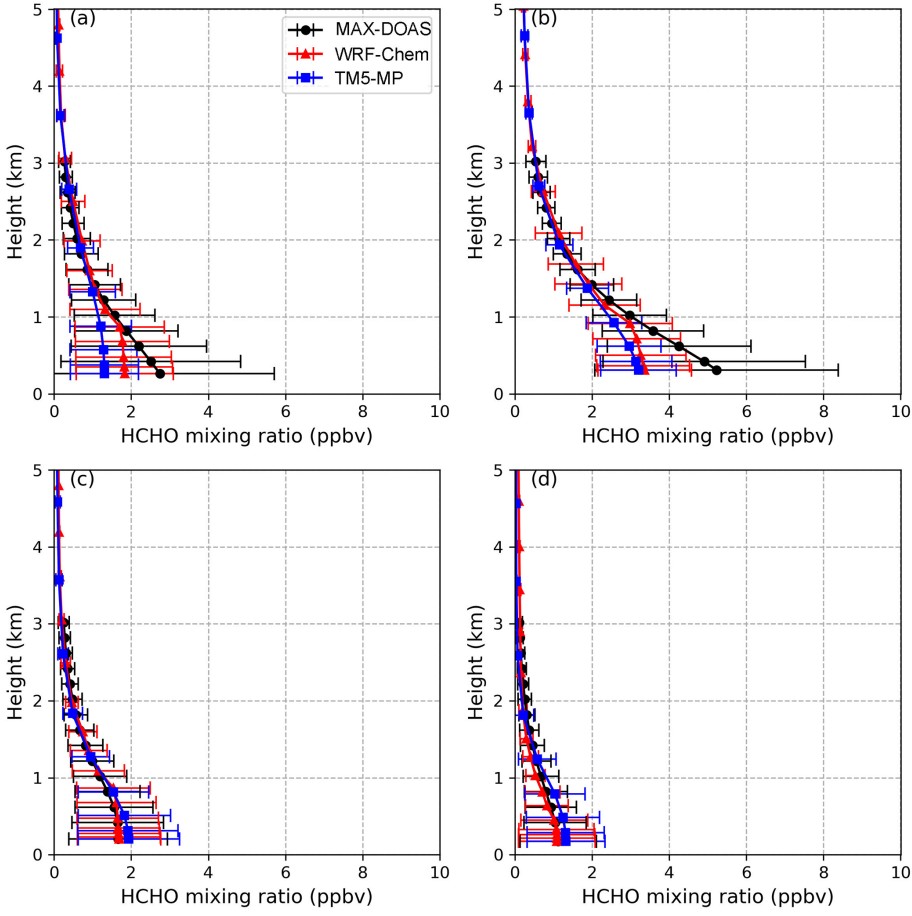

**Figure 8.** Seasonal average of vertical HCHO profiles obtained from MAX-DOAS, WRF-Chem and the TM5-MP model in spring (**a**), summer (**b**), autumn (**c**) and winter (**d**) at the UCAS site. The error bars represent $1\sigma$ standard deviation of variation.

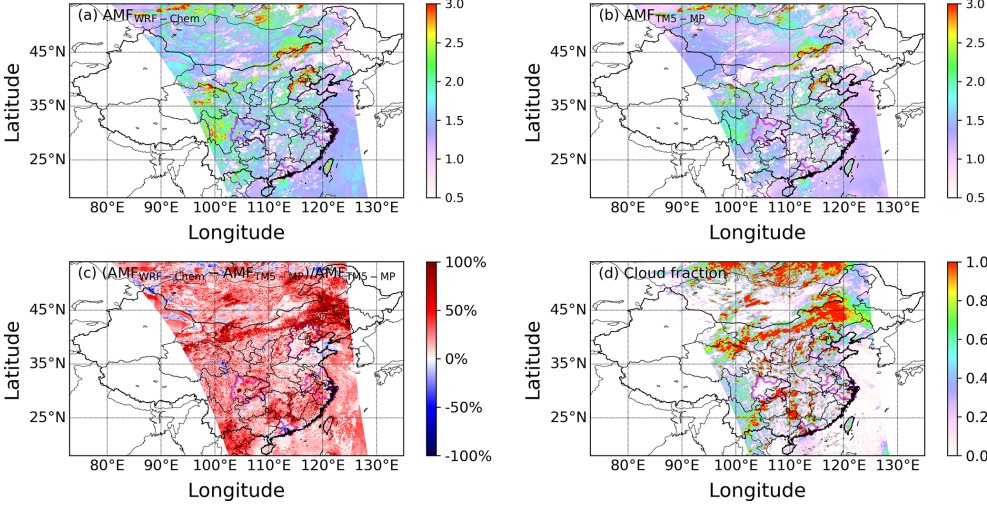

**Figure 9.** Spatial distributions of AMFs calculated using WRF-Chem (**a**) and TM5-MP (**b**) simulations as a priori HCHO profiles of orbit 04211 on 6 August 2018. Spatial distribution of difference of the AMFs (**c**) and of cloud fraction (**d**) of orbit 04211 on 6 August 2018.

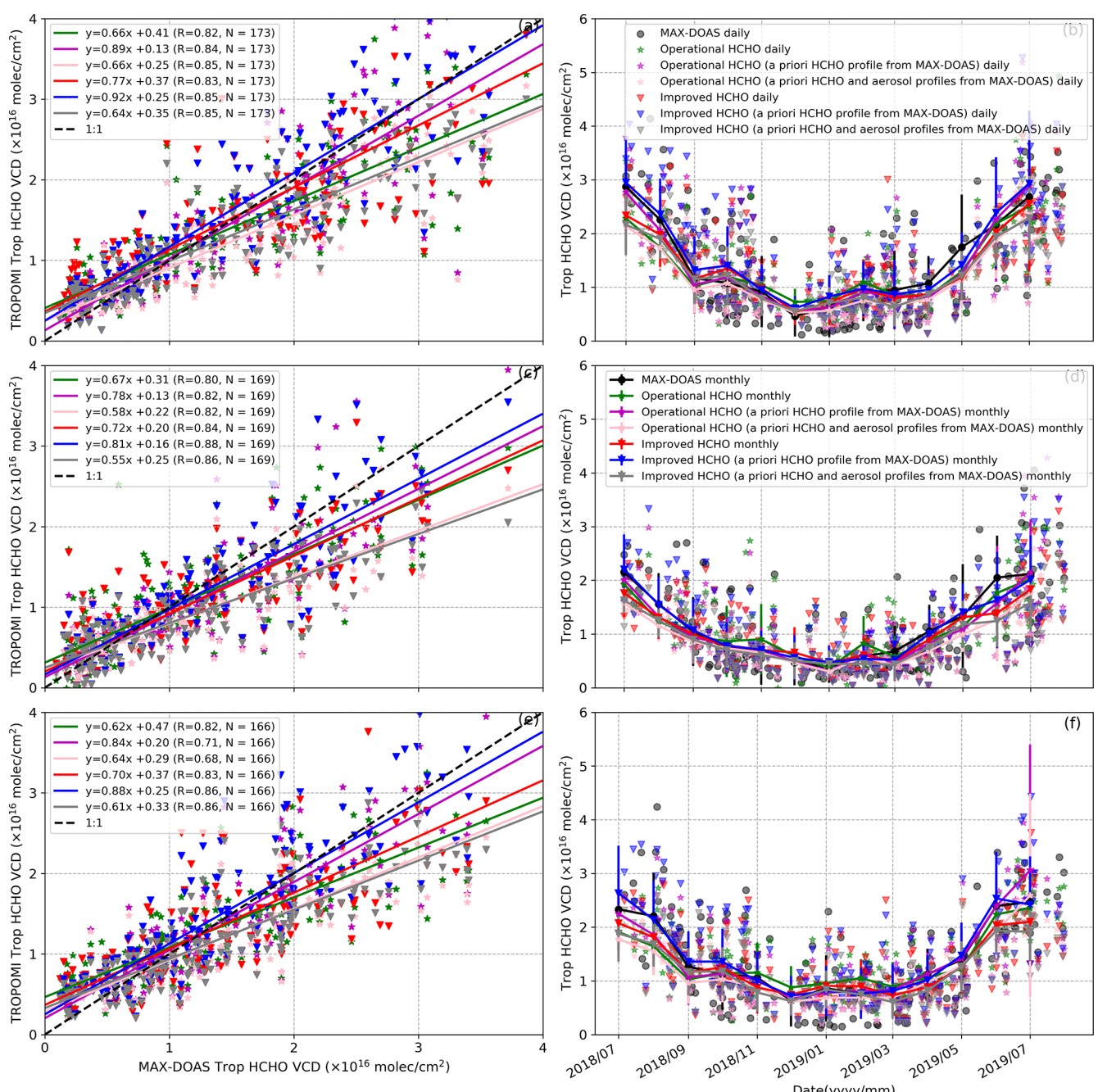

**Figure 10.** Comparison of tropospheric HCHO VCDs between TROPOMI and MAX-DOAS measurements from July 2018 to July 2019 at CAMS **(a)**, UCAS **(c)** and NC **(e)** sites. Time series of TROPOMI and MAX-DOAS tropospheric HCHO VCDs from July 2018 to July 2019 at CAMS **(b)**, UCAS **(d)** and NC **(f)** sites.

SCDs agree better with the ground-based observations over China.

Figure 10b, d and f show the time series of HCHO VCDs from MAX-DOAS, our retrieval and the operational product. HCHO VCDs from our retrieval, the operational product and MAX-DOAS show a similar temporal variation pattern, reaching a maximum value in summer and a

minimum value in winter. At CAMS site, HCHO VCDs are underestimated by TROPOMI observations, and the underestimation of our retrieval ($5.78 \pm 3.49\%$) is slightly smaller ($29.77 \pm 22.83\%$) than that of the operational product ($8.23 \pm 3.09\%$). At the UCAS site, both our retrieval and the operational product overestimate HCHO VCDs, and the overestimation of our retrieval ($5.22 \pm 3.49\%$) is

smaller ($51.44 \pm 33.70\%$) than that of the operational product ($10.75 \pm 3.09\%$). The overestimation of the operational product at the UCAS site is opposite to a previous study which showed that the TROPOMI operational product underestimated HCHO VCDs in Xianghe located $\sim 50\,km$ southeast of Beijing compared to FTIR measurements (Vigouroux et al., 2020). In order to investigate the reason for the overestimation, we separated the data by seasons for comparison at three MAX-DOAS sites (Table 5). In summer, high HCHO columns are due to the oxidation of VOCs related to enhanced biogenic emissions. Both our retrieval and the operational product underestimate HCHO VCD in summer, which is consistent with previous studies (Vigouroux et al., 2020; Chan et al., 2020). As shown in Figs. 7b and 8b, WRF-Chem and the TM5-MP model both underestimate HCHO concentration in the lower troposphere in summer and result in an overestimation of AMFs. The vertical HCHO profiles simulated by the WRF-Chem model better match the one measured by MAX-DOAS in summer compared to TM5-MP. Therefore, our retrieval shows slightly better agreement with the ground-based measurements (underestimation of $15.81 \pm 2.71\%$) compared to the operational product (underestimation of $18.33 \pm 3.10\%$) during summer. We have also calculated the AMFs using the a priori HCHO profile measured by MAX-DOAS for comparison. The underestimation of our retrieval reduces to $1.39\%$ ($\pm 2.67\%$), and the underestimation of the operational product reduces to $9.74\%$ ($\pm 2.78\%$) by using MAX-DOAS profiles for AMF calculations. In winter, lower HCHO columns ($< 1 \times 10^{16}\,molec\,cm^{-2}$) are mainly related to anthropogenic emissions, including vehicle exhaust and industrial emissions. The vertical HCHO profiles simulated by WRF-Chem model are similar to the one measured by MAX-DOAS in winter, while the TM5-MP profiles show larger difference to the MAX-DOAS measurements. Both our retrieval and operational product overestimate HCHO VCDs in winter. The overestimation of our retrieval ($11.23 \pm 4.61\%$) is $63.18\%$ ($\pm 22.63\%$) smaller than the operational product ($29.08 \pm 5.59\%$), and the overestimation in urban area (QKY) is smaller than that in suburban areas (UCAS and NC sites). The improvements of our retrieval in wintertime are more significant, and the improvement at urban sites is also more significant than at suburban sites in winter, which are mainly related to the better anthropogenic emission inventory and the finer spatial resolution over China in WRF-Chem simulations. In order to eliminate effects from the a priori HCHO profile, a priori HCHO profiles from MAX-DOAS measurements are used for AMF calculations for comparison. The overestimations at CAMS and NC sites become less significant. The overestimation at the UCAS site of our retrieval reduces to $3.94\%$ ($\pm 6.87\%$), while the overestimation of the operational product reduces to $10.60\%$ ($\pm 8.71\%$). Our result shows an overestimation of TROPOMI HCHO VCDs during winter at the UCAS site. Our result is different from the previous FTIR comparison study (Vigouroux et al., 2020) which shows that TROPOMI underestimated HCHO columns in winter. On the one hand, the polluted conditions at three MAX-DOAS sites are different from the FTIR sites used in Vigouroux et al. (2020). On the other hand, HCHO concentrations in the lower troposphere are lower in winter, resulting in a relatively larger portion of HCHO above the MAX-DOAS retrieval height of $3\,km$. Therefore, the MAX-DOAS measurements show an underestimation in winter. Our findings are consistent with the previous study that showed that SCIAMACHY HCHO VCDs are in general lower than FTIR measurements while higher than MAX-DOAS observations (Vigouroux et al., 2009). The remaining overestimation is mainly related to a portion of HCHO above $3\,km$ where the MAX-DOAS is not sensitive. In addition, the TROPOMI retrieval assumes an aerosol-free atmosphere, which might also lead to the overestimations.

## 5.3 Aerosol effect on TROPOMI HCHO retrieval

Aerosol extinction profiles are also retrieved by MAX-DOAS. Aerosol optical properties, such as single-scattering albedo (SSA) and Ångström exponent obtained from the Aerosol Robotic Network (AERONET) station in Beijing (https://aeronet.gsfc.nasa.gov/, last access: 22 May 2019,) are used as input parameters for the MAX-DOAS aerosol profile retrieval. As the aerosol profiles from MAX-DOAS are retrieved at $360\,nm$, we further converted the profile to $340\,nm$ using the Ångström exponent obtained from the AERONET measurements. To estimate the aerosol effect on TROPOMI HCHO retrieval, we calculate the AMFs using MAX-DOAS-measured aerosol extinction profiles using VLIDORT (version 2.6; Spurr, 2008). The AMFs are applied to the operational product and our retrieval. The TROPOMI HCHO VCDs with and without considering aerosols are compared to MAX-DOAS HCHO VCDs. The comparison results are shown in Fig. 10. The results show that considering aerosol in the AMF calculations does not improve the agreement with ground-based measurements. Considering the aerosol effect in TROPOMI retrieval reduces HCHO VCDs by $11.46\%$ ($\pm 1.48\%$) for the operational product and $17.61\%$ ($\pm 1.92\%$) for our retrieval in winter. The reduction over urban sites is more significant than suburban sites, mainly due to a higher aerosol load. The operational product using both HCHO and aerosol extinction profiles from MAX-DOAS shows an underestimation of $8.36\%$ ($\pm 4.63\%$). Our retrieval using MAX-DOAS HCHO and aerosol extinction profiles for AMF calculation underestimates HCHO VCD by $18.53\%$ ($\pm 4.04\%$).

## 5.4 Spatial–temporal characteristics of HCHO over China

Figure 11 shows the spatial distribution of HCHO VCDs over China in spring, summer, autumn and winter. HCHO

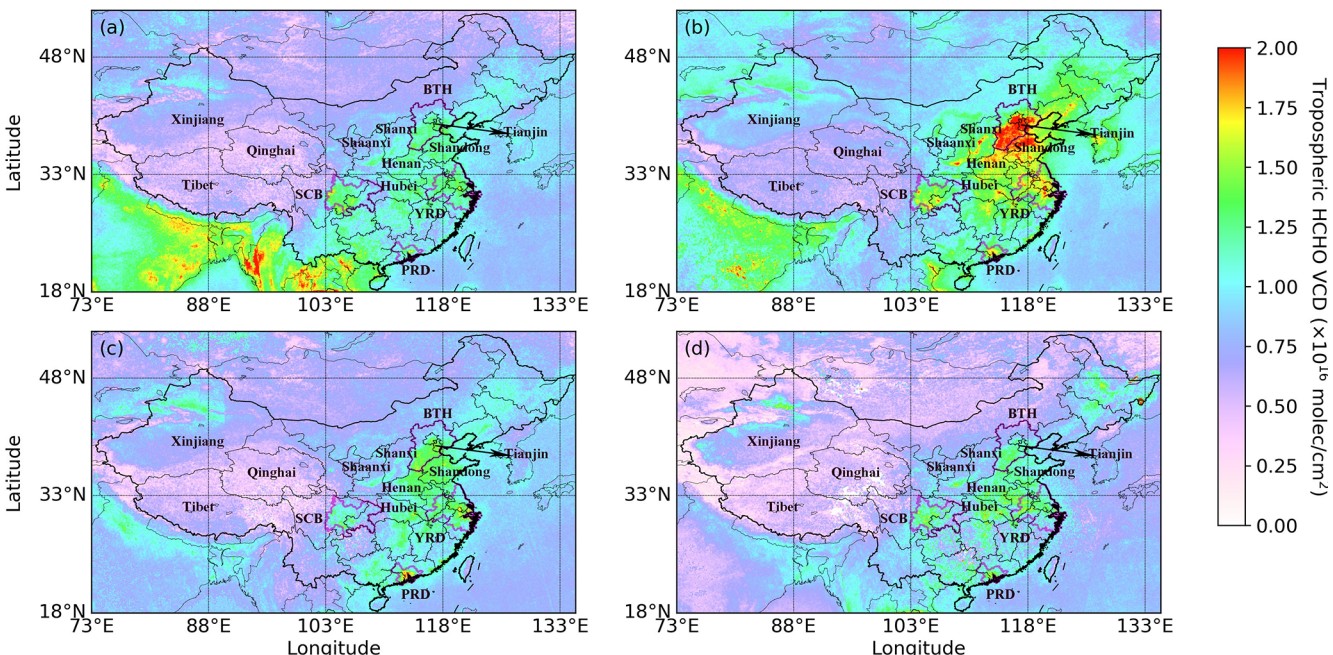

**Figure 11.** The spatial distribution of average tropospheric HCHO VCDs over China on a $0.1° \times 0.1°$ grid in spring **(a)**, summer **(b)**, autumn **(c)** and winter **(d)**.

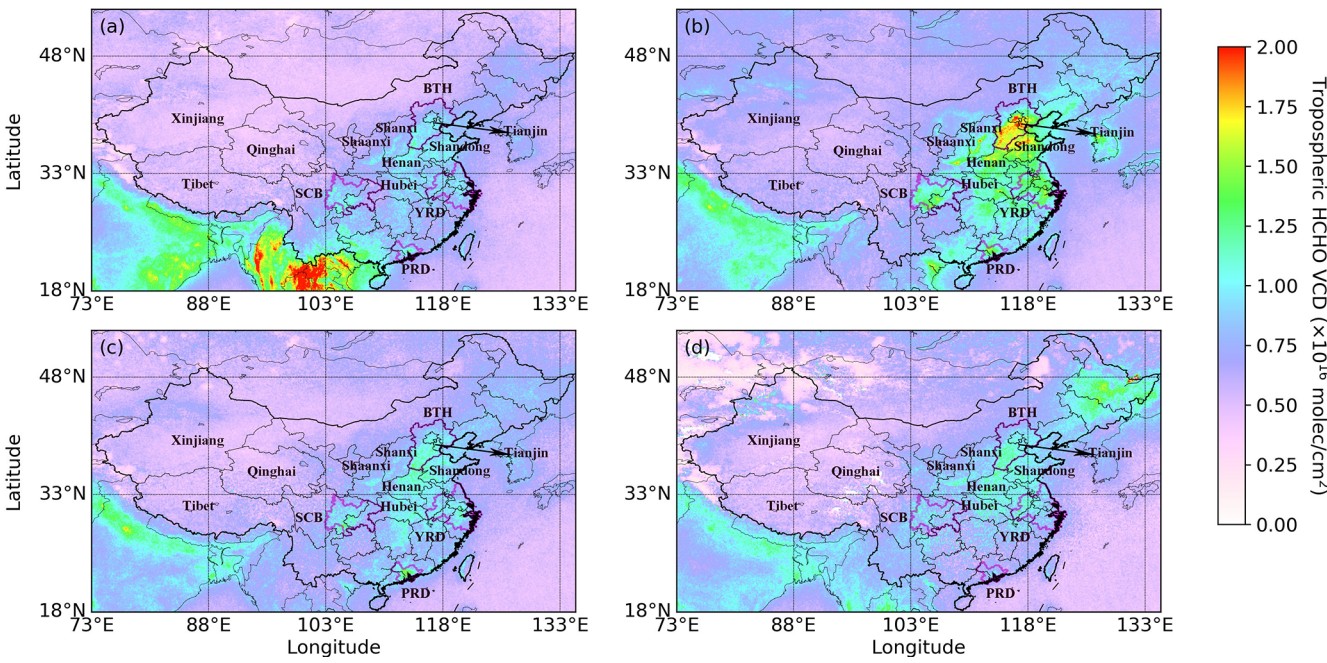

**Figure 12.** The spatial distribution of average TROPOMI operational tropospheric HCHO VCDs over China on a $0.1° \times 0.1°$ grid in spring **(a)**, summer **(b)**, autumn **(c)** and winter **(d)**.

VCDs reach their highest values in summer and lowest values in winter over China. This is due to higher biogenic emissions (Guenther et al., 1995; Li and Xie, 2014; Jiang et al., 2019) and secondary formation in summer (Wang et al., 2016; Su et al., 2019). The average HCHO VCDs over China in autumn 2018, winter 2018, spring 2019 and summer 2019 are 0.93, 0.91, 0.98 and $1.21 \times 10^{16}$ molec cm$^{-2}$, respectively (Table S3). High HCHO VCDs mainly occurred in central and eastern China and SCB. Lower HCHO values are observed in western China. The operational prod-

uct shows similar spatiotemporal distribution to our retrieval over China, while the absolute values are (slight) smaller than our retrieval (Fig. 12). In summer, hotspots can be observed over BTH, YRD, PRD, Shandong province, Henan province, Wuhan (Hubei's provincial capital), SCB and cities along the fen-nutrient-laden valley in Shaanxi and Shanxi provinces in our retrieval. These hotspot patterns are strongly correlated with the population density and industrial emission pattern, indicating a significant anthropogenic contribution. These hotspots are less obvious in the operational product in summer. The distribution of high HCHO VCDs observed in our retrieval over the Xinjiang Uygur Autonomous Region is related to the unique topography and industrial areas. The total industrial VOC emission over the Xinjiang Uygur Autonomous Region is higher than Shanxi province (Zheng et al., 2017).

## 6   Conclusions

In this paper, we present an improved TROPOMI HCHO retrieval over China. We mainly improve the retrieval in two aspects, SCD and AMF calculations. The improved HCHO uses the BOAS technique for the retrieval of HCHO DSCDs. Compared to the DOAS retrieval technique used in the operational product, results retrieved with the BOAS technique are less noisy, while the spatial pattern of HCHO remains similar. DSCDs retrieved with both BOAS and DOAS correlate well with each other, with a correlation coefficient of 0.83 and a linear fitting slope of 0.93. The AMF calculations are improved by using HCHO profiles from the regional chemistry transport model WRF-Chem. The WRF-Chem simulation is featured with higher horizontal and vertical resolution. In addition, the emission inventories used in the model are more up to date. A priori HCHO profiles from WRF-Chem in general agree much better with the MAX-DOAS measurements compared to the TM5-MP simulation. The improvement related to the use of a finer resolution a priori profile is more significant at urban sites, as it better resolved the spatial gradient over these areas.

We also compared the improved and operational TROPOMI HCHO VCDs to MAX-DOAS observations at three MAX-DOAS sites in Beijing. The improved HCHO shows better correlation with MAX-DOAS HCHO VCDs compared to the operational product. The improvement of our HCHO retrieval is significant in wintertime, mainly related to the use of a better a priori profile. Using different spectral retrieval techniques for the retrieval of HCHO slant columns only shows a less significant effect in winter. The bias between the improved TROPOMI HCHO VCDs and MAX-DOAS observations is 63.18 % (± 22.63 %) smaller than the operational product during winter. Moreover, the improvement at urban sites in winter is more significant than at suburban sites. These results suggest our retrieval can better capture the fine-scale variation of HCHO in both urban and suburban regions. The improved TROPOMI HCHO VCDs and MAX-DOAS show a similar temporal variation pattern, with higher values in summer and lower values in winter, while the operational product shows a less pronounced seasonal pattern. The improved and operational products both underestimate HCHO VCDs in summer, while they overestimate HCHO VCDs in winter. Using MAX-DOAS measurements as a priori HCHO profiles in the TROPOMI VCD retrieval improves the correlation and reduces the bias between TROPOMI and MAX-DOAS HCHO VCDs. Considering the aerosol effect in AMF calculation reduces the operational product by 11.46% (± 1.48%) and our retrieval by 17.61% (± 1.92%) in winter.

The analysis of HCHO spatial distribution shows higher values over SCB, central and eastern China and SCB. Lower HCHO VCDs can be observed over western China. In summer, hotspots can be observed over BTH, YRD, PRD, Shandong province, Henan province, Wuhan (Hubei's provincial capital), SCB and cities along the fen-nutrient-laden valley in Shaanxi and Shanxi provinces. These spatial patterns indicate significant anthropogenic contributions of atmospheric HCHO. These hotspots are less obvious, indicating that the improved TROPOMI data set provides better support for the monitoring and controlling of VOC-related pollution.

*Data availability.* The data used in this paper are available on request from the corresponding author (chliu81@ustc.edu.cn).

*Supplement.* The supplement related to this article is available online at: https://doi.org/10.5194/amt-13-1-2020-supplement.

*Author contributions.* WS and CL contributed equally. CL and KLC designed and supervised the study, and CL also helped retrieve the satellite data. WS prepared and analyzed satellite data and wrote the manuscript. QH provided useful comments regarding the results. HL and XJ contributed to providing MAX-DOAS HCHO measurements. YZ and TL helped run the WRF-Chem model. CZ and YC helped prepare satellite retrieval. JL supported the project.

*Competing interests.* The authors declare that they have no conflict of interest.

*Special issue statement.* This article is part of the special issue "TROPOMI on Sentinel-5 Precursor: first year in operation (AMT/ACP inter-journal SI)". It is not associated with a conference.

*Acknowledgements.* The authors acknowledge the Copernicus Open Access Hub and the European Space Agency for making the

Sentinel-5P Level 1 and Level 2 products available online (https://scihub.copernicus.eu/).

*Financial support.* This research has been supported by the National Key Research and Development Program of China (grant nos. 2018YFC0213104, 2017YFC0210002 and 2016YFC0203302), the Anhui Science and Technology Major Project (grant no. 18030801111), the National Natural Science Foundation of China (grant nos. 41722501, 51778596 and 41977184), the Strategic Priority Research Program of the Chinese Academy of Sciences (grant no. XDA23020301), the National Key Project for Causes and Control of Heavy Air Pollution (grant nos. DQGG0102 and DQGG0205), the National High-Resolution Earth Observation Project of China (grant no. 05-Y20A16-9001-15/17-2), the Natural Science Foundation of Guangzhou Province (grant no. 2016A030310115), the Civil Aerospace Technology Advance Research Project (grant no. Y7K00100KJ), and the China Postdoctoral Science Foundation (grant no. 2020TQ0320).

*Review statement.* This paper was edited by Ben Veihelmann and reviewed by three anonymous referees.

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
