# Peer review of "An improved TROPOMI tropospheric HCHO retrieval over China"

_Atmospheric Measurement Techniques, 2020_

## Referee Comment (RC1) · Anonymous Referee #1 · 19 May 2020

**Review of "An improved TROPOMI tropospheric HCHO retrieval over Chine" submitted by Wenjing Su et al., to AMT**

**General comments**

This paper presents TROPOMI formaldehyde retrievals over China. The main difference with respect the operational HCHO retrievals distributed by ESA is the use of specific a-priori information during the air mass factor (AMF) calculation to much better the current knowledge of HCHO distributions over China.

Overall, the results presented here are interesting but the paper lacks details essential to understand the value and implications of the new retrieval. The specific comments section suggests may aspects where the manuscript could improve by including further details. Definitely providing information about TROPOMI level 1 radiances and irradiances is absolutely necessary since they are the base for the retrievals described in section 3. Likewise, for the S5P cloud operational product whose uncertainties will become of paramount importance while assessing the uncertainties of the HCHO retrieval presented here.

The description of the retrieval process also lacks many details. To start with, Figure 1 is included but never referenced in the text. Figure 2 shows two HCHO signals superimposed with the spectral fit residual. It will be more informative to include a similar figure showing the contributions from each one of the parameters included in Table 2. It is very difficult to estimate the quality of this new retrieval without seen the results for one full orbit at least (where issues such as stripes and noise are easier to appreciate). While current figure 3 does this in part, the color scale employed masks may of the possible issues (leaving out negative values) and providing little sensitivity to the region between 0.5 and 1.5 x $10^{16}$ molecules cm$^{-2}$ most relevant for HCHO retrievals. It will also be very helpful to have plots showing the WRF-Chem simulations used to extract a priories as well as the GEOS-Chem simulation used for background corrections. Finally and of outmost importance, a discussion of the retrieval uncertainties is completely missing. While section 4 provides some bias estimates, the retrieval section should include a description of the random and systematic uncertainties linked to spectral fit and AMF calculation.

Because there is no description of the operational HCHO TROPOMI retrieval in detail, it is impossible to assess the weight and validity of the conclusions derived from the comparison exercise between both products (the one presented here and the operational one). The authors focus on the impact due to the ΔSCD retrieval and the AMF calculation but do not devout any time evaluating the impact of using different earthshine radiances. It is also necessary to include a discussion of the MAX-DOAS errors. Figure 5 shows some vertical profiles. What are they? Seasonal averages, daily averages,… Are they representative of similar periods of time and time of day? This results should be place into context with literature recently published (https://www.atmos-meas-tech-discuss.net/amt-2020-30/). Also keep in mind the discussion about QA values in that paper. Given the statistical uncertainties and the methodology used, an affirmation such as "These results suggest that our retrieval is better than the operational product both in urban and suburban in China" is probably over confident. First the MAX-DOAS measurements are representative of a small region in China dominated by large urban development. Second, small differences in the SCDs and the use of a higher spatial resolution model in the new retrieval indicates that both perform similarly. If the authors could provide some details about

the appearance and differences of box-AMFs between both retrievals it will be possible to get a better idea of the impact of the a priori profiles.

The organization of the paper is clear (despite missing important aspects). However, while the use of English in the paper is good enough to be understandable it could benefit from further proofreading and grammar correction. Some language specific suggestions provided in the technical corrections section represent few of the possible improvements. While this reviewer will love to provide detailed language corrections it is time consuming and out of the scope of the reviewer duties.

For the reasons mentioned above the paper needs several major revisions and needs to expand to include essential details. The results shown are promising but in its current form, it will be difficult for anyone outside the authors to use the presented retrievals for a scientific study.

**Specific comments**

**Abstract**

A reference to the paper describing the operational TROPOMI HCHO retrieval by De Smedt et al., should be added here.

Recently Vigouroux et al., have published an excellent operational TROPOMI HCHO retrieval validation paper (https://www.atmos-meas-tech-discuss.net/amt-2020-30/). To cite in the introduction will provide a nice platform to place the results from this study into context later on.

**Data sets**

Overall, the description of the datasets lacks details that will help the reader to understand better the results of this paper. The authors should expend sometime increasing the description of the datasets employed to ensure that their results can be reproduced by other groups. The retrieval of slant column densities requires the use of calibrated TROPOMI radiances. This is maybe, the most important dataset that necessary to obtain the results presented in this work. Its description and how the authors obtained it should be a section in data sets.

These are some of the questions that would be nice to have described in the paper:

1. The original spatial resolution of TROPOPMI observations at NADIR was 3.5 km x 7 km but it improved to 3.5 km x 5.5 km in August 2019.
2. Fitting parameters considered in the spectral fitting of the operational product?
3. How is the reference sector correction applied in the operational product?
4. Procedures used to generate earthshine radiance over the remote Pacific in the operational HCHO retrieval.
5. Besides a-priori profiles and observation geometry, information about clouds, aerosols and surface reflectance play important roles in AMFs calculations. How is the operational product accounting for them?
6. Does the model simulation include information about pyrogenic sources?
7. What was the spin off period of the simulations?
8. It will be interesting to have a reference for the MEIC inventory.

9. What are the uncertainties associated with TROPOMI HCHO retrievals and MAX-DOAS observations. Table 1 could be expanded with information about the estimated uncertainties for each site as well as the dates and amount of available data.

**HCHO SCD retrieval: wavelength calibration**

One has to assume that S represents the preflight instrument slit function. It will be very beneficial to discuss the behavior of the preflight slit function. Are they available to the public? How stable the instrument slit function has been after launch?

**HCHO SCD retrieval: Radiance fitting**

As mentioned above, there is lack of detail in the description of the methodology employed. Please explain the following questions: (1) Methodology employed to calculate the daily average earthshine radiance over the Pacific. (2) Any $I_0$ corrected cross sections in the spectral fit. (3) What is the impact of not including $SO_2$ when if $SO_2$ optical thickness becomes relevant. (4) What is the impact of not including water vapor? (5) Which method is employed to estimate Raman spectra? (5) How was the fitting window selected? HCHO retrievals show big dependencies with fitting windows. (6) New $O_3$ cross sections have become available in recent years (for example Serdyuchenko et al., 2014 (https://www.atmos-meas-tech.net/7/609/2014/)). (7) Have the authors taken into account the effect of ozone in the fitted slant columns as described by Pukite et al., 2008 (https://www.atmos-meas-tech.net/3/631/2010/amt-3-631-2010.pdf).

**AMF calculation**

Details missing in the AMF calculation include: (1) Terrain height and surface pressure corrections. (2) Origin of cloud information. (3) Descriptions of the nodes of the box AMF look up table. (4) VLIDORT set up. (4) Impact of aerosols. (5) Error analysis.

**Reference sector correction of SCDs**

Add description of the GEOS-Chem configuration employed in the reference sector correction. Which longitudes define remote Pacific? Is the correction applied only to -30 degrees to 30 degrees? That region is used to calculate the earthshine radiance. Any contributions outside -30 to 30 degrees will not be correcting for the residual HCHO column but for biases of different origin.

**Comparison of operational and improved HCHO product**

What are the coincidence criteria to match TROPOMI and MAX-DOAS measurements? To use daily averages in the case of MAX-DOAS seems inappropriate considering the diurnal variations of HCHO columns.

**Comparisons of SCD retrievals**

This discussion applies to SCDs or ΔSCDs? The result of the spectral fit is in both cases the differential slant column but the title of the section indicates the comparison of slant columns. In that case, differences are not only due to the spectral fit but also to the reference sector correction. What is the impact of using different earthshine radiances?

What are the reasons to filter out retrievals with RMS higher than $10^{-3}$. What is the percentage of retrievals with RMS above the threshold? What is the definition of outliers? Showing a histogram of the distributions of both datasets will help to understand the outlier definition.

The authors make a distinction between non-corrected and corrected SCDs. It is not clear in the text what corrected implies. One has to assume we are talking about reference sector corrected and non-corrected SCDs. If that is the case, one could argue that part of the differences observed between SCDs in the non-corrected case are due to the different selection (or calculation) of earthshine radiance reference. To make a real comparison of the performance of booth fitting algorithms they should be using consistent earthshine radiances. The results seem to indicate that part of the biases are due to using different earthshine radiances since the correlation for the non-corrected and corrected SCDs columns is similar for both products but the bias between them is significantly reduced after applying the correction.

**Comparisons of retrievals after AMF calculations**

One question that raises this section and always permeates the use of high-resolution models is how much information is folded back from the model in the retrieval. To better understand this question it will be very useful to know more about the WRF-Chem simulations and how they were used. The operational product employs daily forecast interpolated in time and space. This procedure should be added to the methodology. How do the box-AMFs of the different retrievals compare?

**Comparison between HCHO VCDs observed by MAX-DOAS and TROPOMI**

Blue lines in figure 7 right panels, more than speaking of the retrieval, speak about the difference between the a priori profiles for both models and the retrieved MAX-DOAS profiles. Slopes and correlation coefficients are similar (within error estimates) for both retrievals when using model a priori. It is easier to imagine how the comparison of the operational retrieval with MAX-DOAS observations will also suffer a dramatic improvement if MAX-DOAS a priori were to be used in the AMF calculations.

**Technical corrections**

Line 18 (grammatical suggestion (GS)): Since what is improved is the HCHO retrieval a more correct grammatical structure will be "We present an improved retrieval of formaldehyde (HCHO) over China from the TROPO…"

Line 23 (GS): remove "the" from "agreement with the ground based…" since it is possible there are more than MAXDOAS measurement than those used in this study.

Line 25 (GS): add "s" to "profile" "…higher resolution a priori profiles".

Line 26: The percentage of what is reported by 61.11% and 0.15%? With the current text it is impossible to know if it refers to the change in the mean VCD, or the percentage of the bias correction attributed to each step. Please specify.

Line 29 (GS): Change "indicating" to "indicate".

Line 45 (GS): Add "to" after compared "Compared to its predecessor"

Line 53 (GS): Add "s" to "profile" and "a" after "from". Which regional model is employed?

Line 64 (GS): Change "is" to "are" ".., which are devided in …".

Line 91 (GS): Add "the" in "… is located in the Chinese…".

Line 105: The first step of the methodology explained in this paper is the calculation of ΔSCD, not SCD.

Line 111: There can be other causes for wavelength miss-alignment can be Doppler shift, non-uniform illumination of the slit due to presence of clouds or other high reflectance surface in the pixel.

Line 115: What is the benefit of having two closure polynomials during wavelength calibration considering that it is done using TOA irradiances and there for are not affected by the presence of clouds or aerosols that may introduce low-frequency structures?

Line 142 (GS): Add "presence of" to "… atmosphere (presence of clouds, vertical HCHO distribution…"

Line 143: What is a comprehensive radiative transfer model?

Line 150 (GS): Add "on" as "The box AMF depends on wavelength, …"

Line 173 (GS): Add "be" to "… is known to be caused by …"

Line 173: Could the authors provide a reference for the known cause of stripping?

Line 175: AMFs are not retrieved; they are calculated and are independent of a remote Pacific earthshine radiance.

Line 204 & 209 (GS): Change "outliners" to "outliers".

Line 216 & 217: What is the error associated to the average SCDs? Standard deviations?

Line 222: How is the 32.32% lower calculated? A difference of $0.02 \times 10^{16}$ molecules cm$^{-2}$ looks rather small.

Line 223-226 (GS): Please rewrite sentence.

Line 228: How does BOES compute mean random errors? Does this calculation use the operation product random error definition? Without describing both is impossible to interpret BOAS random errors lower than DOAS by 22%.

Figure 1: What is figure 1 trying to illustrate. It is not mention in the text.

Table 1: What is the definition of viewing azimuth angle? Probably clock wise with respect to North.

Table 2: What is the methodology employed to calculate the Raman spectra? What is the definition of Pacific (remote)? Will it be possible to provide a reference for the pre-flight instrument slit function?

Figure 2: What the figure shows is the ΔSCD not SCD. Please correct. The size of the residuals in plot (b) seem to be large considering an RMS of $2.78 \times 10^{-4}$.

Figure 3: A different color map, extending to negative values, will provide a better picture of the BOAS retrieval performance since there is a significant number of pixels with negative columns. Which orbits contribute to these plots?

Figure 4: Does plot (a) show ΔSCDs and plot (b) SCDs? Please clarify. Does this regression consider an independent variable? Probably is better to consider the errors of both variables in the linear fit.

Table 3: It is very difficult to interpret. How can be the NMBs between satellite and MAX-DOAS be 0% for the operational product? The caption needs to be re-written to provide a proper description of what the table is showing.

Figure 5: What is the methodology used to calculate the vertical profiles? Time averaging, filtering of MAX-DOAS observations… Co-registration of models and models with in-situ measurements.

Figure 6: Using a divergent color map scale in plot © will help to appreciate the positive and negative differences. What is the effect of clouds and aerosols? Some of the big differences resemble cloud structures. What is the correlation between both AMF calculations?

Figure 8: Do these plots show improve HCHO data? It seems to show gridded data. What is the spatial sampling of the grid? Which methodology was used to calculate the gridded fields.

---

## Referee Comment (RC2) · Anonymous Referee #2 · 27 May 2020

**Review of Su et al., 2020: "An improved TROPOMI tropospheric HCHO retrieval over China".**

**General comments**

The paper aims to present an improved TROPOMI HCHO retrieval over China. Compared to the ESA operational product, two major differences are highlighted: (1) the use of BOAS instead of DOAS for the of fit the slant columns, (2) the use of a priori profiles from a regional model in order to recalculate the AMF, with a finer spatial resolution, and optimized emissions over China.
Overall, the paper fails in showing an improvement of the slant columns, the differences between the two products being negligible.
The scientific interest of the paper lies in the second improvement. The authors should focus more on this aspect, and go further into a detailed analysis of the spatio temporal effects of using more precise profiles for satellite HCHO observations. However, it is not demonstrated how the finer spatial resolution of the model improves the validation. Here it could help to show that the improvement is more important over the urban site compared to the sub-urban sites. Or is it more an effect of the different model chemistry/emissions, and not a spatial resolution effect?
Along the paper, the numbers are often used in a quite subjective way. (0.15% difference being called an improvement for exemple). I recommend writing quantitative comparisons with a more rigorous analysis to strengthen the message of the paper.
The paper needs major revisions before being published in AMT.

**Specific comments**

The title does not fairly reflect the contents of the paper. Unless the results are significantly extended, the title of the paper should focus more on the "improved AMF calculation over China".

The section called "Improved HCHO retrieval algorithm" presents the retrieval algorithm developed for this study. As described in this section, it is actually very similar to the ESA operational product. The similarities and the differences need to be clearly explained. For example, the wavelength calibration. The description is the same as for the operational product. Why a specific section dedicated to this aspect? It would be interesting to see a comparison of the calibration results between the 2 products. The same holds for the AMF calculation part. It is very similar to the Tropomi HCHO ATBD and the differences are not clearly explained, except for the a priori profiles. Same for the reference sector correction.

It is not shown in the paper that the SCDs have been improved. There is a contradiction between the introduction ("BOAS has been reported featured with lower fitting uncertainties to the standard DOAS method") and the result section, where it is stated (page 10) that the RMS are identical. So the "lower fitting uncertainties" of the BOAS technique are not demonstrated. As for Figure 4 with the slant columns, a figure with RMS comparison needs to be added.

In section 4.1.2, it is explained that the operational product has been updated using a priori profile from WRF model. Does it mean that operational averaging kernel have been used? Or did the authors used their own radiative transfer calculation? It is important to know in order to understand if other sources of differences, such as the albedo, can play a role in the observed vcd differences.

Improved AMF for China should include some tests about the aerosol effects. They are not even mentioned.

When comparing to the MAX-DOAS data in Beijing, MAX DOAS profiles have been used to re-calculate the amf of the improved Chinese HCHO product (Figure 7). For a fair comparison, the same method needs to be applied to the operational product. All the needed information are provided in the operational L2 files.

I have some concerns about the way validation results are presented; The operational product, such as most of existing HCHO satellite products, is rather known to be underestimated over emission regions such as Beijing. See for example Jung et al. 2019 (https://doi.org/10.1029/2019EA000702) or Vigouroux et al. 2020 (https://doi.org/10.5194/amt-2020-30) and references therein. Here the authors claim to find an opposite result. The operational product is overestimated, and the improved Chinese product is lower and in better match with the MAX-DOAS. But actually this result holds for winter time only. The results should be discussed more in terms of low column (winter) or high columns (summer). Finally, a link to previous satellite HCHO validation studies should be made, and the reasons for such different conclusions need to be discussed.
The last paragraph of section 4.2 is the most interesting part of the paper and deserves to be extended. It is found that both algorithms remain underestimated in summer time, when the columns are the largest and mainly related to biogenic emissions. Both models simulate profiles not peaked enough near the surface. However, in winter time, when the columns are the lowest (no biogenic emissions), an improvement is observed compared to the MAX-DOAS observations when using WRF-Chem model as a priori profiles. Can you say something about possible reasons for this? Does the WRF-Chem model perform better than TM5 for anthropogenic emissions? Is it related to the spatial resolution or to the chemistry?

The discussion about seasonal variation of the improved Chinese product, and its spatial distribution over China does not bring anything new about current HCHO satellite observations.

I advise to either remove this part, either extend with meaningful observations going much more into details.

Comparison maps of SCD and AMF are shown. It would be good to do the same for the final VCD.

**Technical corrections**

**Abstract**

L18: We present  an improved retrieval…
L19:  The new retrieval optimizes the slant column density retrieval: this is not demonstrated in the paper. Please rephrase.
L24: MAX-DOAS measurements in  Beijing
L26: while the SCD retrieval only shows a minor effect of 0.15%. This is negligible! We can even talk about a perfect agreement between the SCD retrievals.
L29-30: The last sentence is not demonstrated in the paper.

**Introduction:**

L48: Again, It is not shown in the paper that the SCDs have been improved. The "lower fitting uncertainties" of the BOAS technique are not demonstrated.
L54: the result is expected to be more realistic for the investigation of spatio temporal variation of HCHO over China. ok but this needs to be demonstrated. How the spatio temporal variation of HCHO over China has been improved? In the current version, only a reduction of the bias compared to MAX-DOAS data is shown in winter time.

Figure 1: The scale could be reduced to better show emission spots.

**WRF-model**

L79: more up to date emission inventory of China: this is really vague and needs to be explained

**Improved HCHO retrieval algorithm**

L133: in Table  2
Table 2:
- Why the use of DSCD in the caption?
- Please indicate the differences compared to the operational product.
- What about the Ring correction?

- Do you include corrections for non-linear Ozone absorption effects?
- How is the radiance reference sector calculated? Per instrument row? Per day?
- Why this particular choice of O3 and BrO cross-sections? Can these choices explain the differences with the operational product?

L168: the surface albedo is obtained from the S5P operational cloud product. This is a bit surprising. Please specify the wavelength.

L187: specify the meaning of k and m

**Results and discussions**

L196: VT and Vm is the average tropospheric CHO VCD measured by TROPOMI and MAX-DOAS. How are the data averaged in space / time?

Figure 3: Do the maps show SCD, DSCD (as mentioned in Table 2) or corrected SCDs? It would be good to show corrected SCDs (with a color scale including negative values), since an offset is found in the SCDs. It would help to better see differences in the two spatial distributions.

L206: Please compare numbers for the corrected slant columns over Tibet.

L218: Please compare the RMS.

L218-219: This sentence is vague. Please be more specific

L223: Please give the numbers in brackets for the Chinese product as well.

L228. It is not clear how using the BOAS HCHO SCDs reduces the overestimation if changing SCD retrieval method only shows a tiny effect of 0.15%? There is a contradiction here.

L228: The mean random errors relative to BOAS are mentioned. Can you give a definition? And where are those errors presented in the paper?

**AMF calculation**

Table 3: This table is difficult to understand. The presentation of the numbers can be improved. The legend says that NMBs between satellite and MAX-DOAS are provided, but it seems to be more than that (NMB s1, s2). Error bars should be added. It would be more relevant to separated numbers for winter and summer periods.

Figure 5:
- Profiles are shown at the more urban CAMS station. It would be interesting to also show a suburban station, in order to detect the gain in spatial resolution.
- How many profiles are averaged? What is the spatial resolution?

L249: The operational data are filtered using the QA value. Is the same selection applied to the improved Chinese product? If not, which selection is applied?

L255: Validation results are discussed at the 3 sites using correlation, slope and offset. Looking at those 3 parameters, mainly the offset is improved compared to the operational product. Correlations are almost identical. This needs to be discussed more in detail, related to the observed offset in the AMFs.

L263: Please explain how the MAX-DOAS are used to recompute the AMFs. Do you use the averaging kernels? The same needs to be done with the operational product.

L272: The vertical profiles simulated by WRF-Chem are similar to the one measured by MAX-DOAS in  winter !

L273: The underestimation of both retrievals in summer time are similar. 9.96% versus 10.88% is not significant. Please add error bars. Only the differences in winter time are significant.

**Section 4.3**

Not much useful information is given in this short paragraph. I suggest extending with a comparison with maps of VCD from the operational product, for the 4 seasons.

**Conclusion**

As for the abstract and the title, the conclusions need to be redirected towards the real content of the paper, which is the use of a regional model to compute the AMFs, and the validation at 3 sites in Beijing.

---

## Author Comment (AC1) · 26 Jul 2020

Response to reviewer #1

We really appreciate your constructive comments and suggestions on our manuscript. We have considered every comment carefully, and responded on a point to point and marked every change in red in the revised version.

**General comments**

This paper presents TROPOMI formaldehyde retrievals over China. The main difference with respect the operational HCHO retrievals distributed by ESA is the use of specific a-priori information during the air mass factor (AMF) calculation to much better the current knowledge of HCHO distributions over China.

Overall, the results presented here are interesting but the paper lacks details essential to understand the value and implications of the new retrieval. The specific comments section suggests may aspects where the manuscript could improve by including further details.

Definitely providing information about TROPOMI level 1 radiances and irradiances is absolutely necessary since they are the base for the retrievals described in section 3.

**Responses:** Thank you very much for this suggestion. Information about TROPOMI level 1 radiances and irradiances and the S5P cloud operational product are added in Section 2.1. TROPOMI has four spectrometers for medium wave ultraviolet (UV), long wave ultraviolet combined with visual (UVIS), near infrared (NIR), and short wave infrared (SWIR), covering non-overlapping and non-contiguous wavelengths from 270 to 2385 nm which are divided into eight spectral bands. The Band 3 with wavelength from 320 to 405 nm is used for HCHO retrieval. Radiance in Band 3 is measured by UVIS spectrometer. The detector for UVIS spectrometer is a two-dimensional charge-coupled device (CCDs) with one dimension for wavelengths and the other dimension for across track spatial coverage (450 rows). Earth radiance is collected along the dayside of the earth, while solar irradiance measurements are performed near the North Pole every 15 orbits, approximately once a day. In the UVIS channel, the spectral resolution and spectral sampling is about 0.5 nm and 0.2 nm, respectively. Individual ground pixels size of radiance measurement is approximately 3.5 km in the across-track and 7 km (5.5 km since August 2019), with integration time of 1.08 s (0.84 s since August 2019). The Level 1B radiance and solar irradiance are available at the Copernicus Open Access Hub (https://scihub.copernicus.eu/, last access: 22 May 2019).

**Changes in manuscript:** L66-76, P3-4 in the revised version.

Likewise, for the S5P cloud operational product whose uncertainties will become of paramount importance while assessing the uncertainties of the HCHO retrieval presented here.

**Responses:** Thank you very much for this suggestion. Information about the S5P cloud operational product are added in Section 2.3.

**Changes in manuscript:** L100-105, P5 in the revised version.

The description of the retrieval process also lacks many details. To start with, Figure 1 is included but never referenced in the text.

**Responses:** Thank you very much for this suggestion. Figure 1 is cited at L123, P6 in Section 2.5

Figure 2 shows two HCHO signals superimposed with the spectral fit residual. It will be more informative to include a similar figure showing the contributions from each one of the parameters included in Table 2.

**Responses:** Thank you very much for this suggestion. We plotted the spectral fitting residuals with considering different cross sections (shown in Figure. A1) on 6 August 2018 over China (orbit 4211). For HCHO polluted case, considering $O_3$ cross sections at 228K and 295K reduces residual significantly while considering $NO_2$, BrO and $O_4$ cross sections affects residual little (Figure. A1 (a)). Moreover, Ring effect and $O_3$ cross sections affect SCD largely and help reduce random error. For HCHO clean case, considering $NO_2$, BrO and $O_4$ cross sections reduces residual largely (Figure. A1 (b)). Moreover, the uncertainties of selection of fitting window, polynomial and TROPOMI slit function are discussed in Section 4.1.

[Figure]

Figure. A1 Fitting residuals with considering different cross sections on 6 August 2018 over China for (a) polluted case and (b) clean case. The legend "HCHO" represents only considering HCHO cross section in spectral fitting and the legend "HCHO+Ring+O3(228K)+ O3(295K)+NO2+BRO+O4+Talor expansion(O3)" represents considering all cross sections listed in Table 1 and taking into account the first order Taylor series expansion for $O_3$ SCD at two temperatures in spectral fitting.

**Changes in manuscript:** L183-185, P8 in the revised version and L25-45, P3 and Figure S3 in supplement file.

Table A1. The spectral fitting results with considering different cross sections. The unit of DSCD and random error of DSCD is $10^{16}$ molec cm$^{-2}$ and the RMS is expressed in $10^{-4}$. The fitting results which cross sections are considered in our DSCD retrieval are marked in red.

| | Polluted case | | | Clean case | | |
|---|---|---|---|---|---|---|
| | DSCD | Random error of DSCD | RMS | DSCD | Random error of DSCD | RMS |
| HCHO | 4.51 | 0.43 | 6.3 | 0.59 | 0.42 | 6.3 |
| HCHO+Ring | 4.32 | 0.44 | 6.3 | 0.57 | 0.44 | 6.3 |
| HCHO+Ring+O$_3$(228K) | 4.27 | 0.34 | 4.9 | 0.57 | 0.44 | 6.3 |
| HCHO+Ring+O$_3$(228K)+O$_3$(295K) | 3.99 | 0.35 | 4.7 | 0.68 | 0.46 | 6.3 |
| HCHO+Ring+O$_3$(228K)+O$_3$(295K)+NO$_2$ | 3.96 | 0.34 | 4.6 | 0.64 | 0.45 | 6.1 |
| HCHO+Ring+O$_3$(228K)+O$_3$(295K)+BrO | 4.09 | 0.37 | 4.6 | 1.15 | 0.48 | 6.0 |
| HCHO+Ring+O$_3$(228K)+O$_3$(295K)+BrO+O$_4$ | 4.09 | 0.37 | 4.6 | 1.17 | 0.46 | 5.7 |
| HCHO+Ring+O$_3$(228K)+O$_3$(295K)+BrO+O$_4$ +Taylor expansion (O$_3$) | 4.38 | 0.42 | 4.6 | 1.10 | 0.49 | 5.6 |
| HCHO+Ring+O$_3$(228K)+O$_3$(295K)+BrO+O$_4$ +SO$_2$ | 4.40 | 0.40 | 4.5 | 1.73 | 0.51 | 5.6 |
| HCHO+Ring+O$_3$(223K)+O$_3$(243K)+BrO+O$_4$ | 4.12 | 0.37 | 4.6 | 1.19 | 0.46 | 5.8 |

It is very difficult to estimate the quality of this new retrieval without seen the results for one full orbit at least (where issues such as stripes and noise are easier to appreciate). While current figure 3 does this in part, the color scale employed masks may of the possible issues (leaving out negative values) and providing little sensitivity to the region between 0.5 and 1.5 x 1016 molecules cm$^{-2}$ most relevant for HCHO retrievals. It will also be very helpful to have plots showing the WRF-Chem simulations used to extract a priories as well as the GEOS-Chem simulation used for background corrections. Finally and of outmost importance, a discussion of the retrieval uncertainties is completely missing. While section 4 provides some bias estimates, the retrieval section should include a description of the random and systematic uncertainties linked to spectral fit and AMF calculation. Because there is no description of the operational HCHO TROPOMI retrieval in detail, it is impossible to assess the weight and validity of the conclusions derived from the comparison exercise between both products (the one presented here and the operational one). The authors focus on the impact due to the ΔSCD retrieval and the AMF calculation but do not devout any time evaluating the impact of using different earthshine radiances. It is also necessary to include a discussion of the MAX-DOAS errors.

Responses: These comments are also referred in Specific comments. We responded on a point to point in Specific comments.

Figure 5 shows some vertical profiles. What are they? Seasonal averages, daily averages,… Are they representative of similar periods of time and time of day? This results should be place into context with literature recently published (https://www.atmos-meas-tech-discuss.net/amt-2020-30/).

**Responses:** Thank you very much for this suggestion. The Figure 5 in the old version is Figure 7 in the revised version. Figure

7 shows seasonal averages of vertical HCHO profiles from MAX-DOAS, WRF-Chem and TM5-MP model at CAMS site. In AMF calculations, both WRF-Chem and TM5-MP simulations are interpolated to TROPOMI spatial resolution. Interpolated WRF-Chem and TM5-MP simulations within 20 km of the MAX-DOAS site are spatially averaged to compare with MAX-DOAS profiles. MAX-DOAS profiles are temporally averaged in the period 13:30-14:30 (Local Time) within ±1 h around the TROPOMI overpass time. And the error bars in Figure 7 represent 1σ standard deviation of seasonal variation.

**Changes in manuscript:** L366-370, P16 in the revised version.

Given the statistical uncertainties and the methodology used, an affirmation such as "These results suggest that our retrieval is better than the operational product both in urban and suburban in China" is probably over confident. First the MAX-DOAS measurements are representative of a small region in China dominated by large urban development. Second, small differences in the SCDs and the use of a higher spatial resolution model in the new retrieval indicates that both perform similarly. If the authors could provide some details about the appearance and differences of box-AMFs between both retrievals it will be possible to get a better idea of the impact of the a priori profiles.

**Responses:** Thank you very much for this suggestion. Due to limitation of MAX-DOAS data sets, we validate TROPOMI HCHO observations at three MAX-DOAS sites in Beijing including one urban site (CAMS site) and two suburban sites (NC and UCAS sites) (shown in Figure A3). The distances of three MAX-DOAS sites from central Beijing are 8, 27 and 61 km of CAMS, NC, and UCAS sites. The three MAX-DOAS sites are representative of urban and suburban region.

We using a different spectral retrieval technique (BOAS method) of HCHO slant columns. Moreover, the background correction has also been improved. Although DOAS and BOAS HCHO DSCDs show a similar spatial pattern. The spatial distribution of BOAS HCHO SCD is expected to be smooth, less noisy. Besides, operational product using different SCD retrieval methods is compared with MAX-DOAS HCHO measurements. Using the BOAS HCHO SCDs reduced the underestimation in summer and overestimation in winter of the operational product. In summer, using different SCD retrieval methods results a difference of 7.00% (± 1.71%, ± Error) from the TROPOMI operational HCHO VCD. The result shows that using the BOAS HCHO SCDs reduced the underestimation in summer and overestimation in winter of the operational product. Fig. A2 (a) and (c) show daily averaged a priori HCHO profiles from WRF-Chem and TRM5-MP simulations and MAX-DOAS measured profiles. Besides higher horizontal resolution, WRF-Chem simulation also has a higher vertical spatial resolution compared to TM5-MP data set. Although the difference in vertical resolution only shows negligible effect on box AMF under clear sky condition (Fig. A2 (b)), lower vertical resolution profiles would cause significant impact for cloudy cases due to the interpolation of coarse grid (Fig. A2 (d)).

[Figure]

Figure A2. Daily averaged vertical HCHO profiles obtained from MAX-DOAS, WRF-Chem and TM5-MP model in clean case on 03 March 2019 (a) and in polluted case on 26 June 2019. Comparisons of box AMF using WRF-Chem and TM5-MP simulations in clean case with clear sky on 03 March 2019 (b) and in polluted case with cloudy sky on 26 June 2019 (c). The locations of two pixels are within 20 km of the CAMS site.

**Changes in manuscript:** L126-128, P6 and L370-374, P16-17 in the revised version.

[Figure]

Figure A3. Locations of three MAX-DOAS sites in Beijing

The organization of the paper is clear (despite missing important aspects). However, while the use of English in the paper is good enough to be understandable it could benefit from further proofreading and grammar correction. Some language specific suggestions provided in the technical corrections section represent few of the possible improvements. While this reviewer will love to provide detailed language corrections it is time consuming and out of the scope of the reviewer duties.

For the reasons mentioned above the paper needs several major revisions and needs to expand to include essential details. The results shown are promising but in its current form, it will be difficult for anyone outside the authors to use the presented retrievals for a scientific study.

**Specific comments**

**Abstract**

A reference to the paper describing the operational TROPOMI HCHO retrieval by De Smedt et al., should be added here.

Recently Vigouroux et al., have published an excellent operational TROPOMI HCHO retrieval validation paper (https://www.atmos-meas-tech-discuss.net/amt-2020-30/). To cite in the introduction will provide a nice platform to place the results from this study into context later on.

**Responses:** Thank you very much for this suggestion. We have cited De Smedt et al. and Vigouroux et al. in the introduction.

**Changes in manuscript:** L50, P1 and L54-57, P3 in the revised version.

**Data sets**

The retrieval of slant column densities requires the use of calibrated TROPOMI radiances. This is maybe, the most important

dataset that necessary to obtain the results presented in this work. Its description and how the authors obtained it should be a section in data sets.

**Responses:** Thank you very much for this suggestion. We have added the introduction of TROPOMI radiance in Sect. 2.1.

**Changes in manuscript:** L66-76, P3-4 in the revised version.

1. The original spatial resolution of TROPOPMI observations at NADIR was 3.5 km x 7 km but it improved to 3.5 km x 5.5 km in August 2019.

**Responses:** Thank you very much for this suggestion. We have modified it.

**Changes in manuscript:** L74-75, P4 in the revised version.

2. Fitting parameters considered in the spectral fitting of the operational product?

**Responses:** Thank you very much for this suggestion. The fitting parameters considered in the spectral fitting of the operational product are added in Table 1.

**Changes in manuscript:** L81-82, P4 and Table 1 in the revised version.

3. How is the reference sector correction applied in the operational product?

**Responses:** Thank you very much for this suggestion. The reference sector correction in the operational product is described in Sec. 2.2.

**Changes in manuscript:** L90-99, P4-5 in the revised version.

4. Procedures used to generate earthshine radiance over the remote Pacific in the operational HCHO retrieval.

**Responses:** Thank you very much for this suggestion. Daily detector row averaged radiance over the equatorial Pacific (latitude from 5°S to 5°N and longitude from 180°W to 140°W) is used as reference spectra. Due to residual HCHO signals in reference, the differential SCD (DSCD) is retrieved in spectra fitting.

**Changes in manuscript:** L85-87, P4 in the revised version.

5. Besides a-priori profiles and observation geometry, information about clouds, aerosols and surface reflectance play important roles in AMFs calculations. How is the operational product accounting for them?

**Responses:** Thank you very much for this suggestion. Information about data sets used in AMF calculation was added in Table 1.

**Changes in manuscript:** L222, P10 and Table 1 in the revised version.

6. Does the model simulation include information about pyrogenic sources?

**Responses:** Thank you very much for this suggestion. The open burning emission is obtained from the Fire INventory from NCAR (FINN) model (Wiedinmyer et al., 2010).

**Changes in manuscript:** L118-119, P5 in the revised version.

7. What was the spin off period of the simulations?

**Responses:** Thank you very much for this suggestion. Maybe the referee has a doubt about the spin-up period of the WRF-Chem simulations. WRF-Chem simulation is carried out from July 2019 to July 2019 with five days spun up prior to the simulation.

**Changes in manuscript:** L119-120, P5 in the revised version.

8. It will be interesting to have a reference for the MEIC inventory.

**Responses:** Thank you very much for this suggestion. The paper of (Li et al., 2017) was cited in the revised version.

**Changes in manuscript:** L116, P5 in the revised version.

9. What are the uncertainties associated with TROPOMI HCHO retrievals and MAX-DOAS observations. Table 1 could be expanded with information about the estimated uncertainties for each site as well as the dates and amount of available data.

**Responses:** Thank you very much for this suggestion. Error analysis of TROPOMI HCHO retrieval is added in Section 4. Available days and averaged relative error of MAX-DOAS measurements within ±1 h around the TROPOMI overpass time are added in Table 2.

**Changes in manuscript:** L241-292, P11-13 and Table 2 in the revised version.

**HCHO SCD retrieval: wavelength calibration**

One has to assume that S represents the preflight instrument slit function. It will be very beneficial to discuss the behavior of the preflight slit function. Are they available to the public? How stable the instrument slit function has been after launch?

**Responses:** Thank you very much for this suggestion. The Full-Width at Half-Maximum (FWHM) and asymmetric factor of the instrument slit function are obtained by fitting the daily irradiance to the high resolution solar spectra using the Gauss-Newton Nonlinear Least Squares (NLLS) method assuming the asymmetric Gaussian shape of slit function. The time series of the fitted TROPOMI slit function parameter from August 2018 to July 2019 in Fig. A4 shows that the TROPOMI slit function is stable after lunch. Therefore, the preflight TROPOMI slit function is used in wavelength calibration procedure. The preflight slit function is obtained from the TROPOMI Calibration Key Data (CKD) (available at http://www.tropomi.eu/data-products/isrf-dataset, last access: 22 May 2019) which is derived from TROPOMI calibration measurements performed in March 2015 at CSL in Liege. Comparing the spectral fit residual of using different versions of preflight slit function in the spectral fitting, we found that using version v3.0.0 results in lowest residual (Fig. S3). The preflight instrument slit function version v3.0.0 is used in our retrieval, while the operational product uses version v1.0.0 preflight slit function.

[Figure]

Figure A4. (a) The slit function from TROPOMI CKD at 340nm and online fitted for row 1 and row 450 on 01 August 2018. (b) Time series of FWHM and asymmetric factor of online fitted slit function from August 2018 to July 2019.

**Changes in manuscript:** L159-169, P7-8 in the revised version.

**HCHO SCD retrieval: Radiance fitting**

As mentioned above, there is lack of detail in the description of the methodology employed. Please explain the following questions:

(1) Methodology employed to calculate the daily average earthshine radiance over the Pacific.

**Responses:** Thank you very much for this suggestion. Radiances measured 1 day before the processing day over the Pacific with latitudes ranging from 30°S to 30°N and longitude ranging from 180°W to 140°W are averaged and used as reference in the spectral fit.

**Changes in manuscript:** L179-180, P8 in the revised version.

(2) Any I0 corrected cross sections in the spectral fit.

**Responses:** Thank you very much for this suggestion. $O_3$ cross sections at two temperatures are $I_0$ corrected in the spectral fitting.

**Changes in manuscript:** Table 1 in the revised version.

(3) What is the impact of not including $SO_2$ when if $SO_2$ optical thickness becomes relevant.

**Responses:** Thank you very much for this suggestion. We plotted the spectral fitting residual with (red lines) and without (wathet lines) considering $SO_2$ cross section for HCHO polluted (a) and clean (b) cases in $SO_2$ polluted region (Fig. A1). Considering $SO_2$ cross section has no contribution on reducing residual and increases residual at some wavelengths on the contrary.

(4) What is the impact of not including water vapor?

**Responses:** Thank you very much for this suggestion. Laboratory measurements of water vapor absorption lines only extend to 25,470 $cm^{-1}$ (393 nm) (Dupre et al., 2005). (Lampel et al., 2016) indicated that water vapor can potentially have an impact on the spectral retrievals of tropospheric HCHO while the absorption at 335 nm could not be unambiguously identified in measurements so far.

(5) Which method is employed to estimate Raman spectra?

**Responses:** Thank you very much for this suggestion. The high resolution solar spectrum (Chance and Kurucz, 2010) is convolved with the rotational Raman spectra of $O_2$ and $N_2$ to produce the Raman spectrum (Chance and Spurr, 1997). The paper (Chance and Spurr, 1997) is cited in the revised version.

**Changes in manuscript:** L181, P8 and Table 1 in the revised version.

(5) How was the fitting window selected? HCHO retrievals show big dependencies with fitting windows.

**Responses:** Thank you very much for this suggestion. We use the same fitting window with operational product in DSCD retrieval. The uncertainty of fitting window selection is discussed in Section 4.1.

(6) New O3 cross sections have become available in recent years (for example Serdyuchenko et al., 2014 (https://www.atmos-meas-tech.net/7/609/2014/)). (7) Have the authors taken into account the effect of ozone in the fitted slant columns as described by Pukite et al., 2008 (https://www.atmos-meas-tech.net/3/631/2010/amt-3-631-2010.pdf).

**Responses:** Thank you very much for this suggestion. In our retrieval, we didn't take into account the first order Taylor series expansion for $O_3$ SCD. The uncertainty without considering Taylor expression for $O_3$ SCD is discussed in Section 4.1. We did the sensitivity tests about effect from difference of $O_3$ cross sections and Taylor series approach for HCHO polluted and clean cases. The spectral fitting residuals are plotted in Figure. A1 and fitting SCDs, random errors and RMS are presented in Table A1. For HCHO polluted case, using newly published $O_3$ cross sections changes SCD by 0.7% with random error and RMS unchanged. For HCHO clean case, using newly published $O_3$ cross sections changes SCD by 1.7% with random error unchanged and RMS increasing 0.1. The effect of using Taylor series of $O_3$ cross section in the spectral retrieval is evaluated through sensitivity analysis (Table 3). After reference sector correction, systematic difference regarding to Taylor series of $O_3$ cross section is estimated to be 3.49%. The uncertainties from $O_3$ cross section and without using Taylor series approach are discussed in Section 4.1.

**AMF calculation**
Details missing in the AMF calculation include: (1) Terrain height and surface pressure corrections.

**Responses:** Thank you very much for this suggestion. The surface pressure and terrain height we used are obtained from the operational HCHO product which are corrected.

(2) Origin of cloud information.

**Responses:** Thank you very much for this suggestion. Cloud parameters used in AMF calculation in our retrieval are from the operational TROPOMI cloud product. Cloud fraction is retrieved using the Optical Cloud Recognition Algorithm (OCRA) and cloud top height (pressure) and optical thickness (albedo) are retrieved using Neural Networks (ROCINN) algorithm using the "Clouds-as-Reflecting-Boundaries" (CRB) model, treating clouds as simple Lambertian surfaces (Loyola et al., 2018). The operational TROPOMI cloud product is also available at the Copernicus Open Access Hub (https://scihub.copernicus.eu/, last access: 22 May 2019).

**Changes in manuscript:** L100-105, P5 in the revised version.

(3) Descriptions of the nodes of the box AMF look up table.

**Responses:** Thank you very much for this suggestion. The grid points of parameters in creating the LUT are same with De Smedt et al (2018).

**Changes in manuscript:** L208, P9 in the revised version.

(4) VLIDORT set up.

**Responses:** Thank you very much for this suggestion. The paper (Spurr, 2008) is cited introducing VLIDORT 2.6

**Changes in manuscript:** L207, P9 in the revised version.

(4) Impact of aerosols.

**Responses:** Thank you very much for this suggestion. Aerosol extinction profiles are also retrieved by the MAX-DOAS. Aerosol optical properties, such as single-scattering albedo (SSA) and Ångström exponent obtained from the Aerosol Robotic Network (AERONET) station in Beijing (https://aeronet.gsfc.nasa.gov/, last access: 22 May 2019) are used as input parameters for the MAX-DOAS aerosol profile retrieval. As the aerosol profiles from MAX-DOAS are retrieved at 360nm, we further converted the profile to 340nm using Ångström exponent obtained from the AERONET measurements. To estimate aerosol effect on TROPOMI HCHO retrieval, we calculate the AMFs using MAX-DOAS measured aerosol extinction profiles using VLIDORT (version 2.6) (Spurr, 2008). The AMFs are applied on the operational product and our retrieval. The TROPOMI HCHO VCDs with and without considering aerosols are compared to MAX-DOAS HCHO VCDs. The comparison results are shown in Figure 10. The results show that considering aerosol in the AMF calculations does not improve the agreement with ground based measurements. Considering aerosol effect in TROPOMI retrieval reduces HCHO VCDs by 11.46% ($\pm$ 1.48%) for the operational product and 17.61% ($\pm$ 1.92%) for our retrieval in winter. The reduction over urban site is more significant than suburban sites, mainly due to higher aerosol load. Operational product using both HCHO and aerosol extinction profiles from MAX-DOAS shows underestimation of 8.36 % ($\pm$ 4.63 %). Our retrieval using MAX-DOAS HCHO and aerosol extinction profiles for AMF calculation underestimates HCHO VCD by 18.53% ($\pm$ 4.04 %).

**Changes in manuscript:** L452-466, P20 in the revised version.

(5) Error analysis.

**Responses:** Thank you very much for this suggestion. Error analysis in TROPOMI HCHO retrieval is discussed in Section 4.2.

**Changes in manuscript:** L241-292, P11-13 in the revised version.

**Reference sector correction of SCDs**

Add description of the GEOS-Chem configuration employed in the reference sector correction. Which longitudes define remote Pacific? Is the correction applied only to -30 degrees to 30 degrees? That region is used to calculate the earthshine radiance. Any contributions outside -30 to 30 degrees will not be correcting for the residual HCHO column but for biases of different origin.

**Responses:** Thank you very much for this suggestion. Radiances measured 1 day before the processing day over the Pacific with latitudes ranging from 30°S to 30°N and longitude ranging from 180°W to 140°W are averaged and used as reference in the spectral fit. The first step of reference sector correction is retrieving the DSCDs using average earthshine radiance reference, calculating the corresponding AMFs and storing them as a separate database. The HCHO VCD over remote Pacific Ocean is simulated by GEOS-Chem assuming HCHO over this region is mainly from the oxidation of $CH_4$. The simulated HCHO SCD is calculated by multiplying the VCD ($VCD_G$) taken from GEOS-Chem with the corresponding AMF ($M_0$). Assuming HCHO in the reference sector correction is well simulated by GEOS-Chem, the difference between the simulated and retrieved DSCD ($DSCD_0$) is recognized as the SCD bias caused by residual HCHO signal in reference spectrum. TROPOMI measurements over the Pacific (latitude from 90°S to 90°N and longitude from 160°W to 140°W) are first binned according to their latitude to 500 bins with a resolution of 0.36°. The median value of each bin is then used for the calculation of the SCD background correction (González Abad et al., 2015; González Abad et al., 2016). Assuming the SCD correction is constant in the longitude direction, the SCD correction at 500 gridded latitude points from 90°S to 90°N are linearly interpolated to the latitude of each pixel over China. The interpolated SCD correction is then applied on the retrieved DSCDs to calculate SCDs. Therefore, DSCDs outside -30 to 30 degrees are also corrected. We have clarified in the revised version.

**Changes in manuscript:** L179-180, P8 and L231-240, P11 in the revised version.

**Comparison of operational and improved HCHO product**

What are the coincidence criteria to match TROPOMI and MAX-DOAS measurements? To use daily averages in the case of MAX-DOAS seems inappropriate considering the diurnal variations of HCHO columns.

**Responses:** MAX-DOAS measurements are temporally averaged within $\pm1$ h around the TROPOMI overpass time, while TROPOMI pixels within 20 km of the MAX-DOAS site are spatially averaged for comparison. TROPOMI pixels in our retrieval and operational product are both filtered for intensity-weighted cloud fraction smaller than 0.3, root mean square of spectral fit residual (RMS) smaller than $10^{-3}$, AMF larger than 0.1 and SZA smaller than 70°, quality assurance value (QA value) larger than 0.55 and successful SCD retrieval. The histograms showing the distributions of RMS of spectral fit of the operational product and our retrieval on 06 August 2018 over China are shown in Figure. S5. About 15% and 17% of measurements in the operational product and our retrieval show RMS larger than $10^{-3}$.

**Changes in manuscript:** L307-313, P14 in the revised version.

**Comparisons of SCD retrievals**

This discussion applies to SCDs or $\Delta$SCDs? The result of the spectral fit is in both cases the differential slant column but the

title of the section indicates the comparison of slant columns. In that case, differences are not only due to the spectral fit but also to the reference sector correction. What is the impact of using different earthshine radiances?

**Responses:** Thank you very much for this suggestion. In section 5.1.1, we compare DSCDs and SCDs after applying reference sector correction between operational product and the new retrieval. The biases between DSCDs and RMS in operational product and our retrieval are mainly related to the difference of retrieval method, retrieval settings and selection of reference. To investigate the impact of selection of earthshine radiance reference, we retrieved HCHO DSCDs using daily detector row averaged radiance over the equatorial Pacific (latitude from 5°S to 5°N and longitude from 180°W to 120°W) as reference using BOAS method. Comparisons of HCHO DSCDs and HCHO SCDs after applying reference sector correction using different earthshine radiance reference are presented in Figure. A5. The retrieved DSCDs using different earthshine radiance reference correlated well (R=0.99, Slope=0.99) and with a difference of 30% ($0.11 \pm 0.08 \times 10^{16}$ molec cm$^{-2}$). The bias is compensated (1.9%) by the reference sector correction (Figure. A5 (b)).

[Figure]

Figure A5. Pixel to pixel comparisons of BOAS HCHO DSCDs (a) and BOAS HCHO SCDs (b) using different earthshine radiance reference on 06 August 2018 in the region between 73° E and 130° E, and 18° N and 54° N. The labels with [30°S, 30°N] represent that average of radiances of the equatorial Pacific (latitude from 30°S to 30°N and longitude from 180°W to 140°W) is used as reference spectra. The labels with [5°S, 5°N] represent that average of radiances of the equatorial Pacific (latitude from 5°S to 5°N and longitude from 180°W to 120°W) is used as reference spectra.

**Changes in manuscript:** L339-345, P15 in the revised version and Figure. S6 in the supplement file.

What are the reasons to filter out retrievals with RMS higher than $10^{-3}$. What is the percentage of retrievals with RMS above the threshold? What is the definition of outliers? Showing a histogram of the distributions of both datasets will help to understand the outlier definition.

**Responses:** Thank you very much for this suggestion. The histograms of the distributions of RMS in operational product retrieval and our retrieval are shown in Figure. A6. About 15% and 17% of measurements in the operational product and our

retrieval show RMS larger than $10^{-3}$.

**Changes in manuscript:** L312-313, P14 in the revised version and Figure. S5 in the supplement file.

[Figure]

Figure A6. The histograms of the distributions of RMS in operational product retrieval (a) and our retrieval (b).

The authors make a distinction between non-corrected and corrected SCDs. It is not clear in the text what corrected implies. One has to assume we are talking about reference sector corrected and non-corrected SCDs. If that is the case, one could argue that part of the differences observed between SCDs in the non-corrected case are due to the different selection (or calculation) of earthshine radiance reference. To make a real comparison of the performance of booth fitting algorithms they should be using consistent earthshine radiances. The results seem to indicate that part of the biases are due to using different earthshine radiances since the correlation for the non-corrected and corrected SCDs columns is similar for both products but the bias between them is significantly reduced after applying the correction.

**Responses:** Thank you very much for this suggestion. Due to residual HCHO signals in reference, the differential SCD (DSCD) is retrieved in spectra fitting. Therefore, reference sector correction has to be applied on the retrieved DSCDs to calculate SCDs. The reference sector correction is described on Sec. 3.3. We have cleared it in the revised version. To investigate the impact of selection of earthshine radiance reference, we retrieved HCHO DSCDs using daily detector row averaged radiance over the equatorial Pacific (latitude from 5°S to 5°N and longitude from 180°W to 120°W) as reference using BOAS method. Comparisons of HCHO DSCDs and HCHO SCDs after applying reference sector correction using different earthshine radiance reference are presented in Figure. A5. The retrieved DSCDs using different earthshine radiance reference correlated well (R=0.99, Slope=0.99) and with a difference of 30% ($0.11 \pm 0.08 \times 10^{16}$ molec cm$^{-2}$). The bias is compensated (1.9%) by the reference sector correction (Figure. A5 (b)). To investigate the impact from retrieval method, BOAS HCHO DSCDs using same retrieval settings with operational DSCD retrieval are compared with DOAS HCHO DSCDs (Figure. A7). Using same retrieval settings, difference between DOAS HCHO DSCDs and BOAS HCHO DSCDs (27.33%) is significantly reduced and the remaining difference is due to retrieval algorithm. Besides, smaller difference (4.41%) in HCHO SCDs indicates that

reference sector correction reduces the effect from retrieval method (Figure. A7 (b)).

[Figure]

Figure A7. (a) Pixel to pixel comparisons of DOAS and BOAS HCHO DSCDs, (b) DOAS and BOAS HCHO SCDs and (c) DOAS and BOAS fitting RMS on 06 August 2018 in the region between 73° E and 130° E, and 18° N and 54° N. Same retrieval settings are used in DOAS and BOAS retrieval.

**Changes in manuscript:** L340-349, P15-16 in the revised version and Figure. S7 in the supplement file.

**Comparisons of retrievals after AMF calculations**

One question that raises this section and always permeates the use of high-resolution models is how much information is folded back from the model in the retrieval. To better understand this question it will be very useful to know more about the WRF-Chem simulations and how they were used. The operational product employs daily forecast interpolated in time and space. This procedure should be added to the methodology. How do the box-AMFs of the different retrievals compare?

**Responses:** Thank you very much for this suggestion. Information about more WRF-Chem is added in the Sect. 2.4. A priori HCHO profile for TROPOMI AMF calculations are calculated by interpolating WRF-Chem simulation spatio-temporally to the measurement time and location. This information is added in Sect. 3.2. The information that the operational product employs daily forecast interpolated in time and space is also added in Sect. 2.2.

Besides higher horizontal resolution, WRF-Chem simulation also has a higher vertical spatial resolution compared to TM5-MP data set. Although the difference in vertical resolution only shows negligible effect on box AMF under clear sky condition (Fig. 6 (b)), lower vertical resolution profiles would cause significant impact for cloudy cases due to the interpolation of coarse grid (Fig. 6 (d)).

**Changes in manuscript:** L89, P4, L200-201, P9 and L371-374, P17 in the revised version.

**Comparison between HCHO VCDs observed by MAX-DOAS and TROPOMI**

Blue lines in figure 7 right panels, more than speaking of the retrieval, speak about the difference between the a priori profiles for both models and the retrieved MAX-DOAS profiles. Slopes and correlation coefficients are similar (within error estimates) for both retrievals when using model a priori. It is easier to imagine how the comparison of the operational retrieval with MAX-DOAS observations will also suffer a dramatic improvement if MAX-DOAS a priori were to be used in the AMF calculations.

**Responses:** Thank you very much for this suggestion. We have recalculated HCHO VCD for the operational product by using MAX-DOAS HCHO profiles as a priori profiles. The Pearson correlation coefficient (R) between the recalculated operational

product and MAX-DOAS HCHO VCD decreases by 0.02 to 0.79. The slope of the regression line increases by 0.19 to 0.84 with offset reduces by $0.24 \times 10^{16}$ molec cm$^{-2}$ to $0.15 \times 10^{16}$ molec cm$^{-2}$. Although the recalculated TROPOMI operational product shows improved correlation with MAX-DOAS observations, our retrieval using MAX-DOAS measurements as a priori profile still shows a better agreement with MAX-DOAS HCHO VCDs. It also indicates that the BOAS spectral retrieval of SCDs agree better to the ground based observations over China.

**Changes in manuscript:** L406-412, P18 in the revised version.

**Technical corrections**

Line 18 (grammatical suggestion (GS)): Since what is improved is the HCHO retrieval a more correct grammatical structure will be "We present an improved retrieval of formaldehyde (HCHO) over China from the TROPO…"

**Responses:** Thank you very much for this suggestion. Changed.

Line 23 (GS): remove "the" from "agreement with the ground based…" since it is possible there are more than MAXDOAS measurement than those used in this study.

**Responses:** Thank you very much for this suggestion. Changed.

Line 25 (GS): add "s" to "profile" "…higher resolution a priori profiles".

**Responses:** Thank you very much for this suggestion. Changed.

Line 26: The percentage of what is reported by 61.11% and 0.15%? With the current text it is impossible to know if it refers to the change in the mean VCD, or the percentage of the bias correction attributed to each step. Please specify.

**Responses:** Thank you very much for this suggestion. We added the error bar in Table 3 and find the conclusion that changing SCD retrieval method only shows a tiny effect of 0.15% is less rigorous. The sentence is changed into "The improvements are mainly related to the AMF calculation with more precise a priori profiles in winter. Using more precise a priori profiles in general reduces HCHO VCDs by 52.37 % ($\pm$ 27.09 %) in winter.

**Changes in manuscript:** L24-25, P1 in the revised version.

Line 29 (GS): Change "indicating" to "indicate".

**Responses:** Thank you very much for this suggestion. Changed.

Line 45 (GS): Add "to" after compared "Compared to its predecessor"

**Responses:** Thank you very much for this suggestion. Changed.

Line 53 (GS): Add "s" to "profile" and "a" after "from". Which regional model is employed?

**Responses:** Thank you very much for this suggestion. The AMF calculation is improved by using higher resolution a priori profiles from the regional Weather Research and Forecasting model (WRF-Chem). Changed. A priori HCHO profile for TROPOMI AMF calculations are calculated by interpolating WRF-Chem simulation spatio-temporally to the measurement

time and location.

Line 64 (GS): Change "is" to "are" ".., which are devided in …".
**Responses:** Thank you very much for this suggestion. Changed.

Line 91 (GS): Add "the" in "… is located in the Chinese…"
**Responses:** Thank you very much for this suggestion. Changed.

Line 105: The first step of the methodology explained in this paper is the calculation of ΔSCD, not SCD.
**Responses:** Thank you very much for this suggestion. Changed.

Line 111: There can be other causes for wavelength miss-alignment can be Doppler shift, non-uniform illumination of the slit due to presence of clouds or other high reflectance surface in the pixel.
**Responses:** Thank you very much for this suggestion. Added.
**Changes in manuscript:** L147-148, P7 in the revised version.

Line 115: What is the benefit of having two closure polynomials during wavelength calibration considering that it is done using TOA irradiances and there for are not affected by the presence of clouds or aerosols that may introduce low-frequency structures?
**Responses:** Thank you very much for this suggestion. We compared fitting residuals with and without considering the third baseline and scaling polynomials during wavelength calibration. Considering the third scaling and baseline polynomials decreases the residual largely.

[Figure]

Figure A8. Comparison of fitting residuals with and without considering the third baseline and scaling polynomials during wavelength calibration.
**Changes in manuscript:** L68-171, P8 in the revised version.

Line 142 (GS): Add "presence of" to "… atmosphere (presence of clouds, vertical HCHO distribution…"

**Responses:** Thank you very much for this suggestion. Changed.

Line 143: What is a comprehensive radiative transfer model?

**Responses:** Thank you very much for this suggestion. The sentence seems superfluous because the following content has a detailed introduction about AMF calculation. The sentence is deleted.

Line 150 (GS): Add "on" as "The box AMF depends on wavelength, …"

**Responses:** Thank you very much for this suggestion. Changed.

Line 173 (GS): Add "be" to "… is known to be caused by …"

**Responses:** Thank you very much for this suggestion. The sentence is changed into "Using earthshine radiance over remote Pacific Ocean as reference significantly reduces the influence from unresolved spectral structures which could significantly improve the spectral retrieval of weak absorber, i.e., HCHO."

Line 173: Could the authors provide a reference for the known cause of stripping?

**Responses:** Thank you very much for this suggestion. The sentence is changed into "Using earthshine radiance over remote Pacific Ocean as reference significantly reduces the influence from unresolved spectral structures which could significantly improve the spectral retrieval of weak absorber, i.e., HCHO."

Line 175: AMFs are not retrieved; they are calculated and are independent of a remote Pacific earthshine radiance.

**Responses:** Thank you very much for this suggestion. Changed. The first step of reference sector correction is retrieving the DSCDs using average earthshine radiance reference, calculating the corresponding AMFs and storing them as a separate database.

Line 204 & 209 (GS): Change "outliners" to "outliers".

**Responses:** Thank you very much for this suggestion. Changed.

Line 216 & 217: What is the error associated to the average SCDs? Standard deviations?

**Responses:** Thank you very much for this suggestion. Standard deviations are shown in the brackets associated to the average DSCDs. Changed.

Line 222: How is the 32.32% lower calculated? A difference of 0.02 x $10^{16}$ molecules cm$^{-2}$ looks rather small.

**Responses:** Thank you very much for this suggestion. The sentence is changed to "Averaged SCD taken from the operational product on 06 August 2018 over China ($0.85 \pm 0.69 \times 10^{16}$ molec cm$^{-2}$) is on average 4.49 % lower than our retrieval ($0.89 \pm 0.61 \times 10^{16}$ molec cm$^{-2}$)."

Line 223-226 (GS): Please rewrite sentence.

**Responses:** Thank you very much for this suggestion. The sentences are changed into "We use SCDs in our retrieval and AMFs in operational product to calculated the updated HCHO VCD. The NMB of the two data sets are shown in Table 5. The bias between the updated HCHO VCD and operational HCHO VCD is caused by difference in SCD retrieval. The updated and operational product is also compared to the MAX-DOAS observations."

Line 228: How does BOAS compute mean random errors? Does this calculation use the operation product random error definition? Without describing both is impossible to interpret BOAS random errors lower than DOAS by 22%.

**Responses:** Thank you very much for this suggestion. The random uncertainty in DSCDs retrieval is described in Sect. 4.1.

Figure 1: What is figure 1 trying to illustrate. It is not mention in the text.

**Responses:** Thank you very much for this suggestion. Figure 1 shows the location and HCHO concentration of three MAX-DOAS sites. It is cited in L123, P6 in Section 2.5.

Table 1: What is the definition of viewing azimuth angle? Probably clock wise with respect to North.

**Responses:** Thank you very much for this suggestion. The original Table 1 is Table 2 in the revised version. Viewing azimuth angle of the north is taken as zero degree. It is added in the caption of Table 2.

Table 2: What is the methodology employed to calculate the Raman spectra? What is the definition of Pacific (remote)? Will it be possible to provide a reference for the pre-flight instrument slit function?

**Responses:** Thank you very much for this suggestion. The original Table 2 is Table 1 in the revised version. The high resolution solar spectrum (Chance and Kurucz, 2010) is convolved with the rotational Raman spectra of $O_2$ and $N_2$ to produce the Raman spectrum (Chance and Spurr, 1997). The paper (Chance and Spurr, 1997) is cited in the revised version.

Radiances measured 1 day before the processing day over the Pacific with latitudes ranging from 30°S to 30°N and longitude ranging from 180°W to 140°W are averaged and used as reference in the spectral fit.

The preflight TROPOMI slit function is obtained from the TROPOMI Calibration Key Data (CKD) (available at http://www.tropomi.eu/data-products/isrf-dataset) which is derived from TROPOMI calibration measurements performed in March 2015 at CSL in Liege.

The information is added in the revised version.

**Changes in manuscript:** Table 2, L163-165, P7 and L179-180, P8, in the revised version.

Figure 2: What the figure shows is the ΔSCD not SCD. Please correct. The size of the residuals in plot (b) seem to be large considering an RMS of 2.78 x 10⁻⁴.

**Responses:** Thank you very much for this suggestion. Figure 2 shows the spectral retrieval of HCHO DSCDs and Figure 2 is updated. The values are updated.

Figure 3: A different color map, extending to negative values, will provide a better picture of the BOAS retrieval performance since there is a significant number of pixels with negative columns. Which orbits contribute to these plots?

**Responses:** Thank you very much for this suggestion. The color bar range in Figure 4 is changed into from -2 x $10^{16}$ to 2 x $10^{16}$. And the Figure 3 is updated. The orbits from 04210 to 04213 contribute to these plots.

Figure 4: Does plot (a) show ΔSCDs and plot (b) SCDs? Please clarify. Does this regression consider an independent variable? Probably is better to consider the errors of both variables in the linear fit.

**Responses:** Thank you very much for this suggestion. Figure 5 (a) shows the DSCDs comparison and (b) shows SCDs comparison. It is clarified in the revised version. We don't consider an independent variables in the linear fit. The linear fit in our study is more direct to show the difference between the two data sets.

Table 3: It is very difficult to interpret. How can be the NMBs between satellite and MAX-DOAS be 0% for the operational product? The caption needs to be re-written to provide a proper description of what the table is showing.

**Responses:** Thank you very much for this suggestion. $NMB_{S1,S2}$ is NMBs between TROPOMI HCHO VCDs with different retrieval settings, and $NMB_{S1,M}$ is NMBs between TROPOMI and MAX-DOAS observations. The caption is changed into "NMBs between TROPOMI HCHO VCDs with different retrieval settings ($NMB_{S1,S2}$) and NMBs between TROPOMI and MAX-DOAS observations ($NMB_{S1,M}$). TROPOMI HCHO VCDs are calculated with four different settings, (1) operational retrieval setting (2) replacing DOAS SCDs using BOAS SCDs in the operational product, (3) changing the a priori profiles from TM5 to regional WRF-Chem simulations in the operational product and (4) both (2) and (3) changes in the operational product. The error bars ( ± 2 SE) are also presented. All values are in %."

Figure 5: What is the methodology used to calculate the vertical profiles? Time averaging, filtering of MAX-DOAS observations... Co-registration of models and models with in-situ measurements.

**Responses:** Thank you very much for this suggestion. In AMF calculations, both WRF-Chem and TM5-MP simulations are interpolated to TROPOMI spatial resolution. Interpolated WRF-Chem and TM5-MP simulations within 20 km of the MAX-DOAS site are spatially averaged to compare with MAX-DOAS profiles. MAX-DOAS profiles are temporally averaged in the period 13:30-14:30 (Local Time) within ±1 h around the TROPOMI overpass time.

**Changes in manuscript:** L366-370, P16 in the revised version.

Figure 6: Using a divergent color map scale in plot © will help to appreciate the positive and negative differences. What is the effect of clouds and aerosols? Some of the big differences resemble cloud structures. What is the correlation between both AMF calculations?

**Responses:** Thank you very much for this suggestion. The original Figure 6 is Figure 7 in the revised version. The color bar of Figure. 7(c) is updated to appreciate the positive and negative differences easily. We added the spatial distribution of cloud fraction (d) of orbit 04211 on 06 August 2018 in Figure. 7(d). The results show a similar spatial pattern. However, WRF-Chem AMF is mostly higher than the one calculated with TM5-MP HCHO profiles. The big differences of two AMFs are mainly

occurred in pixels with large cloud fraction.

**Changes in manuscript:** L383-384, P17 in the revised version.

Figure 8: Do these plots show improve HCHO data? It seems to show gridded data. What is the spatial sampling of the grid? Which methodology was used to calculate the gridded fields?

**Responses:** Thank you very much for this suggestion. Figure 8 shows the averaged HCHO VCDs in four seasons. Before averaging, TROPOMI HCHO VCDs in our retrieval are gridded into same grids (0.1°×0.1°). The grid methodology follows the method in (Zhu et al., 2017). The weights are calculated using relative error (Re) and intensity-weighted cloud fraction (cf):

$$\overline{N_V}(i) = \frac{\sum_{p=1}^{p=n(i)} \frac{N_v(p)}{\left(Re(p) * (1 + 3 * cf(p))\right)^2}}{\sum_{p=1}^{p=n(i)} \frac{1}{\left(Re(p) * (1 + 3 * cf(p))\right)^2}}$$

Where $\overline{N_V}(i)$ is gidded HCHO VCDs of grid cell. n(i) is number of satellite pixels falling in the grid cell i.

Boersma, K., Eskes, H., Dirksen, R., Veefkind, J., Stammes, P., Huijnen, V., Kleipool, Q., Sneep, M., Claas, J., and Leitão, J.: An improved tropospheric NO 2 column retrieval algorithm for the Ozone Monitoring Instrument, Atmos. Meas. Tech., 4, 1905-1928, https://doi.org/10.5194/amt-4-1905-2011, 2011a.

Boersma, K. F., Eskes, H., Dirksen, R., Veefkind, J. P., Stammes, P., Huijnen, V., Kleipool, Q., Sneep, M., Claas, J., and Leitao, J.: An improved tropospheric NO2 column retrieval algorithm for the ozone monitoring instrument, Atmos. Meas. Tech., 4, 1905-1928, https://doi.org/10.5194/amt-4-1905-2011, 2011b.

Chance, K., and Kurucz, R. L.: An improved high-resolution solar reference spectrum for earth's atmosphere measurements in the ultraviolet, visible, and near infrared, Journal of quantitative spectroscopy and radiative transfer, 111, 1289-1295, https://doi.org/10.1016/j.jqsrt.2010.01.036, 2010.

Chance, K. V., and Spurr, R. J.: Ring effect studies: Rayleigh scattering, including molecular parameters for rotational Raman scattering, and the Fraunhofer spectrum, Appl Optics, 36, 5224-5230, https://doi.org/10.1364/AO.36.005224, 1997.

De Smedt, I., Theys, N., Yu, H., Danckaert, T., Lerot, C., Compernolle, S., Van Roozendael, M., Richter, A., Hilboll, A., and Peters, E.: Algorithm theoretical baseline for formaldehyde retrievals from S5P TROPOMI and from the QA4ECV project, https://doi.org/10.5194/amt-11-2395-2018, 2018.

Dupre, P., Gherman, T., Zobov, N. F., Tolchenov, R. N., and Tennyson, J.: Continuous-wave cavity ringdown spectroscopy of the 8v polyad of water in the 25195−25340cm−1 range, J. Chem. Phys., 123, 154307, 2005.

González Abad, G., Liu, X., Chance, K., Wang, H., Kurosu, T. P., and Suleiman, R.: Updated Smithsonian Astrophysical Observatory Ozone Monitoring Instrument (SAO OMI) formaldehyde retrieval, Atmos. Meas. Tech., 8, 19-32, https://doi.org/10.5194/amt-8-19-2015, 2015.

González Abad, G., Vasilkov, A., Seftor, C., Liu, X., and Chance, K.: Smithsonian Astrophysical Observatory Ozone Mapping and Profiler Suite (SAO OMPS) formaldehyde retrieval, Atmos. Meas. Tech., 9, 2797-2812, https://doi.org/10.5194/amt-9-2797-2016, 2016.

Lampel, J., Pohler, D., Polyansky, O. L., Kyuberis, A. A., Zobov, N. F., Tennyson, J., Lodi, L., Fries, U., Wang, Y., and Beirle,

S.: Detection of water vapour absorption around 363 nm in measured atmospheric absorption spectra and its effect on DOAS evaluations, Atmos. Chem. Phys., 17, 1271-1295, 2016.

Li, M., Zhang, Q., Kurokawa, J., Woo, J., He, K., Lu, Z., Ohara, T., Song, Y., Streets, D. G., and Carmichael, G. R.: MIX: a mosaic Asian anthropogenic emission inventory under the international collaboration framework of the MICS-Asia and HTAP, Atmos. Chem. Phys., 17, 935-963, https://doi.org/10.5194/acp-17-935-2017, 2017.

Spurr, R.: LIDORT and VLIDORT: Linearized pseudo-spherical scalar and vector discrete ordinate radiative transfer models for use in remote sensing retrieval problems, in: Light Scattering Reviews 3: Light Scattering and Reflection, edited by: Kokhanovsky, A. A., Springer Berlin Heidelberg, Berlin, Heidelberg, 229-275, 2008.

Wiedinmyer, C., Akagi, S. K., Yokelson, R. J., Emmons, L. K., Alsaadi, J. A., Orlando, J. J., and Soja, A. J.: The Fire INventory from NCAR (FINN): a high resolution global model to estimate the emissions from open burning, Geosci Model Dev, 4, 625-641, https://doi.org/10.5194/gmd-4-625-2011, 2010.

Zhou, Y., Brunner, D., Boersma, K. F., Dirksen, R., and Wang, P.: An improved tropospheric NO2 retrieval for OMI observations in the vicinity of mountainous terrain, Atmos. Meas. Tech., 2, 401-416, https://doi.org/10.5194/amt-2-401-2009, 2009.

Zhu, L., Jacob, D. J., Keutsch, F. N., Mickley, L. J., Scheffe, R., Strum, M., Gonzalez Abad, G., Chance, K., Yang, K., Rappengluck, B., Millet, D. B., Baasandorj, M., Jaegle, L., and Shah, V.: Formaldehyde (HCHO) As a Hazardous Air Pollutant: Mapping Surface Air Concentrations from Satellite and Inferring Cancer Risks in the United States, Environ Sci Technol, 51, 5650-5657, https://doi.org/10.1021/acs.est.7b01356, 2017.

---

## Author Comment (AC2) · 26 Jul 2020

Response to reviewer #2

We really appreciate your constructive comments and suggestions on our manuscript. We have considered every comment carefully, responded on a point to point and marked every change in red in the revised version.

**General comments**

The paper aims to present an improved TROPOMI HCHO retrieval over China. Compared to the ESA operational product, two major differences are highlighted: (1) the use of BOAS instead of DOAS for the of fit the slant columns, (2) the use of a priori profiles from a regional model in order to recalculate the AMF, with a finer spatial resolution, and optimized emissions over China.

Overall, the paper fails in showing an improvement of the slant columns, the differences between the two products being negligible.

**Responses:** Thank you very much for this suggestion. We using a different spectral retrieval technique (BOAS method) of HCHO slant columns. Moreover, the background correction has also been improved. Although DOAS and BOAS HCHO DSCDs show a similar spatial pattern. The spatial distribution of BOAS HCHO SCD is expected to be smooth, less noisy. Besides, operational product using different SCD retrieval methods is compared with MAX-DOAS HCHO measurements. Using the BOAS HCHO SCDs reduced the underestimation in summer and overestimation in winter of the operational product. In summer, using different SCD retrieval methods results a difference of 7.00% ($\pm$ 1.71%, $\pm$ Error) from the TROPOMI operational HCHO VCD. The result shows that using the BOAS HCHO SCDs reduced the underestimation in summer and overestimation in winter of the operational product.

The scientific interest of the paper lies in the second improvement. The authors should focus more on this aspect, and go further into a detailed analysis of the spatio-temporal effects of using more precise profiles for satellite HCHO observations. However, it is not demonstrated how the finer spatial resolution of the model improves the validation. Here it could help to show that the improvement is more important over the urban site compared to the sub-urban sites. Or is it more an effect of the different model chemistry/emissions, and not a spatial resolution effect?

**Response:** Thank you very much for this suggestion. We followed the reviewer's comment. We also compared simulated a priori HCHO profiles and MAX-DOAS HCHO profiles at suburban site (UCAS) in the revised version. The improvement of WRF-Chem simulations at urban site is more significant than suburban site, which is mainly related to the finer spatial resolution and more up to date emission inventory over China used in the simulations. The bias between simulated and measured HCHO profiles at urban site is larger than that at suburban site, which is mainly due to smaller spatial gradient over suburban areas.

Moreover, TROPOMI HCHO VCDs are compared with MAX-DOAS measurements. The results show that the improvements of our VCD retrieval in winter time is more significant than summer and the improvement at urban site is also more significant than suburban sites in winter. The results indicates better anthropogenic emission inventory and the finer spatial resolution over China in WRF-Chem simulations.

Distributions of operational HCHO VCD in four seasons are compared with distributions of the improved HCHO VCD in Section 5.4. Operational product shows similar spatio-temporal distribution with our retrieval over China while the absolute

values are (slight) smaller than our retrieval (Figure 12). In summer, hotspots can be observed over BTH, YRD, PRD, Shandong province, Henan province, Wuhan (Hubei's provincial capital), SCB and cities along Fen nutrient-laden valley in Shaanxi and Shanxi provinces in our retrieval. These hotspot patterns are strongly correlated to the population density and industrial emission pattern indicating a significant anthropogenic contribution. These hotspots are less obvious in the operational product in summer.

In a conclusion, both enhanced emission inventory and resolution of WRF-Chem simulations contribute to improving TROPOMI HCHO retrieval.

Along the paper, the numbers are often used in a quite subjective way. (0.15% difference being called an improvement for exemple). I recommend writing quantitative comparisons with a more rigorous analysis to strengthen the message of the paper.

**Responses:** Thank you very much for this suggestion. In the revised version, we added the error bar in calculating the NMBs (Table 3). The error bars of NMBs are two times standard error (SE) and is calculated following:

$$\text{Error} = 2 \times \sqrt{\frac{1}{n(n-1)} \frac{\sum_{i=1}^{i=n}\left(V_T(i) - V_M(i) - \overline{V_T(i) - V_M(i)}\right)^2}{\left(\sum_{i=1}^{i=n} V_M(i)/n\right)^2}} \times 100\% \qquad (A1)$$

The error shows 95% confidence interval (Streiner, 1996) and The NMBs larger than error are statistically significant. The 0.15% difference is statistically insignificant. The sentence "while the SCD retrieval only shows a minor effect of 0.15 %" is deleted.

**Changes in manuscript:** L302-306, P14-15 in the revised version.

The paper needs major revisions before being published in AMT.

**Specific comments**

The title does not fairly reflect the contents of the paper. Unless the results are significantly extended, the title of the paper should focus more on the "improved AMF calculation over China".

**Responses:** Thank you very much for this suggestion. In addition to the improvement of AMF calculation, we are also using a different spectral retrieval technique of HCHO slant columns. Moreover, the background correction has also been improved. We think the title suggested by the reviewer neglected these points, and therefore, would like to keep the title of the manuscript as is.

The section called "Improved HCHO retrieval algorithm" presents the retrieval algorithm developed for this study. As described in this section, it is actually very similar to the ESA operational product. The similarities and the differences need to be clearly explained. For example, the wavelength calibration. The description is the same as for the operational product. Why a specific section dedicated to this aspect? It would be interesting to see a comparison of the calibration results between the 2 products.

**Responses:** Thank you very much for this suggestion. The theory of wavelength calibration in our retrieval is same with operational product. While the parameters used in the wavelength calibration including the TROPOMI slit function and polynomial orders are different from operational product. The preflight slit function is obtained from the TROPOMI

Calibration Key Data (CKD) (available at http://www.tropomi.eu/data-products/isrf-dataset, last access: 22 May 2019) which is derived from TROPOMI calibration measurements performed in March 2015 at CSL in Liege. Comparing the spectral fit residual of using different versions of preflight slit function in the spectral fitting, we found that using version v3.0.0 results in lowest residual (Fig. A1). The preflight instrument slit function version v3.0.0 is used in our retrieval, while the operational product uses version v1.0.0 preflight slit function. In the operational algorithm, polynomials are not considered in wavelength calibration. In our retrieval, the third order polynomials are selected through sensitivity analysis (Fig. A1). The result shows that using the third order polynomials contributes to reducing residual in wavelength calibration.

**Changes in manuscript:** L163-171, P7-8 in the revised version.

[Figure]

Figure A1. Comparisons of spectral fit residuals using different version of preflight slit function and using different polynomials during wavelength calibration.

The same holds for the AMF calculation part. It is very similar to the Tropomi HCHO ATBD and the differences are not clearly explained, except for the a priori profiles. Same for the reference sector correction.

**Responses:** Thank you very much for this suggestion. The AMF calculation is improved by using more precise a priori HCHO profiles in my study. Cloud information, surface albedo and surface pressure used in AMF calculations in our retrieval is same with operational product. The similarities and differences are listed in Table 1.

In Reference sector correction, we improved the reference sector correction by considering the variability of $M_0/M$ ratio.

**Changes in manuscript:** Table 1 and L240, P11 in the revised version.

It is not shown in the paper that the SCDs have been improved. There is a contradiction between the introduction ("BOAS has been reported featured with lower fitting uncertainties to the standard DOAS method") and the result section, where it is stated (page 10) that the RMS are identical. So the "lower fitting uncertainties" of the BOAS technique are not demonstrated. As for Figure 4 with the slant columns, a figure with RMS comparison needs to be added.

**Responses:** Thank you very much for this suggestion. The sentence at Line 341 in page 15 "On the other hand, RMS of both methods is very similar." is not rigorous. We added RMS comparisons in the Figure 5 in the revised version. RMS comparisons shown in Fig. 5(c) indicate that averaged RMS of the DOAS retrieval ($6.1 \pm 1.56\times10^{-4}$) is slightly higher than that of BOAS

retrieval ($5.95 \pm 1.50 \times 10^{-4}$). The sentence "BOAS has been reported featured with lower fitting uncertainties to the standard DOAS method" is deleted.

**Changes in manuscript:** L336-337, P15 in the revised version.

In section 4.1.2, it is explained that the operational product has been updated using a priori profile from WRF model. Does it mean that operational averaging kernel have been used? Or did the authors used their own radiative transfer calculation? It is important to know in order to understand if other sources of differences, such as the albedo, can play a role in the observed vcd differences.

**Responses:** Thank you very much for this suggestion. The operational averaging kernel (AK) has not been used in AMF calculation. We recalculated AMF using LUT table following Sect 3.2. The parameters used in AMF calculation are same with operational algorithm except for a priori HCHO profiles. Besides higher horizontal resolution, WRF-Chem simulation also has a higher vertical spatial resolution compared to TM5-MP data set. Although the difference in vertical resolution only shows negligible effect on box AMF (below 2%) under clear sky condition (Fig. A2 (b)), lower vertical resolution profiles would cause significant impact for cloudy cases due to the interpolation of coarse grid (Fig. A2 (d)). Therefore, the operational averaging kernel was not used.

**Changes in manuscript:** L371-374, P17 in the revised version.

[Figure]

Figure A2. Daily averaged vertical HCHO profiles obtained from MAX-DOAS, WRF-Chem and TM5-MP model in clean case on 03 March 2019 (a) and in polluted case on 26 June 2019 (c). Comparisons of box AMF using WRF-Chem and TM5-MP simulations in clean case with clear sky on 03 March 2019 (b) and in polluted case with cloudy sky on 26 June 2019 (d). The locations of two pixels are within 20 km of the CAMS site.

Improved AMF for China should include some tests about the aerosol effects. They are not even mentioned.

**Responses:** Thank you very much for this suggestion. Aerosol effect on TROPOMI HCHO retrieval is discussed in Section 5.3. To estimate aerosol effect on TROPOMI HCHO retrieval, we calculate the AMFs using MAX-DOAS measured aerosol extinction profiles using VLIDORT (version 2.6) (Spurr, 2008). The AMFs are applied on the operational product and our retrieval. The TROPOMI HCHO VCDs with and without considering aerosols are compared to MAX-DOAS HCHO VCDs. The comparison results are shown in Figure 10. The results show that considering aerosol in the AMF calculations does not improve the agreement with ground based measurements. Considering aerosol effect in TROPOMI retrieval reduces HCHO VCDs by 11.46% (± 1.48%) for the operational product and 17.61% (± 1.92%) for our retrieval in winter. The reduction over urban site is more significant than suburban sites, mainly due to higher aerosol load. Operational product using both HCHO and aerosol extinction profiles from MAX-DOAS shows underestimation of 8.36 % (± 4.63 %). Our retrieval using MAX-DOAS HCHO and aerosol extinction profiles for AMF calculation underestimates HCHO VCD by 18.53% (± 4.04 %).

**Changes in manuscript:** L453-466, P20 in the revised version.

When comparing to the MAX-DOAS data in Beijing, MAX DOAS profiles have been used to re-calculate the amf of the improved Chinese HCHO product (Figure 7). For a fair comparison, the same method needs to be applied to the operational product. All the needed information are provided in the operational L2 files.

**Responses:** Thank you very much for this suggestion. We have recalculated HCHO VCD for the operational product by using MAX-DOAS HCHO profiles as a priori profiles. The Pearson correlation coefficient (R) between the recalculated operational product and MAX-DOAS HCHO VCD decreases by 0.02 to 0.79. The slope of the regression line increases by 0.19 to 0.84 with offset reduces by $0.24 \times 10^{16}$ molec cm$^{-2}$ to $0.15 \times 10^{16}$ molec cm$^{-2}$.

**Changes in manuscript:** L406-409, P18 and Figure 10 in the revised version.

I have some concerns about the way validation results are presented; The operational product, such as most of existing HCHO satellite products, is rather known to be underestimated over emission regions such as Beijing. See for example Jung et al. 2019 (https://doi.org/10.1029/2019EA000702) or Vigouroux et al. 2020 (https://doi.org/10.5194/amt-2020-30) and references therein. Here the authors claim to find an opposite result. The operational product is overestimated, and the improved Chinese product is lower and in better match with the MAX-DOAS. But actually this result holds for winter time only. The results should be discussed more in terms of low column (winter) or high columns (summer). Finally, a link to previous satellite HCHO validation studies should be made, and the reasons for such different conclusions need to be discussed.

**Responses:** Thank you very much for this suggestion. The study of Jung et al. 2019 (https://doi.org/10.1029/2019EA000702) shows that excluding the aerosol effect underestimates HCHO VCDs over East China during 2006-2007. While HCHO VCDs were not compared with ground-based instruments.

At CAMS site, HCHO VCDs are underestimated by TROPOMI observations and the underestimation of our retrieval (5.78 $\pm$ 3.49 %) is slightly smaller (29.77 $\pm$ 22.83 %) than that of the operational product (8.23 $\pm$ 3.09 %). At UCAS site, both our retrieval and operational product overestimate HCHO VCDs and the overestimation of our retrieval (5.22 $\pm$ 3.49 %) is smaller (51.44 $\pm$ 33.70 %) than that of the operational product (10.75 $\pm$ 3.09 %). The overestimation of operational product at UCAS site is opposite to previous study that TROPOMI operational product underestimated HCHO VCDs in Xianghe located at ~ 50 km southeast of Beijing compared to FTIR measurements (Vigouroux et al., 2020). In order to investigate the reason of the overestimation, we separated the data by seasons for comparison at three MAX-DOAS sites (Table 5). In summer, high HCHO columns are due to oxidation of VOCs related to enhanced biogenic emissions. Both our retrieval and operational product underestimate HCHO VCD in summer which is consistent with previous studies (Vigouroux et al., 2020; Chan et al., 2020). In wither, lower HCHO columns ($< 1\times10^{16}$ molec cm$^{-2}$) are mainly related to anthropogenic emissions including vehicle exhaust and industrial emissions. The vertical HCHO profiles simulated by WRF-Chem model are similar to the one measured by MAX-DOAS in winter, while the TM5-MP profiles show larger difference to the MAX-DOAS measurements. Both our retrieval and operational product overestimate HCHO VCDs in winter. The overestimation of our retrieval (11.23 $\pm$ 4.61 %) is 63.18 % ($\pm$ 22.63 %) smaller than operational product (29.08 $\pm$ 5.59 %) and the overestimation in urban area (QKY) is smaller than that in suburban areas (UCAS and NC sites). The improvements of our retrieval in winter time are more significant and the improvement at urban site is also more significant than suburban sites in winter, which are mainly related to better anthropogenic emission inventory and the finer spatial resolution over China in WRF-Chem simulations. In order to eliminate effect from a priori HCHO profile, a priori HCHO profiles from MAX-DOAS measurements are used for AMF calculations for comparison. The overestimations at CAMS and NC sites become less significant. The overestimation at UCAS site of our retrieval reduces to 3.94 % ($\pm$ 6.87 %), while the overestimation of operational product reduces to 10.60 % ($\pm$ 8.71 %). Our result shows an overestimation of TROPOMI HCHO VCDs during winter at UCAS site. Our result is different from the previous FTIR comparison study (Vigouroux et al., 2020) which shows TROPOMI underestimated HCHO columns in winter. On the one hand, the pollution conditions at three MAX-DOAS sites are different from the FTIR sites used in Vigouroux et al. (2020). On the other hand, HCHO concentrations in the lower troposphere is lower in winter, resulting a relatively larger portion of HCHO above the MAX-DOAS retrieval height of 3km. Therefore, the MAX-DOAS measurements show an underestimation in winter. Our findings are consistence with the previous study that SCIAMACHY HCHO VCDs are in general lower than FTIR measurements while higher than MAX-DOAS observations (Vigouroux et al. 2009). The remaining overestimation is mainly related to a portion of HCHO above 3 km where the MAX-DOAS is not sensitive. In addition, the TROPOMI retrieval assumes aerosol free atmosphere which might also lead to the overestimations.

**Changes in manuscript:** L415-424, P18-19, L427-431, P19 and L434-451, P19-20 in the revised version.

The last paragraph of section 4.2 is the most interesting part of the paper and deserves to be extended. It is found that both algorithms remain underestimated in summer time, when the columns are the largest and mainly related to biogenic emissions. Both models simulate profiles not peaked enough near the surface. However, in winter time, when the columns are the lowest (no biogenic emissions), an improvement is observed compared to the MAX-DOAS observations when using WRF-Chem model as a priori profiles. Can you say something about possible reasons for this? Does the WRF-Chem model perform better than TM5 for anthropogenic emissions? Is it related to the spatial resolution or to the chemistry?

**Responses:** Thank you very much for this suggestion. The anthropogenic emission in WRF-Chem is obtained from The Multi-resolution Emission Inventory for China (MEIC). The MEIC emission inventory has improved the emissions estimation from power plants (Liu et al., 2015), vehicles (Zheng et al., 2014), and residential combustions of non-methane volatile organic compounds (NMVOCs) (Li et al., 2013; Peng et al., 2019).

Comparisons of seasonal averaged a priori HCHO profiles from MAX-DOAS, WRF-Chem and TM5-MP simulations at suburban (UCAS) sites in four seasons are added in the revised version. The improvement of WRF-Chem simulations at urban site is more significant than suburban site, which is mainly related to the finer spatial resolution and more up to date emission inventory over China used in the simulations.

Moreover, TROPOMI HCHO VCDs are compared with MAX-DOAS measurements. The improvements of our VCD retrieval in winter time is more significant than summer and the improvement at urban site is also more significant than suburban sites in winter, which are mainly related to better anthropogenic emission inventory and the finer spatial resolution over China in WRF-Chem simulations.

We think that both enhanced emission inventory and resolution of WRF-Chem simulations contribute to improving TROPOMI HCHO retrieval.

**Changes in manuscript:** L437-439, P19 in the revised version.

The discussion about seasonal variation of the improved Chinese product, and its spatial distribution over China does not bring anything new about current HCHO satellite observations. I advise to either remove this part, either extend with meaningful observations going much more into details. Comparison maps of SCD and AMF are shown. It would be good to do the same for the final VCD.

**Responses:** Thank you very much for this suggestion. The maps of operational HCHO VCDs in four seasons are added in the revised version. Spatial distributions of operational VCD and our retrieval are compared in Section 5.3. Operational product shows similar spatio-temporal distribution with our retrieval over China while the values are smaller than our retrieval (Figure 10). In summer, hotspots can be observed over BTH, YRD, PRD, Shandong province, Henan province, Wuhan (Hubei's provincial capital), SCB and cities along Fen nutrient-laden valley in Shaanxi and Shanxi provinces in our retrieval. These hotspot patterns are strongly correlated to the population density and industrial emission pattern indicating a significant anthropogenic contribution. These hotspots is less obvious in map of the operational HCHO VCDs in summer. The distribution of high HCHO VCDs observed in our retrieval over the Xinjiang Uygur Autonomous Region is related to the unique topography and industrial areas. The total industrial VOCs emission over the Xinjiang Uygur Autonomous Region is higher than Shanxi province (Zheng et al., 2017).

**Changes in manuscript:** L473-474 and L477-480, P21 and Figure 12 in the revised version.

**Technical corrections**

**Abstract**

L18: We present  an improved retrieval…

**Responses:** Thank you very much for this suggestion. Changed.

L19: The new retrieval optimizes the slant column density retrieval: this is not demonstrated in the paper. Please rephrase.

**Responses:** Thank you very much for this suggestion. We using a different spectral retrieval technique (BOAS method) of HCHO slant columns. Moreover, the background correction has also been improved. Although DOAS and BOAS HCHO DSCDs show a similar spatial pattern. The spatial distribution of BOAS HCHO SCD is expected to be smooth, less noisy. Besides, operational product using different SCD retrieval method is compared with MAX-DOAS HCHO measurements. Using the BOAS HCHO SCDs reduced the underestimation in summer and overestimation in winter of the operational product. In summer, using different SCD retrieval methods results a difference of 7.00% (± 1.71%, ± Error) from the TROPOMI operational HCHO VCD. The result also shows that using the BOAS HCHO SCDs reduced the underestimation in summer and overestimation in winter of the operational product.

L24: MAX-DOAS measurements in  Beijing

**Responses:** Thank you very much for this suggestion.

L26: while the SCD retrieval only shows a minor effect of 0.15%. This is negligible! We can even talk about a perfect agreement between the SCD retrievals.

**Responses:** Thank you very much for this suggestion. The sentence is deleted. Using different SCD retrieval methods results a difference of 7.00% (± 1.71%, ± Error) from the TROPOMI operational HCHO VCD in summer. So we cannot talk about a perfect agreement between the SCD retrievals.

L29-30: The last sentence is not demonstrated in the paper.

**Responses:** Thank you very much for this suggestion.

TROPOMI HCHO VCDs are compared with MAX-DOAS measurements. The improvements of our VCD retrieval in winter time is more significant than summer and the improvement at urban site is also more significant than suburban sites in winter, which are mainly related to better anthropogenic emission inventory and the finer spatial resolution over China in WRF-Chem simulations.

Moreover, the spatio-temporal distribution of the improved and operational HCHO VCDs is compared in Section 5.4 in the revised version. Operational product shows similar spatio-temporal distribution with our retrieval over China while the values are smaller than our retrieval (Figure 10). In summer, hotspots can be observed over BTH, YRD, PRD, Shandong province, Henan province, Wuhan (Hubei's provincial capital), SCB and cities along Fen nutrient-laden valley in Shaanxi and Shanxi provinces in our retrieval. These hotspot patterns are strongly correlated to the population density and industrial emission pattern indicating a significant anthropogenic contribution. These hotspots is less obvious in map of the operational HCHO VCDs in summer. These hotspots are less obvious in the operational product in summer. The distribution of high HCHO VCDs observed in our retrieval over the Xinjiang Uygur Autonomous Region is related to the unique topography and industrial areas. The total industrial VOCs emission over the Xinjiang Uygur Autonomous Region is higher than Shanxi province (Zheng et al., 2017).

These results show that our retrieval is more suitable for the analysis of regional and city scale pollution in China.

**Changes in manuscript:** L473-474 and L477-480, P21 and Figure 12 in the revised version.

**Introduction:**

L48: Again, It is not shown in the paper that the SCDs have been improved. The "lower fitting uncertainties" of the BOAS technique are not demonstrated.

**Responses:** Thank you very much for this suggestion. Thank you very much for this suggestion. We using a different spectral retrieval technique (BOAS method) of HCHO slant columns. Moreover, the background correction has also been improved. Although DOAS and BOAS HCHO DSCDs show a similar spatial pattern. The spatial distribution of BOAS HCHO SCD is expected to be smooth, less noisy. Besides, operational product using different SCD retrieval method is compared with MAX-DOAS HCHO measurements. Using the BOAS HCHO SCDs reduced the underestimation in summer and overestimation in winter of the operational product. In summer, using different SCD retrieval methods results a difference of 7.00% ($\pm$ 1.71%, $\pm$ Error) from the TROPOMI operational HCHO VCD. The result also shows that using the BOAS HCHO SCDs reduced the underestimation in summer and overestimation in winter of the operational product. The sentence "This technique has been reported featured with lower fitting uncertainties compared to the standard differential optical absorption spectroscopy (DOAS) method (Chance and Kurosu, 2003)." is deleted.

L54: the result is expected to be more realistic for the investigation of spatio temporal variation of HCHO over China. ok but this needs to be demonstrated. How the spatio temporal variation of HCHO over China has been improved? In the current version, only a reduction of the bias compared to MAX-DOAS data is shown in winter time.

**Responses:** Thank you very much for this suggestion.

Operational and improved HCHO VCDs are compared with MAX-DOAS measurements. The improvements of our VCD retrieval in winter time is more significant than summer and the improvement at urban site is also more significant than suburban sites in winter, which are mainly related to better anthropogenic emission inventory and the finer spatial resolution over China in WRF-Chem simulations.

Moreover, the spatial distributions of the improved and operational HCHO VCDs in four seasons are compared in Section 5.4 in the revised version. In summer, hotspots can be observed over BTH, YRD, PRD, Shandong province, Henan province, Wuhan (Hubei's provincial capital), SCB and cities along Fen nutrient-laden valley in Shaanxi and Shanxi provinces in our retrieval. These hotspot patterns are strongly correlated to the population density and industrial emission pattern indicating a significant anthropogenic contribution. These hotspots are less obvious in the operational product in summer.

These results show that our retrieval is expected to be more realistic for the investigation of spatio temporal variation of HCHO over China.

**Changes in manuscript:** L473-474 and L477-480, P21 and Figure 12 in the revised version.

Figure 1: The scale could be reduced to better show emission spots.

**Responses:** Thank you very much for this suggestion. The scale is reduced to 1.5.

**WRF-model**

L79: more up to date emission inventory of China: this is really vague and needs to be explained

**Responses:** Thank you very much for this suggestion. The anthropogenic and biogenic emissions are obtained from The Multi-resolution Emission Inventory for China (MEIC) and the Model of Emissions of Gases and Aerosols from Nature (MEGAN) (Guenther et al., 2006; Li et al., 2017), respectively. The MEIC emission inventory has improved the emissions estimation from power plants (Liu et al., 2015), vehicles (Zheng et al., 2014), and residential combustions of non-methane volatile organic compounds (NMVOCs) (Li et al., 2013; Peng et al., 2019).
**Changes in manuscript:** L116-118 in the revised version.

**Improved HCHO retrieval algorithm**
L133: in Table  2
**Responses:** Thank you very much for this suggestion. Changed

Table 2:
● Why the use of DSCD in the caption?
**Responses:** Thank you very much for this suggestion. Daily detector row averaged radiance over the equatorial Pacific is used as reference spectra. Due to residual HCHO signals in reference, the differential SCD (DSCD) is retrieved in spectra fitting.
**Changes in manuscript:** L86-87, P4 in the revised version.

● Please indicate the differences compared to the operational product.
**Responses:** Thank you very much for this suggestion. The similarities and differences are added in Table 1.
**Changes in manuscript:** Table 1 in the revised version.

● What about the Ring correction?
**Responses:** Thank you very much for this suggestion. The term $\alpha_r X_r(\lambda)$ in Eq. (3) represents Ring effect. $X_r(\lambda)$ is Raman spectrum calculated in Chance and Spurr (1997).
**Changes in manuscript:** Table 1 in the revised version.

● Do you include corrections for non-linear Ozone absorption effects?
**Responses:** Thank you very much for this suggestion. In our fitting, wavelength dependency of $O_3$ SCDs are not considered. We added the uncertainty analysis from wavelength dependency of $O_3$ SCDs. The uncertainty it causes on SCD is about 3.49%.
**Changes in manuscript:** L267-269, P12 and Table 3 in the revised version.

● How is the radiance reference sector calculated? Per instrument row? Per day?
**Responses:** Thank you very much for this suggestion. Radiances measured 1 day before the processing day over the Pacific with latitudes ranging from 30°S to 30°N and longitude ranging from 180°W to 140°W are averaged and used as reference in the spectral fit.
**Changes in manuscript:** L179-180, P8 in the revised version.

● Why this particular choice of $O_3$ and BrO cross-sections? Can these choices explain the differences with the operational product?

**Responses:** Thank you very much for this suggestion. The $O_3$ and BrO cross-sections are chose following the study of González Abad et al. (2015). The biases between DSCDs and RMS in operational product and our retrieval are mainly related to the difference of retrieval method, retrieval settings and selection of reference.

To eliminate the impact from retrieval settings, BOAS HCHO DSCDs using same retrieval settings with operational DSCD retrieval are compared with DOAS HCHO DSCDs (Figure. A3). Using same retrieval settings, difference between DOAS HCHO DSCDs and BOAS HCHO DSCDs (27.33%) is significantly reduced and the remaining difference is due to retrieval algorithm. Besides, smaller difference (4.41%) in HCHO SCDs indicates that reference sector correction reduces the effect from retrieval method (Figure. A3 (b)).

[Figure]

**Figure A3.** (a) Pixel to pixel comparisons of DOAS and BOAS HCHO DSCDs, (b) DOAS and BOAS HCHO SCDs and (c) DOAS and BOAS fitting RMS using same retrieval settings on 06 August 2018 in the region between 73° E and 130° E, and 18° N and 54° N.

**Changes in manuscript:** L345-349, P15-16 in the revised version.

L168: the surface albedo is obtained from the S5P operational cloud product. This is a bit surprising. Please specify the wavelength.

**Responses:** Thank you very much for this suggestion. The surface albedo is extracted from the S5P operational HCHO product in which surface albedo is from OMI-based monthly minimum LER at 342 nm for HCHO fitting window. We deleted the sentence and added the information used in AMF calculation in Table 1.

**Changes in manuscript:** Table 1, P12 in the revised version.

L187: specify the meaning of k and m

**Responses:** Thank you very much for this suggestion. In the old version, k and m are the position along track and across track of satellite pixels. The expression in the old version maybe is difficult to understand. We have changed the expression of the Eq. (10) in the revised version.

**Changes in manuscript:** L239, P11 in the revised version.

**Results and discussions**

L196: VT and Vm is the average tropospheric CHO VCD measured by TROPOMI and MAX-DOAS. How are the data averaged in space / time?

**Responses:** Thank you very much for this suggestion. MAX-DOAS measurements are temporally averaged within ±1 h around the TROPOMI overpass time, while TROPOMI pixels within 20 km of the MAX-DOAS site are spatially averaged for comparison. TROPOMI pixels in our retrieval and operational product are both filtered for intensity-weighted cloud fraction smaller than 0.3, root mean square of spectral fit residual smaller than $10^{-3}$, AMF larger than 0.1 and SZA smaller than 70°, quality assurance value (QA value) larger than 0.55 and successful SCD retrieval.

**Changes in manuscript:** L307-311, P14 in the revised version.

Figure 3: Do the maps show SCD, DSCD (as mentioned in Table 2) or corrected SCDs? It would be good to show corrected SCDs (with a color scale including negative values), since an offset is found in the SCDs. It would help to better see differences in the two spatial distributions.

**Responses:** Thank you very much for this suggestion. Figure 3 in the old version shows the maps of DSCDs which are not corrected. Figure 3 in the old version is Figure 4 in the revised version. We added the maps of SCDs which are calculated by applying reference sector correction on DSCDs (Fig. 4 (b) and (d)). The color scale in Fig. 4 is changed to include negative values.

**Changes in manuscript:** L352-355, P16 and Figure 4 in the revised version.

L206: Please compare numbers for the corrected slant columns over Tibet.

**Responses:** Thank you very much for this suggestion. The numbers of valid satellite measurements over Tibet for BOAS and DOAS retrieval are 22244 and 21987, respectively.

**Changes in manuscript:** L325-326, P15 in the revised version.

L218: Please compare the RMS.

**Responses:** Thank you very much for this suggestion. Figure 5 in the revised version adds pixel to pixel comparison of RMS on 06 August 2018 in the region between 73° E and 130° E, and 18° N and 54° N. The result indicate that averaged RMS of the DOAS retrieval ($6.1 \pm 1.56 \times 10^{-4}$) is slightly higher than that of BOAS retrieval ($5.95 \pm 1.50 \times 10^{-4}$).

**Changes in manuscript:** L336-337, P15 and Figure 5 in the revised version.

L218-219: This sentence is vague. Please be more specific

**Responses:** Thank you very much for this suggestion. The biases between DSCDs and RMS in operational product and our retrieval is mainly related to the difference of retrieval method, retrieval settings and selection of earthshine radiance reference. The impact of selection of earthshine radiance reference and retrieval method are also investigated in the revised version.

**Changes in manuscript:** L339-349, P15-16 in the revised version.

L223: Please give the numbers in brackets for the Chinese product as well.

**Responses:** Thank you very much for this suggestion. Averaged SCD taken from the operational product on 06 August 2018 over China ($0.85 \pm 0.69 \times 10^{16}$ molec cm$^{-2}$) is on average 4.49 % lower than our retrieval ($0.89 \pm 0.61 \times 10^{16}$ molec cm$^{-2}$).

**Changes in manuscript:** L353-355, P16 in the revised version.

L228. It is not clear how using the BOAS HCHO SCDs reduces the overestimation if changing SCD retrieval method only shows a tiny effect of 0.15%? There is a contradiction here.

**Responses:** Thank you very much for this suggestion. We added the error bar in Table 3 and find the conclusion that changing SCD retrieval method only shows a tiny effect of 0.15% is less rigorous. The conclusion is changed into "In summer, using different SCD retrieval methods results a difference of 7.00% ($\pm$ 1.71%, $\pm$ Error) from the TROPOMI operational HCHO VCD. The result shows that using the BOAS HCHO SCDs reduced the underestimation in summer and overestimation in winter of the operational product (Table 5)."

**Changes in manuscript:** L359-362, P16 in the revised version.

L228: The mean random errors relative to BOAS are mentioned. Can you give a definition? And where are those errors presented in the paper?

**Responses:** Thank you very much for this suggestion. Uncertainty analysis is added in Sect. 4 in the revised version. The Random uncertainties can be approximated by the root mean square (RMS) of spectral fitting residual, the degrees of freedom, and the diagonal term of the covariance matrix for HCHO ($C_{j,j}$):

$$\sigma^2_{N_{S,rand}} = RMS^2 \frac{m}{m-n} C_{j,j} C_{j,j} \qquad (A2)$$

Where m is the number of spectral pixels and n is the number of fitted parameters.

**Changes in manuscript:** L255-259, P12 in the revised version.

**AMF calculation**

Table 3: This table is difficult to understand. The presentation of the numbers can be improved. The legend says that NMBs between satellite and MAX-DOAS are provided, but it seems to be more than that (NMB s1, s2). Error bars should be added. It would be more relevant to separated numbers for winter and summer periods.

**Responses:** Thank you very much for this suggestion. The legend of Table 3 is changed into "NMBs between TROPOMI HCHO VCDs with different retrieval settings (NMB $_{S1,S2}$) and NMBs between TROPOMI and MAX-DOAS observations (NMB$_{S1,M}$). TROPOMI HCHO VCDs are calculated with four different settings, (1) operational retrieval setting (2) replacing DOAS SCDs using BOAS SCDs in the operational product, (3) changing the a priori profiles from TM5 to regional WRF-Chem simulations in the operational product and (4) both (2) and (3) changes in the operational product. The error bars are also presented. All values are in %.".

The standard error (SE) of the NMBs is calculated by dividing the standard deviation (SD) by the square root of day numbers. The two times the standard deviation (95% confidence interval (Streiner, 1996)) is regarded as the error of the NMBs in this study. The NMBs larger than the error are considered statistically significant. The error of NMB is calculated following Eq.

(A1). The NMBs in summer and winter are also presented in Table 5.

**Changes in manuscript:** L302-306, P14 and Table 5 in the revised version.

Figure 5:

● Profiles are shown at the more urban CAMS station. It would be interesting to also show a suburban station, in order to detect the gain in spatial resolution.

**Responses:** Thank you very much for this suggestion. Profiles comparison at UCAS site is added in Figure 8 in the revised version. The improvement of WRF-Chem simulations at urban site is more significant than suburban site, which is mainly related to the finer spatial resolution and more up to date emission inventory over China used in the simulations. The bias between simulated and measured HCHO profiles at urban site is larger than that at suburban site, which is mainly due to smaller spatial gradient over suburban areas.

**Changes in manuscript:** L378-381, P17 and Figure 8 in the revised version.

● How many profiles are averaged? What is the spatial resolution?

**Responses:** Thank you very much for this suggestion. In AMF calculations, both WRF-Chem and TM5-MP simulations are interpolated to TROPOMI spatial resolution. Interpolated WRF-Chem and TM5-MP simulations within 20 km of the MAX-DOAS site are spatially averaged to compare with MAX-DOAS profiles. MAX-DOAS profiles are temporally averaged in the period 13:30-14:30 (Local Time) within ±1 h around the TROPOMI overpass time.

**Changes in manuscript:** L366-370, P16 in the revised version.

L249: The operational data are filtered using the QA value. Is the same selection applied to the improved Chinese product? If not, which selection is applied?

**Responses:** Thank you very much for this suggestion. TROPOMI pixels in our retrieval and operational product are both filtered for intensity-weighted cloud fraction smaller than 0.3, root mean square of spectral fit residual (RMS) smaller than $10^{-3}$, AMF larger than 0.1 and SZA smaller than 70°, quality assurance value (QA value) larger than 0.55 and successful SCD retrieval.

**Changes in manuscript:** L308-311, P14 in the revised version.

L255: Validation results are discussed at the 3 sites using correlation, slope and offset. Looking at those 3 parameters, mainly the offset is improved compared to the operational product. Correlations are almost identical. This needs to be discussed more in detail, related to the observed offset in the AMFs.

**Responses:** Thank you very much for this suggestion. The difference in validation results of our retrieval and operational product is caused by not only the AMFs but also the SCD retrieval. The SCD retrieval can effect HCHO VCDs largely by 7.00% ( ± 1.71%, ± Error) in summer (Table 3 in the revised version). Using the BOAS HCHO SCDs help reduce the underestimation in summer and overestimation in winter of the operational product (Table 5).

L263: Please explain how the MAX-DOAS are used to recompute the AMFs. Do you use the averaging kernels? The same

needs to be done with the operational product.

**Responses:** Thank you very much for this suggestion. We didn't use the averaging kernels in calculation of AMFs using MAX-DOAS measurements as a priori file. We have recalculated HCHO VCD for the operational product by using MAX-DOAS HCHO profiles as a priori profiles. The Pearson correlation coefficient (R) between the recalculated operational product and MAX-DOAS HCHO VCD decreases by 0.02 to 0.79. The slope of the regression line increases by 0.19 to 0.84 with offset reduces by $0.24 \times 10^{16}$ molec cm$^{-2}$ to $0.15 \times 10^{16}$ molec cm$^{-2}$.

**Changes in manuscript:** L406-409, P18 and Figure 10 in the revised version.

L272: The vertical profiles simulated by WRF-Chem are similar to the one measured by MAX-DOAS in  winter !

**Responses:** Thank you very much for this suggestion. Changed

L273: The underestimation of both retrievals in summer time are similar. 9.96% versus 10.88% is not significant. Please add error bars. Only the differences in winter time are significant.

**Responses:** Thank you very much for this suggestion. We recalculated the NMBs in summer and added the error bars calculated following Eq. A(1). The sentence is changed to "Therefore, our retrieval shows slightly better agreement with the ground based measurements (underestimation of $15.81 \pm 2.71$ %) compared to the operational product (underestimation of $18.33 \pm 3.10$ %) during summer."

**Changes in manuscript:** L427-428, P19 in the revised version.

**Section 4.3**

Not much useful information is given in this short paragraph. I suggest extending with a comparison with maps of VCD from the operational product, for the 4 seasons.

**Responses:** Thank you very much for this suggestion. The spatial distribution of the improved and operational HCHO VCDs is compared in Section 5.4 in the revised version. Operational product shows similar spatio-temporal distribution with our retrieval over China while the absolute values are (slight) smaller than our retrieval (Figure 12). In summer, hotspots can be observed over BTH, YRD, PRD, Shandong province, Henan province, Wuhan (Hubei's provincial capital), SCB and cities along Fen nutrient-laden valley in Shaanxi and Shanxi provinces in our retrieval. These hotspot patterns are strongly correlated to the population density and industrial emission pattern indicating a significant anthropogenic contribution. These hotspots are less obvious in the operational product in summer. The distribution of high HCHO VCDs observed in our retrieval over the Xinjiang Uygur Autonomous Region is related to the unique topography and industrial areas. The total industrial VOCs emission over the Xinjiang Uygur Autonomous Region is higher than Shanxi province (Zheng et al., 2017).

**Changes in manuscript:** L473-474 and L477-480, P21 and Figure 12 in the revised version.

**Conclusion**

As for the abstract and the title, the conclusions need to be redirected towards the real content of the paper, which is the use of a regional model to compute the AMFs, and the validation at 3 sites in Beijing.

**Responses:** Thank you very much for this suggestion. The abstract and conclusions are rewrote.

Chance, K. V., and Spurr, R. J.: Ring effect studies: Rayleigh scattering, including molecular parameters for rotational Raman scattering, and the Fraunhofer spectrum, Appl Optics, 36, 5224-5230, https://doi.org/10.1364/AO.36.005224, 1997.

Li, M., Zhang, Q., Kurokawa, J., Woo, J., He, K., Lu, Z., Ohara, T., Song, Y., Streets, D. G., and Carmichael, G. R.: MIX: a mosaic Asian anthropogenic emission inventory under the international collaboration framework of the MICS-Asia and HTAP, Atmos. Chem. Phys., 17, 935-963, 2017.

Liu, F., Zhang, Q., Tong, D., Zheng, B., Li, M., Huo, H., and He, K. B.: High-resolution inventory of technologies, activities, and emissions of coal-fired power plants in China from 1990 to 2010, Atmos. Chem. Phys., 15, 13299-13317, 2015.

Peng, L., Zhang, Q., Yao, Z., Mauzerall, D. L., Kang, S., Du, Z., Zheng, Y., Xue, T., and He, K.: Underreported coal in statistics: A survey-based solid fuel consumption and emission inventory for the rural residential sector in China, Applied Energy, 235, 1169-1182, 2019.

Streiner, D. L.: Maintaining Standards: Differences between the Standard Deviation and Standard Error, and When to Use Each, The Canadian Journal of Psychiatry, 41, 498-502, https://doi.org/10.1177/070674379604100805, 1996.

Zheng, C., Shen, J., Zhang, Y., Huang, W., Zhu, X., Wu, X., Chen, L., Gao, X., and Cen, K.: Quantitative assessment of industrial VOC emissions in China: Historical trend, spatial distribution, uncertainties, and projection, Atmos Environ, 150, 116-125, https://doi.org/10.1016/j.atmosenv.2016.11.023, 2017.

Zheng, B., Huo, H., Zhang, Q., Yao, Z., Wang, X., Yang, X., Liu, H., and He, K. B.: High-resolution mapping of vehicle emissions in China in 2008, Atmos. Chem. Phys., 14, 9787-9805, 2014.

---

## Author Response (AR2)

Dear editor,

We thank the editor for the constructive comments and suggestions. We have carefully addressed all the comments, a point to point based response to the comments are below for your consideration.

Comments to the Author:

Main points

1. The comparison of the 'improved' product and the operational product still needs clarifications. The discussion of the results and the reporting of the statistics of the comparisons still needs to be made more transparent.

• Table 5: The current caption, column headers and row headers are confusing. The use of the subscriptsS1, S2 and M, and "vs" is confusing. Clarify what exactly the reported numbers mean.

**Responses:** Thank you very much for the suggestion. We have improved the column headers and row headers. The caption of Table 5 is revised to "Normalized mean biases (NMBs) between the updated and operational TROPOMI HCHO VCDs ($NMB_{U,O}$) and NMBs between TROPOMI and MAX-DOAS observations ($NMB_{T,M}$). TROPOMI HCHO VCDs are updated with three different settings, (1) replacing DOAS SCDs by BOAS SCDs in the operational product, (2) changing the a priori profiles from TM5-MP to regional WRF-Chem simulations in the operational product and (3) both (1) and (2) changes in the operational product. The error bars ($\pm Error_{NMB}$) are also presented. The NMBs and theirs errors are calculated following Eq. (17) and Eq. (18) respectively. All values are in %."

• Which numbers support the statement that the present results compare better with the ground-based reference? How are the correlations coefficients, slopes and intercept of comparisons shown in Figure 10 to be understood in this regard?

**Responses:** Thank you very much for this suggestion. In Sect. 5.2, we discussed the comparison between TROPOMI HCHO VCDs with MAX-DOAS measurements in two aspects. On the one hand, we compared coefficients, slopes and intercepts of linear regression between TROPOMI and MAX-DOAS HCHO VCDs on page 18 line 402-410. The results show that our retrieval shows better agreement with MAX-DOAS measurements with higher Pearson correlation coefficient and the regression line is closer to 1:1 reference line compared to the operational product. On the other hand, we compared normalized mean biases (NMBs) between TROPOMI and MAX-DOAS HCHO VCDs on page 19 line 423-427. Our retrieval also shows smaller NMBs than operational product.

**Changes in manuscript:** We have added the description how the linear fitting results shown in Figure 10 support the statement that our results compare better with the ground-based reference on Page 18 Line 408-409 ("higher Pearson correlation coefficient and the regression line is closer to 1:1 reference line").

• Please clarify whether the improved HCHO product yields better slant column densities as compared to the operational product, as stated in the abstract (line 19). Which numbers support this statement? How are the correlations coefficients, slopes and intercept of comparisons shown in Figure 10 to be understood in this regard?

**Responses:** Thank you very much for this suggestion. In Sect. 5.1.1, we compared SCDs and DSCDs of our retrieval and operational product. The results show that the BOAS technique is less sensitive to measurement noise and the mean random errors relative to BOAS are about 22 % lower

than DOAS at three MAX-DOAS sites (Line 325-335, Page 15 and Line 368-369, Page 16).

In Sect. 5.2, we have recalculated HCHO VCDs for the operational product and our retrieval by using MAX-DOAS HCHO profiles as a priori profiles. The recalculated HCHO VCDs are compared with MAX-DOAS measurements in Fig. 10. The difference between recalculated our and operational VCDs are from SCD retrieval. The recalculated our VCDs (blue lines in Fig. 10) shows better improvements than the recalculated operational VCDs (magenta lines in Fig. 10), also indicates that the BOAS spectral retrieval of SCDs agree better to the ground based observations over China (stated on page 18 line 417-420).

**Changes in manuscript:** We have clarified the improvements of our SCD retrieval on page 1 line 20-21 in the abstract.

• It is stated that the use of prior information in the aerosol profile from MAX-DOAS improves the data; Figure 10: it appears that the use of this information does not improve the data.

**Responses:** Thank you very much for the comment. Our results show that considering aerosol in the AMF calculations does not improve the agreement with ground based measurements (Line 468-469, Page 10-11).

• Line 356-358: Clarify what is meant with the "control variable method" that is used to evaluate the improvement in terms of SCD. It is not clear what is meant with "we use SCD in our retrieval and AMFs in operational retrieval …"

**Responses:** Thank you very much for your comment. The "control variable method" means we replaced one variable (SCD or AMF) at a time in the operational and calculated the difference. We have further clarified this point on page 16 line 361-362.

2. The caption of Table 3 is unclear. It seems that the row "this study" refers to a reference case, and that the rows further below report performance values for cases where specific settings are modified with respect to the reference case (while all other settings are kept unchanged). Please explain in the caption what RMS and the sigma value mean. Clarify to which parameter they belong (SCD or DSCD). What is meant with "4th baseline and scaling polynomials": a polynomial of 4th order? What is meant with "TROPOMI ISRF Calibration Key Data (CKD) v1.0.0"? is here an alternative ISRF employed?

Responses: Thank you very much for your comment. The caption of Table 3 is changed to "Parameter effects on daily mean HCHO DSCDs and SCDs, ±1 standard deviation of DSCDs ($\pm 1\sigma_D$) and SCDs ($\pm 1\sigma_S$), and mean root mean square (RMS) of spectral fitting residual on 06 August 2018 of the region between 73° E and 130° E, and 18° N and 54° N. Units of DSCDs, SCDs, $\pm 1\sigma_D$ and $\pm 1\sigma_S$ are $\times 10^{16}$ molec cm$^{-2}$. RMS units are $\times 10^{-4}$. The detailed retrieval settings for reference case are listed in the column with the header of "Our retrieval" in Table 1. The rows below "Reference case" with specific settings are modified with respect to the reference case while all other settings are kept unchanged."

The "4th baseline and scaling polynomials" means 4th order baseline and scaling polynomials. The baseline and scaling polynomials are expressed in Eq. (4). As shown in Table 1, TROPOMI ISRF Calibration Key Data (CKD) v1.0.0 is used in operational product, while TROPOMI ISRF Calibration Key Data (CKD) v3.0.0 is used in Reference case.

3. Line 302: the introduction of the Standard Error (SE) is not understood. Is eq 18 defining this Standard Error (SE)? The quantity on the left hand side of Eq 18 should be labelled by something more specific than "Error".

**Responses:** Thank you very much for your comment. The sentence on page 14 line 307-308 has been changed to "The standard error (SE) is calculated by dividing the standard deviation (SD) by the square root of the sample size ($SE = SD/\sqrt{n}$)." The Eq. (18) defines the error of the NMBs which is two times SE. The Eq. (18) is changed to:

$$\text{Error}_{NMB} = 2 \times \text{SE} = 2 \times \sqrt{\frac{1}{n(n-1)} \frac{\sum_{i=1}^{i=n}\left(V_T(i) - V_M(i) - \overline{V_T(i) - V_M(i)}\right)^2}{\left(\sum_{i=1}^{i=n} V_M(i)/n\right)^2}} \times 100\% \qquad (18)$$

4. The reporting of values with associated uncertainties is unclear and made using various different notations. E.g. the expressions "0.57 (± 0.68, ± SD) ×10^16 molec cm-2" (Line 334), "0.21± 0.41×10^16 molec cm-2, average± SD" (Line 335/335), and "7.00 % (± 1.71 %, ± Error)" (Line 360) are not clear. What should the reader do with the SD and Error? Please clarify and make consistent (the notation used in "0.36 (± 0.60) ×10^16 molec cm-2" (Line 334/335) is looks ok and will be understood by the reader.

**Responses:** Thank you very much for your comment. The uncertainties in the brackets throughout the revised version are made consistent and are calculated using two times standard error.

5. Line 268: The "first order Taylor series approach for O3 SCDs" needs to be explained. If the approach by Puķīte et al. (2010) is meant, this needs to be referenced (Puķīte, J., Kühl, S., Deutschmann, T., Platt, U., and Wagner, T.: Extending differential optical absorption spectroscopy for limb measurements in the UV, Atmos. Meas. Tech., 3, 631-653, 2010.)

**Responses:** Thank you very much for your comment. The reference (Puķīte, J., Kühl, S., Deutschmann, T., Platt, U., and Wagner, T.: Extending differential optical absorption spectroscopy for limb measurements in the UV, Atmos. Meas. Tech., 3, 631-653, 2010.) is now cited (Page 12, Line 271-273, and Table 3).

Minor Editorials
Legend S5: The caption indicates that the RMS data from the operational (DOAS) product are depicted in the left and the data from the present study (BOAS) in the right panel. The legend suggests the opposite.

**Responses:** Done.

Figure 10 legend of panel f. Typo: "Operatioanl"     "Operational"; Typo: legend of magenta and pink have the same text "(a-priori from MAX-DOAS)"

**Responses:** Done.

Table 5 caption: Expand the acronym "NMB", and "SE"
Responses: Thank you very much for your comment. The description "The NMBs and their s errors are calculated following Eq. (17) and Eq. (18) respectively." is added in the caption of Table 5.

The symbol Ns is used for both SCD and DSCD; please use different symbols.

**Responses:** Thank you very much for your comment. The introductions of $\sigma_{N_S}$ and $\sigma_{N_{S,0}}$ in Eq. (11) are wrong, and we have modified them in the revised version ($\sigma_{N_S}$ is the uncertainties on DSCD and $\sigma_{N_{S,0}}$ is the uncertainties on DSCD in the reference sector). The subtitle of Sect. 4.1 is changed to "Uncertainties in DSCDs".

**Changes in manuscript:** Line 248-249, Line 252-253, Page 11 and Line 265-267, Page12 in the revised version.

Line 234: what is introduced as an "SCD correction" is, according to equations 9 and 10, a "DSCD correction". Proposed to rename.

**Responses:** Thank you very much for your comment. The "SCD correction" is changed to "DSCD correction"

**Changes in manuscript:** Line 234-238, Page 10-11 in the revised version.

Line 256: Introduce the covariance matrix. Clarify the elements of the state vector.

**Responses:** Thank you very much for your comment. The covariance matrix ($C_{j,j}$) is calculated from the Jacobian of the forward model corresponding to the fitting parameters (Chan Miller et al., 2014).

**Changes in manuscript:** Line 260-262, Page 12 in the revised version.

Equation 14: Introduce the index j. Clarify how the expression is to be evaluated, which value for j is to be taken. If the term Cj,j is supposed to be squared please use the square symbol.

**Responses:** Thank you very much for your comment. j is index of HCHO cross section in fitting parameters and Eq. (14) is changed to "$\sigma_{N_{S,rand}}^2 = RMS^2 \frac{m}{m-n} C_{j,j}{}^2$".

**Changes in manuscript:** Line 259-261, Page 12 in the revised version.

Except the changes in response to the editor's comments, we also have implemented additional changes. These changes are listed below.

1. Added a references for WRF-Chem simulations in China (Su et al., 2017; Zhang et al., 2019; Zhang et al., 2020) in Sect. 2.4 (Line 108-109, Page 5)

2. Updated the financial support by adding a project foundation from China Postdoctoral Science Foundation (2020TQ0320) in the Acknowledgments (Line 530-531, Page 23).

3. Updated the legend of Fig. 1 and Fig. 9

[revised manuscript text omitted]